# CONDITIONAL DIFFUSION MODELS WITH CLASSIFIER-FREE ITERATIVE GUIDANCE

## ABSTRACT

Classifier-Free Guidance (CFG) is a widely used technique for improving conditional diffusion models by linearly combining the outputs of conditional and unconditional denoisers. While CFG enhances visual quality and improves alignment with prompts, it often reduces sample diversity, leading to a challenging trade-off between quality and diversity. To address this issue, we make two key contributions. First, CFG generally does not correspond to a well-defined denoising diffusion model (DDM). In particular, contrary to common intuition, CFG does not yield samples from the target distribution associated with the limiting CFG score as the noise level approaches zero—where the data distribution is tilted by a power $w > 1$ of the conditional distribution. We identify the missing component: a Rényi divergence term that acts as a repulsive force and is required to correct CFG and render it consistent with a proper DDM. Our analysis shows that this correction term vanishes in the low-noise limit. Second, motivated by this insight, we propose a novel sampling procedure to draw samples from the desired tilted distribution. This method starts with an initial sample from the conditional diffusion model without CFG and iteratively refines it, preserving diversity while progressively enhancing sample quality. We evaluate our approach on image and text-to-audio generation, showing consistent improvements over CFG across all metrics.

## 1 INTRODUCTION

Diffusion models (Sohl-Dickstein et al., 2015; Song & Ermon, 2019; Ho et al., 2020) have emerged as a powerful framework for generative modeling, achieving state-of-the-art performance in a variety of tasks such as text-to-image generation (Rombach et al., 2022; Podell et al., 2023), video (Blattmann et al., 2023) and audio generation (Kong et al., 2020b). The success of these models can be partly attributed to their ability to produce powerful conditional generative models through guidance. Among various methods, Classifier-Free Guidance (CFG) (Ho & Salimans, 2022) has become a very popular method for sample generation, as it allows strengthening the alignment to the conditioning context via a temperature-like parameter $w > 1$ that acts as a guidance scale. Beyond improving alignment, CFG plays a crucial role in ensuring high-quality samples, as unguided diffusion models typically produce subpar outputs, limiting their practical use (Dieleman, 2022b; Karras et al., 2024a).

CFG is implemented by linearly combining the outputs of a conditional denoiser—one that takes the conditioning context as additional input—and an unconditional one, with linear coefficients parameterized by a scaling $w$. Previous works have demonstrated that although this simple combination effectively enhances the performance of diffusion models, it often results in overly simplistic images and a substantial reduction in output diversity (Karras et al., 2024a; Kynkäänniemi et al., 2024). This is essentially due to the fact that a linear combination of the conditional and unconditional denoisers does not yield a valid denoiser, thereby breaking the correspondence with any underlying diffusion process (Karras et al., 2024a; Bradley & Nakkiran, 2024; Chidambaram et al., 2024).

**Contributions.** Below, we summarize our two key contributions

**1** We revisit guidance from a probabilistic perspective, focusing on the problem of sampling from the data distribution tilted by the conditional distribution of the context given the data raised to some power $w > 1$. We show that the denoisers used in CFG are neither posterior-mean estimators for

$$X_0^{(0)} \qquad X_0^{(1)} \qquad X_0^{(2)} \qquad\qquad X_0^{(0)} \qquad X_0^{(1)} \qquad X_0^{(2)}$$

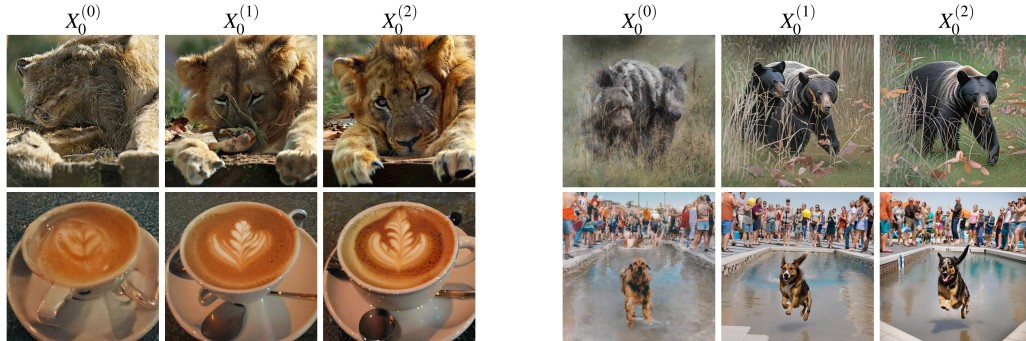

Figure 1: Illustration of sample refinement across iterations. Left, samples generated using EDM-XXL for two `ImageNet` classes: 291 (top) and 967 (bottom). Right, samples generated using Stable Diffusion XL (SDXL) for the prompts: *"A black bear walking in the grass and leaves."* (top) and *"A dog jumping through the air above a pool of water that has been marked for distance, with people watching in the distance."* (bottom). Each row displays an initial sample $X_0^{(0)}$ alongside two subsequent iterates $X_0^{(1)}, X_0^{(2)}$.

this tilted target distribution nor valid denoisers for any other data distribution; see Example 1. We then prove (Proposition 1) that for CFG to target the tilted distribution of interest, it must include an additional term—the gradient of a Rényi divergence—which acts as a repulsive force that encourages output diversity, as illustrated in Figure 2. Finally, as an additional contribution of independent interest, we derive a new expression for the tilted target scores that involves two different noise levels, providing a theoretical justification for recent guidance methods (Sadat et al., 2025; Li et al., 2024).

**2** These theoretical insights motivate our second contribution: an iterative sampling procedure referred to as CLASSIFIER-FREE ITERATIVE GUIDANCE (CFIG), which generates samples from the tilted distribution. As summarized in Algorithm 1, it begins by first drawing a sample from the conditional distribution, without CFG, then iteratively refines it via alternating noising and CFG-denoising steps. We illustrate the iterative refinement procedure in Figure 1. A key advantage of this approach is the ability to preserve the sample diversity of the prior model while improving generation quality. We then analyze the algorithm and quantify the bias introduced by omitting the Rényi divergence term in the Gaussian setting (Proposition 4).

We validate CFIG on class conditional image generation, text-to-image generation, and text-to-audio generation, showing that it significantly outperforms CFG and is competitive with INTERVAL-CFG, a state-of-the-art method recently proposed in Kynkäänniemi et al. (2024).

## 2 BACKGROUND

**Diffusion models.** Denoising diffusion models (DDMs) (Song & Ermon, 2019; Song et al., 2021a;b) define a generative procedure targeting a data distribution $p_0$. It proceeds by first sampling from a highly noised distribution $p_{\sigma_{max}}$ that practically resembles a Gaussian, and then iteratively sampling through a sequence of progressively less noisy distributions $p_\sigma$ for decreasing noise levels $\sigma < \sigma_{max}$. The distribution $p_\sigma$ is the $\mathbf{x}_\sigma$-marginal of $p_{0,\sigma}(\mathbf{x}_0, \mathbf{x}_\sigma)$, the joint distribution of the $(X_0, X_\sigma) :=(X_0, X_0 + \sigma Z)$, where $X_0$ and $Z$ are drawn independently from $p_0$ and $\mathcal{N}(0, \mathbf{I})$. Formally, by letting $q_{\sigma|0}(\mathbf{x}_\sigma|\mathbf{x}_0) := \mathrm{N}(\mathbf{x}_\sigma; \mathbf{x}_0, \sigma^2 \mathbf{I})$, the marginals are defined as $p_\sigma(\mathbf{x}_\sigma) := \int p_{0,\sigma}(\mathbf{x}_0, \mathbf{x}_\sigma)\,\mathrm{d}\mathbf{x}_0 = \int q_{\sigma|0}(\mathbf{x}_\sigma|\mathbf{x}_0)p_0(\mathbf{x}_0)\,\mathrm{d}\mathbf{x}_0$. Following Karras et al. (2022), exact sampling from the target $p_0$ can be achieved by solving, backwards in time over $[0, \sigma_{max}]$ and starting from $\mathbf{x}_{\sigma_{max}} \sim p_{\sigma_{max}}$,

$$\mathrm{d}\mathbf{x}_\sigma/\mathrm{d}\sigma = \big(\mathbf{x}_\sigma - D_\sigma(\mathbf{x}_\sigma)\big)/\sigma \,, \quad \text{where} \quad D_\sigma(\mathbf{x}_\sigma) = \int \mathbf{x}_0\, p_{0|\sigma}(\mathbf{x}_0|\mathbf{x}_\sigma)\,\mathrm{d}\mathbf{x}_0 \,. \quad (2.1)$$

$D_\sigma$ is the denoiser at noise level $\sigma$; see Karras et al. (2022, Equations 1 and 3). The drift in the probability-flow ODE (PF-ODE) (2.1) can be identified as the score function $(\mathbf{x}, \sigma) \mapsto \nabla \log p_\sigma(\mathbf{x})$, since using Tweedie's formula (Robbins, 1956), it follows that $D_\sigma(\mathbf{x}) = \mathbf{x} + \sigma^2 \nabla \log p_\sigma(\mathbf{x})$.

The practical implementation of this generative process involves first estimating $D_\sigma$ using parametric approximations $D_\sigma^\theta$, $\theta \in \Theta$, trained by minimizing the weighted denoising loss $\theta \mapsto \int \mathbb{E}\big[\|X_0 - D_\sigma^\theta(X_\sigma)\|^2\big] \lambda(\sigma)\,\mathrm{d}\sigma$, where $\lambda$ denotes a probability density function over $\mathbb{R}_+$ which assigns weights to noise levels (Karras et al., 2022). This training procedure also provides parametric

approximations of the score function. Therefore, once $(D_\sigma^\theta)_{\sigma \geq 0}$ are trained, a new approximate sample $\hat{X}_0$ from $p_0$ can be drawn by fixing a decreasing sequence $(\sigma_t)_{t=T}^0$ of noise levels and then solving the ODE (2.1) backwards from $\sigma_T$ to $\sigma_0$ using an integration method such as Euler or Heun starting from $\hat{X}_{\sigma_T} \sim \mathcal{N}(0, \sigma_T^2 \mathbf{I})$ (Karras et al., 2022). In particular, the Euler method corresponds to the Denoising Diffusion Implicit Model (DDIM) scheme of Song et al. (2021a), with updates

$$\hat{X}_{\sigma_t} = (1 - \sigma_t/\sigma_{t+1}) D_{\sigma_{t+1}}^\theta (\hat{X}_{\sigma_{t+1}}) + (\sigma_t/\sigma_{t+1}) \hat{X}_{\sigma_{t+1}} . \qquad (2.2)$$

**Conditional DDMs.** This generative procedure also extends to sampling conditionally on $C \in \mathcal{C}$, where $\mathcal{C}$ can be a collection of classes or text-prompts. Denote by $\bar{p}_0(\mathbf{c}, \mathbf{x}_0)$ the joint density of $(C, X_0)$. The goal is to approximately sample from the conditional distribution $p_0(\mathbf{x}_0 | \mathbf{c}) \propto \bar{p}_0(\mathbf{c}, \mathbf{x}_0)$, given training samples from the joint distribution. Similar to the unconditional case, we introduce the joint distribution $\bar{p}_{0,\sigma}(\mathbf{c}, \mathbf{x}_0, \mathbf{x}_\sigma) \propto \bar{p}_0(\mathbf{c}, \mathbf{x}_0) q_{\sigma|0}(\mathbf{x}_\sigma | \mathbf{x}_0)$, where only the data $X_0$ is noised. Conditional DDM sampling from $p_0(\mathbf{x}_0 | \mathbf{c})$ thus reduces to estimating the conditional denoiser defined by $D_\sigma(\mathbf{x}_\sigma | \mathbf{c}) := \int \mathbf{x}_0 \, p_{0|\sigma}(\mathbf{x}_0 | \mathbf{x}_\sigma, \mathbf{c}) \, \mathrm{d}\mathbf{x}_0$, by using a parametric family $\{(\mathbf{x}, \mathbf{c}, \sigma) \mapsto D_\sigma^\theta(\mathbf{x} | \mathbf{c}) : \theta \in \Theta\}$ and minimizing the loss $\theta \mapsto \int \mathbb{E} \left[ \|X_0 - D_\sigma^\theta(X_\sigma | C)\|^2 \right] \lambda(\sigma) \, \mathrm{d}\sigma$, where the expectation is taken with respect to the joint law of $(C, X_0, X_\sigma) \sim \bar{p}_{0,\sigma}$. As noted in (Ho & Salimans, 2022; Karras et al., 2024b), it is possible to learn simultaneously a conditional and unconditional denoiser by augmenting $\mathcal{C}$ with a null context $\varnothing$ to represent the unconditional case. Then, during training, $(C, X_0)$ is first sampled from $\bar{p}_0$ and $C$ is replaced by $\varnothing$ with probability $p_{\mathsf{uncd}}$; a procedure also referred to as *conditioning dropout* (Dieleman, 2022a). Henceforth, we denote the unconditional denoisers by $D_\sigma(\cdot | \varnothing) = D_\sigma$,

**Classifier-Free Guidance (CFG).** In many complex applications—such as text-to-image synthesis and audio generation—directly using the conditional diffusion models often results in samples that lack the perceptual quality of the training data. This discrepancy is especially evident in terms of perceived realism, texture fidelity, and fine-grained detail. CFG (Ho & Salimans, 2022) has emerged as a standard approach to mitigate this issue, enhancing both the visual fidelity and the alignment of generated samples and the context $\mathbf{c}$. Yet, this improvement typically comes at the cost of reduced sample diversity. In CFG, the denoiser is defined as a linear combination of the conditional and unconditional denoisers:

$$D_\sigma^{\mathsf{cfg}}(\mathbf{x}_\sigma | \mathbf{c}; w) := w D_\sigma(\mathbf{x}_\sigma | \mathbf{c}) + (1 - w) D_\sigma(\mathbf{x}_\sigma) , \qquad (2.3)$$

where $w > 1$ is a *guidance scale*. To illustrate the impact of the scale $w$, we denote by $g_0(\mathbf{c} | \mathbf{x}_0) := \bar{p}_0(\mathbf{c}, \mathbf{x}_0)/p_0(\mathbf{x}_0)$ the conditional distribution of the context and set $g_\sigma(\mathbf{c} | \mathbf{x}_\sigma) := \int g_0(\mathbf{c} | \mathbf{x}_0) p_{0|\sigma}(\mathbf{x}_0 | \mathbf{x}_\sigma) \, \mathrm{d}\mathbf{x}_0$. By construction, the conditional distribution of $X_\sigma$ given the context is

$$p_\sigma(\mathbf{x}_\sigma | \mathbf{c}) \propto \int \bar{p}_{0,\sigma}(\mathbf{c}, \mathbf{x}_0, \mathbf{x}_\sigma) \, \mathrm{d}\mathbf{x}_0 \propto \int g_0(\mathbf{c} | \mathbf{x}_0) p_0(\mathbf{x}_0) q_{\sigma|0}(\mathbf{x}_\sigma | \mathbf{x}_0) \, \mathrm{d}\mathbf{x}_0 \propto g_\sigma(\mathbf{c} | \mathbf{x}_\sigma) p_\sigma(\mathbf{x}_\sigma) ,$$

where we used that $p_0(\mathbf{x}_0) q_{\sigma|0}(\mathbf{x}_0 | \mathbf{x}_\sigma) = p_\sigma(\mathbf{x}_\sigma) p_{0|\sigma}(\mathbf{x}_0 | \mathbf{x}_\sigma)$. Thus, Tweedie's formula implies that

$$D_\sigma(\mathbf{x}_\sigma | \mathbf{c}) = \mathbf{x}_\sigma + \sigma^2 \nabla \log p_\sigma(\mathbf{x}_\sigma | \mathbf{c}) = \mathbf{x}_\sigma + \sigma^2 \big( \nabla \log g_\sigma(\mathbf{c} | \mathbf{x}_\sigma) + \nabla \log p_\sigma(\mathbf{x}_\sigma) \big) . \qquad (2.4)$$

Substituting back into (2.3), we obtain $D_\sigma^{\mathsf{cfg}}(\mathbf{x}_\sigma | \mathbf{c}; w) = \mathbf{x}_\sigma + \sigma^2 \nabla \log p_\sigma^{\mathsf{cfg}}(\mathbf{x}_\sigma | \mathbf{c}; w)$, where

$$p_\sigma^{\mathsf{cfg}}(\mathbf{x}_\sigma | \mathbf{c}; w) \propto g_\sigma(\mathbf{c} | \mathbf{x}_\sigma)^w p_\sigma(\mathbf{x}_\sigma) . \qquad (2.5)$$

Hence, CFG modifies the conditional denoiser (2.4) solely through the guidance scale $w$ applied to the classifier score. With $w > 1$, CFG amplifies the influence of regions where $g_\sigma(\mathbf{c} | \mathbf{x}_\sigma)$ is large. In practice, this results in enhanced prompt alignment and improved perceptual quality of generated images, at the cost of reduced sample diversity (Kynkäänniemi et al., 2024).

Note that generally, $p_\sigma^{\mathsf{cfg}}(\mathbf{x}_\sigma | \mathbf{c}; w) \neq \int q_{\sigma|0}(\mathbf{x}_\sigma | \mathbf{x}_0) p_0^{\mathsf{cfg}}(\mathbf{x}_0 | \mathbf{c}; w) \, \mathrm{d}\mathbf{x}_0$. This discrepancy prompts a fundamental question: does incorporating the CFG denoiser (2.3) into the ODE (2.1) yield a sampling process that corresponds to a well-defined DDM? **Specifically, does there exist $\pi(\cdot | \mathbf{c}; w)$ such that for all $\sigma > 0$, $p_\sigma^{\mathsf{cfg}}(\mathbf{x}_\sigma | \mathbf{c}; w) = \int q_{\sigma|0}(\mathbf{x}_\sigma | \mathbf{x}_0) \pi(\mathbf{x}_0 | \mathbf{c}; w) \, \mathrm{d}\mathbf{x}_0$?** We show in the next example that this is not the case in general.

**Example 1.** *Let $p_0 = \mathcal{N}(0, 1)$ and $g_0(\mathbf{c} | \mathbf{x}_0) = \mathrm{N}(\mathbf{c}; \mathbf{x}_0, \gamma^2)$ is the one-dimensional Gaussian density with mean $\mathbf{x}_0$ and variance $\gamma^2$; then $p_\sigma^{\mathsf{cfg}}(\cdot | \mathbf{c}; w)$ is Gaussian with variance*

$$\mathrm{v}(w, \sigma^2) := \frac{(1 + \sigma^2) \gamma^2 + \sigma^2}{w/(1 + \sigma^2) + \gamma^2 + \sigma^2/(1 + \sigma^2)} . \qquad (2.6)$$

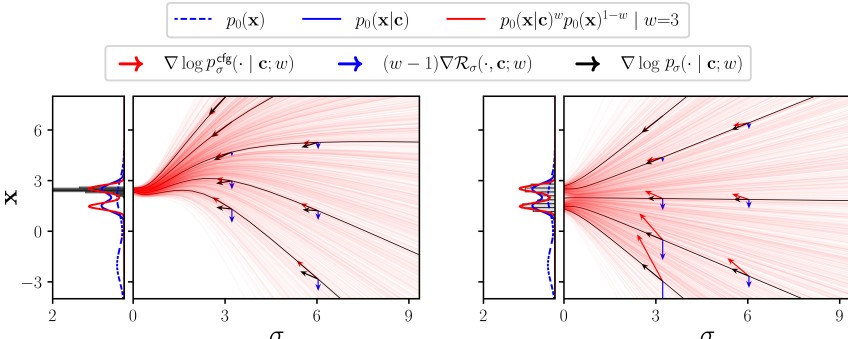

Figure 2: Left: DDIM sampling with CFG denoiser (2.3); Right: DDIM sampling with the ideal denoiser (3.2). The trajectories of 1000 particles are represented with thin red lines and 5 selected trajectories are displayed in black thick line along which scores are being depicted with arrows. The histogram of the simulated particles is represented in light gray. We also plot the ideal score (3.2) (arrow in black) with the contribution of both the CFG score (arrow in red) and the repulsive term arising from the Rényi divergence (arrow in blue).

*We now show that it does not exist $\pi(\cdot|\mathbf{c}; w)$ such that for all $\sigma > 0$,*

$$p_\sigma^{\mathsf{cfg}}(\mathbf{x}_\sigma|\mathbf{c}; w) = \int q_{\sigma|0}(\mathbf{x}_\sigma|\mathbf{x}_0)\pi(\mathbf{x}_0|\mathbf{c}; w)\,\mathrm{d}\mathbf{x}_0 \ . \tag{2.7}$$

*The proof is by contradiction. Suppose that (2.7) holds. Then we may show, by letting $\sigma$ tend to zero, that $\pi(\cdot|\mathbf{c}; w) = p_0^{\mathsf{cfg}}(\cdot|\mathbf{c}; w)$, where $p_0^{\mathsf{cfg}}(\mathbf{x}_0|\mathbf{c}; w) = \mathrm{N}(\mathbf{x}_0; w\mathbf{c}/(w + \gamma^2), \gamma^2/(w + \gamma^2))$; see Appendix B.3. In addition, let $(X_0, X_\sigma)$ be distributed according to the joint distribution with density $q_{\sigma|0}(\mathbf{x}_\sigma|\mathbf{x}_0)p_0^{\mathsf{cfg}}(\mathbf{x}_0|\mathbf{c}; w)$. Then, $\mathbb{V}(X_\sigma) = \sigma^2 + \gamma^2/(w + \gamma^2)$. However, by (2.7), $X_\sigma \sim p_\sigma^{\mathsf{cfg}}(\cdot|\mathbf{c}; w)$ which implies that $\mathrm{v}(w, \sigma^2) < \mathbb{V}(X_\sigma)$ for all $\sigma > 0$ and $w > 1$. Thus, we obtain a contradiction.*

## 3 ANALYZING THE BIAS IN CLASSIFIER-FREE GUIDANCE

As demonstrated in Example 1 and illustrated in Figure 2, CFG does not necessarily yield an approximate sample from $p_0^{\mathsf{cfg}}(\cdot|\mathbf{c}; w)$. We now focus on introducing a DDM specifically designed to target this distribution, which we from now on denote by $p_0(\cdot|\mathbf{c}; w)$—omitting the superscript cfg.

**Tilted distribution scores and denoisers.** Define the joint distribution $\bar{p}_{0,\sigma}(\mathbf{x}_0, \mathbf{x}_\sigma|\mathbf{c}; w) \propto g_0(\mathbf{c}|\mathbf{x}_0)^w p_{0,\sigma}(\mathbf{x}_0, \mathbf{x}_\sigma)$, of which $p_0(\cdot|\mathbf{c}; w)$ is the $\mathbf{x}_0$-marginal. Proposition 1 provides a simple and interpretable form of the scores of the smoothed marginals $p_\sigma(\mathbf{x}_\sigma|\mathbf{c}; w) = \int \bar{p}_{0,\sigma}(\mathbf{x}_0, \mathbf{x}_\sigma|\mathbf{c}; w)\mathrm{d}\mathbf{x}_0$. For $w > 1$, define the Rényi divergence of order $w$ of $p$ from $q$ (Van Erven & Harremos, 2014) as

$$\mathsf{R}_w(p\|q) := \frac{1}{w-1}\log\int\frac{p(\mathbf{x})^w}{q(\mathbf{x})^w}q(\mathbf{x})\,\mathrm{d}\mathbf{x} \ . \tag{3.1}$$

For ease of notation, we set $\mathcal{R}_\sigma(\mathbf{x}_\sigma, \mathbf{c}; w) := \mathsf{R}_w(p_{0|\sigma}(\cdot|\mathbf{x}_\sigma, \mathbf{c})\|p_{0|\sigma}(\cdot|\mathbf{x}_\sigma))$.

**Proposition 1.** *For any $\sigma > 0$, the scores associated with $p_\sigma(\cdot|\mathbf{c}; w)$ are*

$$\nabla \log p_\sigma(\mathbf{x}_\sigma|\mathbf{c}; w) = (w-1)\nabla\mathcal{R}_\sigma(\mathbf{x}_\sigma, \mathbf{c}; w) + \nabla\log p_\sigma^{\mathsf{cfg}}(\mathbf{x}_\sigma|\mathbf{c}; w) \ . \tag{3.2}$$

*Sketch of the proof.* We use that $p_\sigma(\mathbf{x}_\sigma|\mathbf{c}; w) \propto \int q_{\sigma|0}(\mathbf{x}_\sigma|\mathbf{x}_0)p_0(\mathbf{x}_0|\mathbf{c})^w p_0(\mathbf{x}_0)^{1-w}\,\mathrm{d}\mathbf{x}_0$ and that $q_{\sigma|0}(\mathbf{x}_\sigma|\mathbf{x}_0) = q_{\sigma|0}(\mathbf{x}_\sigma|\mathbf{x}_0)^w q_{\sigma|0}(\mathbf{x}_\sigma|\mathbf{x}_0)^{1-w}$. We then gather the terms under the powers $w$ and $1-w$ respectively and apply the Bayes' rule. The complete proof is provided in Appendix B.1. The decomposition (3.2) highlights that the CFG score $\nabla\log p_\sigma^{\mathsf{cfg}}(\mathbf{x}_\sigma|\mathbf{c}; w)$ differs from the *ideal score* $\nabla\log p_\sigma(\cdot|\mathbf{c}; w)$ by an additional term: the gradient of the Rényi divergence. To illustrate Proposition 1 and the bias introduced by CFG, we consider a one-dimensional toy model; see Figure 2. In this example, we compare the trajectories obtained by integrating the ODE (2.1) using the ideal denoiser—defined as $D_\sigma(\mathbf{x}_\sigma|\mathbf{c}; w) := \mathbf{x}_\sigma + \sigma^2\nabla\log p_\sigma(\mathbf{x}_\sigma|\mathbf{c}; w)$—and the CFG denoiser (2.3). The CFG trajectories are observed to collapse onto a single mode, neglecting other regions of significant probability mass, as observed in Kynkäänniemi et al. (2024).

**Intuition behind CFG bias.** The over-concentration of paths in Figure 2 stems from omitting the term $(w-1)\nabla\mathcal{R}_\sigma(\mathbf{x}_\sigma, \mathbf{c}; w)$, which acts as a *repulsive force*. Indeed, since $w > 1$, this term pushes the sample in the direction where the conditional and unconditional distributions $p_{0|\sigma}(\cdot|\mathbf{x}_\sigma, \mathbf{c})$ and $p_{0|\sigma}(\cdot|\mathbf{x}_\sigma)$ differ most, thereby counteracting over-concentration.

Our next result show that the CFG score $\nabla\log p_\sigma^{\mathsf{cfg}}(\cdot|\mathbf{c}; w)$ serves as a good approximation of the ideal score $\nabla\log p_\sigma(\cdot|\mathbf{c}; w)$ for small $\sigma$.

**Proposition 2.** *Under suitable assumptions on $p_0$ and $g_0$, it holds for all $w > 1$, $\mathbf{x} \in \mathbb{R}^d$, $\mathbf{c} \in \mathcal{C}$,*

$$\nabla\mathcal{R}_\sigma(\mathbf{x}, \mathbf{c}; w) = O(\sigma^2) \quad \text{as} \;\; \sigma \to 0 \; .$$

*Sketch of the proof.* We rely on the observation that for densities of the form $\pi_\sigma(\mathbf{x}_\sigma) = \int q_{\sigma|0}(\mathbf{x}_\sigma|\mathbf{x}_0)\,\pi(\mathbf{x}_0)\,\mathrm{d}\mathbf{x}_0$, it holds as $\sigma$ approaches 0 that $\nabla\log\pi_\sigma(\mathbf{x}_\sigma) = \nabla\log\pi(\mathbf{x}_0) + O(\sigma^2)$, a result that we establish in Lemma 1. Applying this lemma to $p_\sigma(\cdot|\mathbf{c}; w)$, $p_\sigma(\cdot|\mathbf{c})$, and $p_\sigma$ yields

$$\nabla\log p_\sigma(\mathbf{x}_\sigma|\mathbf{c}; w) = \nabla\log p_\sigma^{\mathsf{cfg}}(\mathbf{x}_\sigma|\mathbf{c}; w) + O(\sigma^2), \quad \sigma \to 0,$$

which enables to conclude after plugging the results in Equation (3.2). We provide the full proof in Appendix B.2. The CFG denoiser suffers from an intrinsic flaw due to the absence of the term $(w-1)\nabla\mathcal{R}_\sigma(\mathbf{x}_\sigma, \mathbf{c}; w)$, which plays a crucial role at high to medium noise levels in preserving the diversity of the generated samples. Conversely, the denoiser $D_\sigma(\cdot|\mathbf{c}; w)$ is well approximated by $D_\sigma^{\mathsf{cfg}}(\cdot|\mathbf{c}; w)$ in the low-noise regime, where the missing Rényi divergence term is effectively negligible. In Section 4, we present a new method for generating approximate samples from the target $p_0(\cdot|\mathbf{c}; w)$, which relies on using the CFG approximation of the denoiser exclusively in the low-noise regime below a given noise level $\sigma_*$. In this regime, the intractable Rényi divergence term can be safely omitted as we discuss in the next section.

**An alternative expression of the ideal score.** We now present a generalization of the score expression (3.2) that offers an novel alternative expression to $D_\sigma(\cdot|\mathbf{c}; w)$. For conciseness, we set $\mathcal{R}_{\sigma_1,\sigma_2}(\mathbf{x}, \mathbf{c}; w) \coloneqq \mathsf{R}_w(p_{0|\sigma_1}(\cdot|\mathbf{x}, \mathbf{c})\|p_{0|\sigma_2}(\cdot|\mathbf{x}))$, where $\sigma_1, \sigma_2 > 0$ are two noise levels.

**Proposition 3.** *Let $w > 1$. For all $\delta > 0$ and $\sigma > 0$, the scores associated with $p_0(\cdot|\mathbf{c}; w)$ are*

$$\nabla\log p_\sigma(\mathbf{x}_\sigma|\mathbf{c}; w) = (w-1)\nabla\mathcal{R}_{\sigma_-,\sigma_+}(\mathbf{x}_\sigma, \mathbf{c}; w) + w\nabla\log p_{\sigma_-}(\mathbf{x}_\sigma|\mathbf{c}) + (1-w)\nabla\log p_{\sigma_+}(\mathbf{x}_\sigma) \; ,$$

*where $\sigma_- \coloneqq \sigma\sqrt{w/(1+\delta)}$ and $\sigma_+ \coloneqq \sigma\sqrt{(w-1)/\delta}$.*

*Sketch of the proof.* The proof mirrors the argument in Proposition 1. We use that for any $\delta > 0$, $q_{\sigma|0}(\mathbf{x}_\sigma|\mathbf{x}_0) = q_{\sigma|0}(\mathbf{x}_\sigma|\mathbf{x}_0)^{1+\delta}\,q_{\sigma|0}(\mathbf{x}_\sigma|\mathbf{x}_0)^{-\delta}$ and then the key observation that

$$q_{\sigma|0}(\mathbf{x}_\sigma|\mathbf{x}_0)^{1+\delta} \propto q_{\sigma_-|0}(\mathbf{x}_\sigma|\mathbf{x}_0)^w \; , \quad q_{\sigma|0}(\mathbf{x}_\sigma|\mathbf{x}_0)^\delta \propto q_{\sigma_+|0}(\mathbf{x}_\sigma|\mathbf{x}_0)^{1-w} \; ,$$

which follows from the fact that the forward process is Gaussian. The full proof is provided in Appendix B.1. The key novelty of this generalized formula is that it involves score evaluations at two distinct noise levels, $\sigma_-$ and $\sigma_+$. By setting $\delta = w - 1$, we recover the original formula (3.2), while selecting $0 < \delta < w - 1$ yields the ordering $\sigma_- < \sigma < \sigma_+$. Similar to (3.2), the Rényi divergence term vanishes as $\sigma \to 0$, leading to the practical approximation $D_\sigma(\mathbf{x}_\sigma|\mathbf{c}; w) \approx wD_{\sigma_-}(\mathbf{x}_\sigma|\mathbf{c}; w) + (1-w)D_{\sigma_+}(\mathbf{x}_\sigma|\mathbf{c}; w)$ in this regime. The idea of performing guidance by combining denoisers at different noise levels has also been recently explored in Sadat et al. (2025); Li et al. (2024). Specifically, these works introduce the modified denoiser $wD_\sigma(\mathbf{x}_\sigma|\mathbf{c}) + (1-w)D_{\tilde{\sigma}}(\mathbf{x}_\sigma|\mathbf{c})$, where $\tilde{\sigma} \coloneqq \sigma + \Delta\sigma$ for some small increment $\Delta\sigma$. Remarkably, this formulation avoids the use of the unconditional denoiser employed in CFG, thereby allowing guidance solely through the conditional denoiser. Proposition 3 hence provide a theoretical ground for these alternative methods. We further discuss extensions of this score formulation in Appendix D.

## 4 ITERATIVE GUIDANCE

Our method consists in defining a Markov chain $(X_0^{(r)})_{r\in\mathbb{N}}$ that admits $p_0(\cdot|\mathbf{c}; w)$ as stationary distribution. We rely on a fixed noise level $\sigma_*$ that we assume to be small enough. The chain is generated recursively as follows. Given $X_0^{(r)}$ at stage $r$, $X_0^{(r+1)}$ is obtained by

---

**Algorithm 1** CFIG

---

1: **Require:** Guidance scales $w_0 \geq 1$ and $w > w_0$
2: **Require:** Number of repetitions $R$, total steps $T$, and initial steps $T_0$
3: **Require:** Standard deviations $\sigma_*$, $\sigma_{\max}$
4: $X_{\sigma_{\max}} \sim \mathcal{N}(0, \sigma_{\max}^2 \mathbf{I})$
5: $X_0 \leftarrow \texttt{ODE\_Solver}(X_{\sigma_{\max}}, D_\sigma^{\mathsf{cfg}}(\cdot|\mathbf{c}; w_0), T_0)$
6: **for** $r = 1$ **to** $R$ **do**
7: $\quad X_{\sigma_*} \leftarrow X_0 + \sigma_* Z, \quad$ where $Z \sim \mathcal{N}(0, \mathbf{I})$
8: $\quad X_0 \leftarrow \texttt{ODE\_Solver}(X_{\sigma_*}, D_\sigma^{\mathsf{cfg}}(\cdot|\mathbf{c}; w), \lfloor(T - T_0)/R\rfloor)$
9: **end for**
10: **Output:** $X_0$

---

Step 1) sampling an intermediate state $X_{\sigma_*}^{(r)} \sim q_{\sigma_*|0}(\cdot|X_0^{(r)})$;

Step 2) denoising it by integrating the PF-ODE (2.1) with the ideal denoiser $D_\sigma(\cdot|\mathbf{c}; w)$.

These updates can be compactly written as $X_0^{(r+1)} = F_{0|\sigma_*}\big(X_0^{(r)} + \sigma_* Z^{(r+1)}; w\big)$, where $(Z^{(r)})_{r \in \mathbb{N}}$ is a sequence of i.i.d. standard Gaussian random variables and $F_{0|\sigma_*}(\mathbf{x}_{\sigma_*}; w)$ is the solution to the PF-ODE with denoiser $D_\sigma(\cdot|\mathbf{c}; w)$ and initial condition $\mathbf{x}_{\sigma_*}$. We can verify that the associated Markov chain admits $p_0(\cdot|\mathbf{c}; w)$ as its *unique* stationary distribution under appropriate conditions; see Appendix B.3. Taking $\sigma_*$ small enough, we implement this scheme by using the CFG denoiser (2.3) instead of $D_\sigma(\cdot|\mathbf{c}; w)$ in Step 2) for integrating the PF-ODE (2.1).

*Initial distribution.* The initial sample $X_0^{(0)}$ is generated using an ODE solver applied to the PF-ODE (2.1) using either the plain conditional denoisers $D_\sigma^\theta(\cdot|\mathbf{c})$ or the CFG denoisers (2.3) with a moderate guidance scale $1 \leq w_0 \ll w$. In the case of image generation, this tends to provide initial samples that exhibit high diversity but low perceptual quality (Ho & Salimans, 2022; Karras et al., 2024a). These coarse samples are subsequently refined and sharpened in the later stages of the algorithm, where a larger guidance scale $w$ is used. Given a total budget of $T$ function evaluations and $R$ refinements stages, we allocate a number $T_0$ of steps—between $\lfloor T/3 \rfloor$ to $\lfloor T/2 \rfloor$—for generating the initial sample, with the remaining budget evenly distributed across the $R$ refinement stages.

CFIG *algorithm.* The proposed generation pipeline consists of two stages: (1) an initialization stage, where coarse samples are generated with weak guidance to preserve the same level of diversity as the target distribution; and (2) a refinement stage, where these samples are progressively improved through iterative noising-denoising stages. The full procedure, referred to as CFIG, is presented in Algorithm 1. For clarity, Algorithm 1 assumes that $T - T_0$ is divisible by $R$. When this is not the case, the remaining steps are added to the initialization stage. Finally, the $\texttt{ODE\_Solver}$ component may be instantiated with any standard solver, such as the Heun method (Karras et al., 2022); one such example is provided in Algorithm 2.

*The Gaussian case.* We now analyze the behavior of CFIG in the Gaussian setting to gain a deeper understanding of its dynamics, with particular focus on the role of the parameter $\sigma_*$. To enable an exact analysis, we consider the simplified Gaussian scenario in Example 1 with $\mathbf{c} = 0$, which admits an explicit solution to the PF-ODE (2.1) when using the CFG denoisers $D_\sigma^{\mathsf{cfg}}(\cdot|\mathbf{c}; w)$. This simplified setting mirrors the one studied in Bradley & Nakkiran (2024). Formally, we consider the following iterative procedure, which corresponds to Algorithm 1 with no discretization error. Let $X_0^{(0)}$ be a square-integrable random variable, meaning we start from the conditional distribution, and define for all $r \geq 1$, $X_0^{(r)} = c(\sigma_*)\big(X_0^{(r-1)} + \sigma_* Z^{(r)}\big)$, where $c(\sigma_*) := \gamma^w(1 + \sigma_*^2)^{(w-1)/2}/(\gamma^2 + (1+\gamma^2)\sigma_*^2)^{w/2}$; see Appendix B.3 and Lemma 2. The following proposition compares the limiting distribution of $X_0^{(r)}$ (as $r \to \infty$) to the tilted distribution $p_0(\cdot|\mathbf{c}; w) := \mathcal{N}(0, V(w))$, where $V(w) := \gamma^2/(\gamma^2 + w)$. Define $V_\infty(w) := \sigma_*^2 c(\sigma_*^2)/(1 - c(\sigma_*^2))$.

**Proposition 4.** *For all $w > 1$, $(X_0^{(r)})_{r \in \mathbb{N}}$ converges to $\mathcal{N}(0, V_\infty(w))$ exponentially fast in Wasserstein-2 distance, with rate proportional to $\sigma_*^2$. Furthermore,*

$$V_\infty(w) = V(w) + O(\sigma_*^2), \quad as \ \sigma_*^2 \to 0.$$

Thus, in this simplified setting, the limiting distribution of our procedure converges to the target distribution as the parameter $\sigma_*$ tends to zero. However, as highlighted in Proposition 4, a bias–variance trade-off emerges when the number $R$ of refinements stages is finite as shown in Figure 10. More specifically, selecting a very small value of $\sigma_*$ may result in slow mixing, since each noising step induces only a minor change in the state $X_{\sigma_*}^{(r)}$. On the other hand, selecting a larger $\sigma_*$ can enhance mixing speed, albeit at the cost of introducing some bias as detailed in Appendix B.3.

## 5 RELATED WORKS

The seminal work of Ho & Salimans (2022) initiated a line of research to understand and improve guidance mechanisms in conditional diffusion models. Below, we review the key contributions.

*Tilted distribution samplers.* Recent works on CFG (Bradley & Nakkiran, 2024; Chidambaram et al., 2024) have pointed out that $D_\sigma^{\mathsf{cfg}}(\cdot|\mathbf{c}; w)$ does not correspond to a valid denoiser for $p_0(\cdot|\mathbf{c}; w)$. To address this shortcoming, Bradley & Nakkiran (2024) introduce a hybrid predictor–corrector approach, in which each update (2.2) is followed by a few Langevin dynamics steps targeting the intermediate distribution $p_\sigma^{\mathsf{cfg}}(\cdot|\mathbf{c}; w)$; see Bradley & Nakkiran (2024, Algorithm 2). The key insight is that, although the family $(p_\sigma^{\mathsf{cfg}}(\cdot|\mathbf{c}; w))_{\sigma \in [0, \sigma_{\max}]}$ are not the marginals of a valid diffusion process, it nonetheless defines an annealing path connecting the initial distribution $p_{\sigma_{\max}}$ and $p_0(\cdot|\mathbf{c}; w)$, and can therefore be used within an MCMC framework. The recent approach of Skreta et al. (2025) proposes using sequential Monte Carlo methods (Doucet et al., 2001) to iteratively construct empirical approximations of the distributions $p_\sigma^{\mathsf{cfg}}(\cdot|\mathbf{c}; w)$ via a set of $N$ weighted particles. A simple derivation of this sampler is provided in Appendix C. Concurrently, Lee et al. (2025) proposed a similar approach for discrete diffusion models. The present work falls within the same class of methods, sharing the goal of generating approximate samples from $p_0(\cdot|\mathbf{c}; w)$. However, in contrast to particle-based approaches, we use an iterative scheme that do not rely on multiple particles to produce samples.

*Adaptive CFG methods.* Other approaches in the literature do not explicitly aim to sample from the tilted distribution. Instead, they employ various heuristics aimed at enhancing sample quality, diversity, or both simultaneously. To enhance sample diversity under guidance, Chang et al. (2023); Sadat et al. (2024) propose a time-dependent guidance scale that prioritizes the unconditional model in the early stages of the diffusion process, gradually transitioning toward standard CFG as sampling proceeds. In Kynkäänniemi et al. (2024), guidance is activated only when $\sigma$ is within a specified noise interval $[\sigma_{\mathrm{lo}}, \sigma_{\mathrm{hi}}]$, and empirical results indicate that this strategy can significantly improve over the vanilla CFG. Xi et al. provide an empirical study of adaptive CFG schedulers and conclude that increasing the guidance scale throughout the iterations improves the performance, aligning with the conclusions of the previous works. Similarly, Chung et al. (2025) formulate text-guidance as an inverse problem and arrive at the dynamic CFG schedule $w_t = \lambda\sigma_t/(\sigma_t - \sigma_{t-1})$, where $\lambda \in [0, 1]$.

*Guidance through different mechanisms.* Alternatives to the CFG denoiser have also been proposed in the literature. Karras et al. (2024a) propose to combine a denoiser $D_\sigma^\theta(\cdot|\mathbf{c})$ with a smaller or undertrained version of itself $D_\sigma^{\theta-}(\cdot|\mathbf{c})$ as $\widetilde{D}_\sigma^\theta(\mathbf{x}_\sigma|\mathbf{c}; w) = wD_\sigma^\theta(\mathbf{x}_\sigma|\mathbf{c}) + (1-w)D_\sigma^{\theta-}(\mathbf{x}_\sigma|\mathbf{c})$. In terms of scores this is equivalent to substracting the score of a more spread-out density and results in a score of a more peaked density, amplifying sharpness and conditioning alignment. The same effect is achieved by substracting the score at a higher noise level, as in Sadat et al. (2025); Li et al. (2024), and with the score approximation resulting from Proposition 3 when $\delta < w - 1$. Moreover, Pavasovic et al. (2025) develops non-linear guidance mechanisms that automatically reduce or switch off guidance when the difference between the conditional and unconditional scores becomes small.

Compared to the above approaches, our method employs a distinct guidance strategy: it begins by generating an initial sample using no or moderate guidance, then repeatedly applies the forward noising process up to a predefined noise level, followed by denoising using CFG. This noising-denoising procedure was originally introduced in SDEdit Meng et al. (2021) for image editing and has also been adopted later by subsequent work. In particular, it has been leveraged by Yu et al. (2023); Bansal et al. (2023) —under the name of *time-travel*— to guide diffusion models with an auxiliary function, and later utilized by Ma et al. (2025) to improve inference-time scaling of DDMs. Our contribution differs by applying this noising-denoising procedure within the framework of classifier-free guidance to maintain both sample diversity and quality.

| | | Quality metrics | | | |
|---|---|---|---|---|---|
| Algorithm | FID ↓ | FD$_{\text{DINOv2}}$ ↓ | Precision ↑ | Recall ↑ | Density ↑ | Coverage ↑ |
| **EDM2-S** | | | | | | |
| CFG | 2.30 | 88.70 | 0.61 | 0.57 | 0.58 | 0.54 |
| INTERVAL-CFG | **1.71** | 80.75 | 0.61 | **0.61** | 0.58 | 0.56 |
| CFG++ | 2.89 | 95.52 | 0.60 | 0.54 | 0.57 | 0.52 |
| FK | 9.17 | 155.36 | 0.54 | 0.39 | 0.44 | 0.39 |
| PG | 5.79 | 87.54 | 0.67 | 0.41 | **0.67** | 0.55 |
| CFIG | 1.78 | **75.38** | **0.64** | 0.59 | 0.63 | **0.58** |
| **EDM2-XXL** | | | | | | |
| CFG | 1.81 | 56.82 | 0.67 | 0.65 | 0.71 | 0.65 |
| INTERVAL-CFG | 1.50 | **40.08** | 0.70 | **0.68** | 0.78 | **0.70** |
| CFG++ | 2.30 | 65.32 | 0.66 | 0.63 | 0.69 | 0.62 |
| FK | 8.86 | 76.29 | 0.75 | 0.38 | **0.87** | 0.49 |
| PG | 5.76 | 44.23 | **0.8** | 0.5 | 0.85 | 0.53 |
| CFIG | **1.48** | 42.87 | 0.70 | **0.68** | 0.77 | **0.70** |

Table 1: Comparison of average FID, FD$_{\text{DINOv2}}$, Precision/Recall, and Density/Coverage on `ImageNet`-512 for EDM2-S and EDM2-XXL.

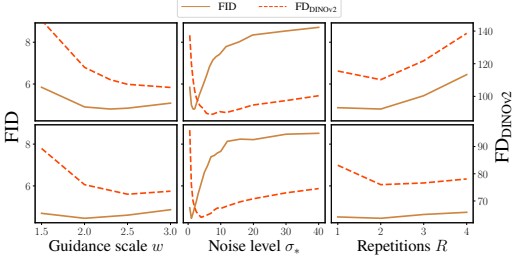

Figure 3: Impact of the hyperparameters for the EDM2-S (top) and EDM2-XXL (bottom) models. The metrics are computed with 10k generated samples.

## 6 EXPERIMENTS

We compare CFIG against five baselines: CFG (Ho & Salimans, 2022), INTERVAL-CFG (Kynkäänniemi et al., 2024), CFG++ (Chung et al., 2025), Feynman-Kac Corrector (FK) (Skreta et al., 2025), and Particle Guidance (PG) (Corso et al., 2023), with the last two belonging to the class of particle-based methods. Unless otherwise specified, we adopt the hyperparameters recommended by the respective authors; when unavailable, and conduct a grid search otherwise. We exclude CADS (Sadat et al., 2024) from our comparisons due to its close similarity to the more recent INTERVAL-CFG, which has demonstrated state-of-the-art performance. In fact, INTERVAL-CFG can be seen as a special case of CADS where the guidance scale is set to zero at high noise levels. We present results on both image and audio generation tasks, considering two computational budget settings: 32 diffusion steps for image generation and 200 for audio. By design, this ensures that CFIG requires no more NFEs than CFG, as shown in Table 7, which reports the exact NFE counts for each method across all experiments. We evaluate performance under two samplers, Heun and DDIM, with the exception of FK, which is an SMC algorithm, see Appendix C.

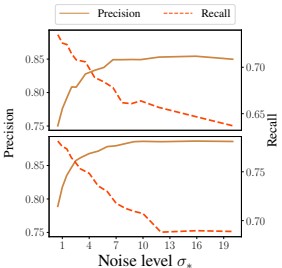

Figure 4: Precision and recall as a function of $\sigma_*$ for EDM2-S (top) and EDM2-XXL (bottom).

We emphasize that **all competitors are run under a fixed number of diffusion steps**. For our method, increasing the number of refinement stages reduces the number of denoising steps allocated per refinement, which may not yield performance gains, as shown in the **Image Experiments ablation** in Figure 3. We additionally analyze the scenario where the compute budget scales linearly with the number of refinement stages $R$ in Figure 7. We provide details about the setups of the used models, the implementation details of the competitors as well as their hyperparameters in Appendix A.2, and, defer the discussion about runtime, NFEs, and memory requirements of each them to Appendix A.4.

**Image experiments.** For *class-conditional generation*, we evaluate the algorithms on the `ImageNet`-512 dataset using EDM2 small (EDM2-S) and largest (EDM2-XXL) models, which operate in the latent space (Karras et al., 2024b). Following the setup of Kynkäänniemi et al. (2024), all algorithms are run with the 2$^{\text{nd}}$ Heun sampler and a fixed budget of 32 deterministic steps (Karras et al., 2022). We assess perceptual quality using FID (Heusel et al., 2017) and FD$_{\text{DINOv2}}$ (Stein et al., 2023), and further analyze fidelity and diversity through Precision/Recall and Density/Coverage (Naeem et al., 2020). Metrics are computed using 50k generated samples follow-

Table 2: CLIP-Score and FID results for SDXL on `MS-COCO` validation data set

| | CLIP-Score ↑ | FID ↓ |
|---|---|---|
| CFG | 31.81 | 49.41 |
| INTERVAL-CFG | 31.75 | 49.91 |
| CFG++ | 31.54 | 47.88 |
| FK | **32.17** | 48.72 |
| PG | 31.82 | 49.70 |
| CFIG (ours) | 31.97 | **47.05** |

ing commun practices and averaged over three independent runs. Results are reported in Table 1, and qualitative examples are provided in Appendix E. For *text-to-image generation*, we conduct experiments on the Microsoft COCO 2017 (`MS-COCO`) validation set (Lin et al., 2014), which contains 5k image-prompt pairs, using the SDXL model (Podell et al., 2023). Evaluation metrics include CLIP-Score (Radford et al., 2021) to measure prompt alignment and FID to assess image fidelity. As in the class-conditional setting, we fix the computational budget to

| | **Quality metrics** | | |
|---|---|---|---|
| Algorithm | FAD↓ | KL↓ | IS↑ |
| CFG | 1.78 | 1.59 | 7.07 |
| INTERVAL-CFG | 1.74 | 1.61 | 6.93 |
| CFG++ | 1.88 | **1.55** | 7.37 |
| FK | 2.22 | 1.67 | 6.84 |
| PG | 2.77 | 1.58 | 7.39 |
| CFIG ($w_0 = 1.0, R = 1$) | 1.71 | 1.65 | 7.05 |
| CFIG ($w_0 = 1.5, R = 1$) | **1.61** | 1.58 | 7.31 |
| CFIG ($w_0 = 1.5, R = 2$) | 1.74 | 1.56 | **7.64** |

Table 3: FAD, KL, and IS results on `AudioCaps` test set for AudioLDM 2-Full-Large model.

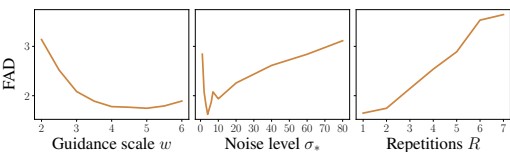

Figure 6: Impact of the hyperparameters on the FAD with the AudioLDM 2-Full-Large model. The metrics are computed with 1k prompts.

32 diffusion steps with the Heun sampler. For each algorithm, hyperparameters are tuned via grid search to maximize CLIP-Score. The corresponding results are summarized in Table 2.

*Results.* In class-conditional generation, CFIG consistently achieves the best or near-best performance across all evaluation metrics. Both CFIG and INTERVAL-CFG outperform CFG and CFG++ on FID/FD$_{\text{DINOv2}}$, indicating better perceptual quality and diversity as measured by Precision/Density and Recall/Coverage. In contrast, FK and PG obtain relatively weak FID/FD$_{\text{DINOv2}}$ scores, primarily due to limited diversity, as evidenced by their lower Recall and Coverage. This phenomenon is attributable to mode-collapse, a well-known limitation of particle-based methods in high-dimensional settings. Importantly, the performance of CFIG also extends to text-to-image generation, where it achieves the best FID and performs on par with FK in terms of CLIP-Score. Taken together, these results demonstrate that CFIG improves visual quality in alignment with conditioning information while maintaining high sample diversity; see Figures 8 and 9.

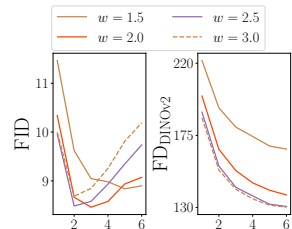

Figure 5: FID as a function of $\sigma_*$ for different guidance scales $w$ on EDM-S model with 5k samples.

**Audio experiments.** We assess CFIG on a text-to-audio task using the `AudioCaps` (Kim et al., 2019) test set. We use the model AudioLDM 2-Full-Large of (Liu et al., 2024) and adopt their experimental setup, namely, 1k prompts were randomly selected from `AudioCaps`, and the following quality metrics were computed for each algorithm: Fréchet Audio Distance (FAD), Kullback–Leibler divergence over softmax outputs (KL), and Inception Score (IS). The KL is computed by applying Softmax to extracted features from the generated and groundtruth samples. Following Liu et al. (2024), for each prompt, the best out of 3 samples were selected. We fix the computatinal budget to 200 diffusion steps with DDIM sampler, we run a grid search over the hyperparameter space of each algorithm to optimize for the FAD. The results are reported in Table 3.

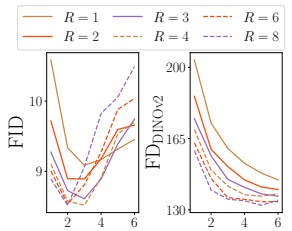

Figure 7: FID as a function of $\sigma_*$ for different number of refinement steps $R$ in the setup of a dynamic computational budget linear w.r.t. $R$. Results are for EDM-S model with 5k samples.

*Results.* The configuration of CFIG with $w_0 = 1.5$ and $R = 2$ achieves the best overall balance across the three evaluated metrics. Interestingly, reducing refinements to $R = 1$ significantly improves the FAD metric. While this scenario closely resembles INTERVAL-CFG, CFIG consistently outperforms it, highlighting fundamental differences between these methods. In all these settings, we set $T_0 = 100$, $\sigma_* = 5$, and $w = 5$. We also observed that omitting initial guidance ($w_0 = 1$) yields better FAD scores compared to CFG, INTERVAL-CFG, and CFG++, but negatively impacts the other metrics. Overall, both text-to-audio and text-to-image models require a moderate degree of guidance, since when unguided they frequently produce samples of poor perceptual quality.

**Ablations.** In Figure 3, we investigate the impact of the hyperparameters $(w, \sigma_*, R)$ **for a fixed number of diffusion steps**. For each model, we vary one parameter at a time while keeping the others fixed to the values used in Table 1. Using $R = 2$ refinement stages improves performance over $R = 1$, yielding the results reported in Table 1. However, increasing $R$ beyond 2 degrades both FID and FD$_{\text{DINOv2}}$, as a higher number of repetitions reduces the number of ODE integration steps per refinement (under a fixed NFE), leading to loss of high-frequency details and attenuation of semantic content. Varying $\sigma_*$ exhibits a different pattern: FID decreases until reaching an optimal point and

CFIG  CFG

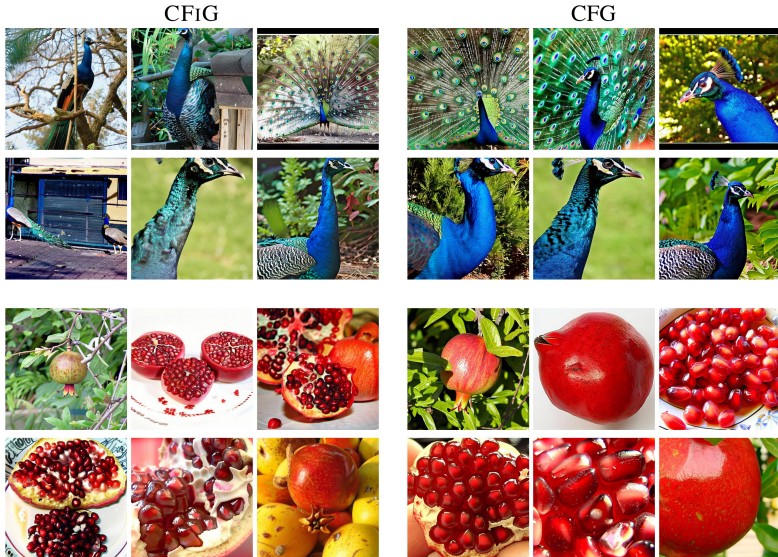

Figure 8: Comparison of sample diversity between CFIG and CFG across a batch of 6 samples generated using EDM-XXL model for two `ImageNet` classes: 84 (top) 957 (bottom). Both algorithms are run with a total of 32 Heun steps and the same seed. CFG is run with a guidance scale $w = 3.5$. CFIG is run with 12 initial steps at guidance scale $w_0 = 1$, followed by two refinement stages starting at $\sigma_* = 1$ and using $w = 3.5$.

then worsens, while $FD_{\text{DINOv2}}$ continues to improve beyond that turning point before eventually showing a slight degradation at higher noise levels. Similar trends are observed for text-to-audio generation, as shown in Figure 6.

*Trade-off diversity/quality.* Figure 4 shows the evolution of Precision and Recall with respect to the noise level $\sigma_*$. As expected, larger $\sigma_*$ reduces diversity—indicated by falling Recall—since subsequent samples $X_0^{(r)}$ ($r \geq 1$) drift away from the initial unguided sample $X_0^{(0)}$, and CFIG effectively collapses to standard CFG. Conversely, Precision increases and correlates with $FD_{\text{DINOv2}}$.

*Different guidance scales.* In Figure 5, we further ablate FID and $FD_{\text{DINOv2}}$ as a function of $\sigma_*$ across different guidance scales $w$. We observe that stronger guidance shifts the optimal value of $\sigma_*$ toward smaller values. This behavior arises because larger guidance amplifies the influence of the Rényi divergence term and thereby increases the bias of CFG when omitting it. Hence, this shrinks the range in which removing the Rényi divergence term remains acceptable.

*Dynamic computational budget as function of R.* We additionally evaluate CFIG under a **dynamic computational budget** in which computation scales linearly with the number of refinement stages $R$ as reported in Figure 7. Increasing refinement stages consistently improves FID and $FD_{\text{DINOv2}}$, but shifts the optimal $\sigma_*$ toward smaller values for FID. In particular, when using more refinement stages, lower noise levels become preferable.

## 7 CONCLUSION

We have provided a detailed analysis of CFG and identified a crucial missing term—the gradient of a Rényi divergence—which naturally promotes diversity in generated samples. Our theoretical analysis demonstrates that this additional term becomes negligible in the low-noise regime, motivating a new sampling algorithm, CFIG, that leverages this insight. Through extensive experiments on both image and audio generation tasks, we have confirmed the effectiveness and practical benefits of the proposed method. The insights gained in this work suggest several promising directions for future research. One especially promising avenue involves utilizing our newly derived score expressions to design novel training procedures for conditional diffusion models to explicitly account for the Rényi divergence term and hence enables guidance without sacrificing diversity or perceptual quality.

**Ethics Statement.** The presented approach enhances sample fidelity and diverity in conditional diffusion models, opening opportunities for use in creative workflows. At the same time, it brings ethical challenges related to the possible misuse in producing synthetic-deceptive media that obscure authenticity, emphasizing the necessity of robust detection mechanisms and careful, responsible deployment.

**Reproducibility statement.** Reproducibility is a key priority in this work. The released codebase includes complete implementations of our approach and all baseline methods, together with experiment scripts and configuration files detailing the hyperparameters and settings used in each evaluation. These materials enable researchers to reproduce all reported results reliably and to build upon our work with minimal effort.

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

CFiG        CFG

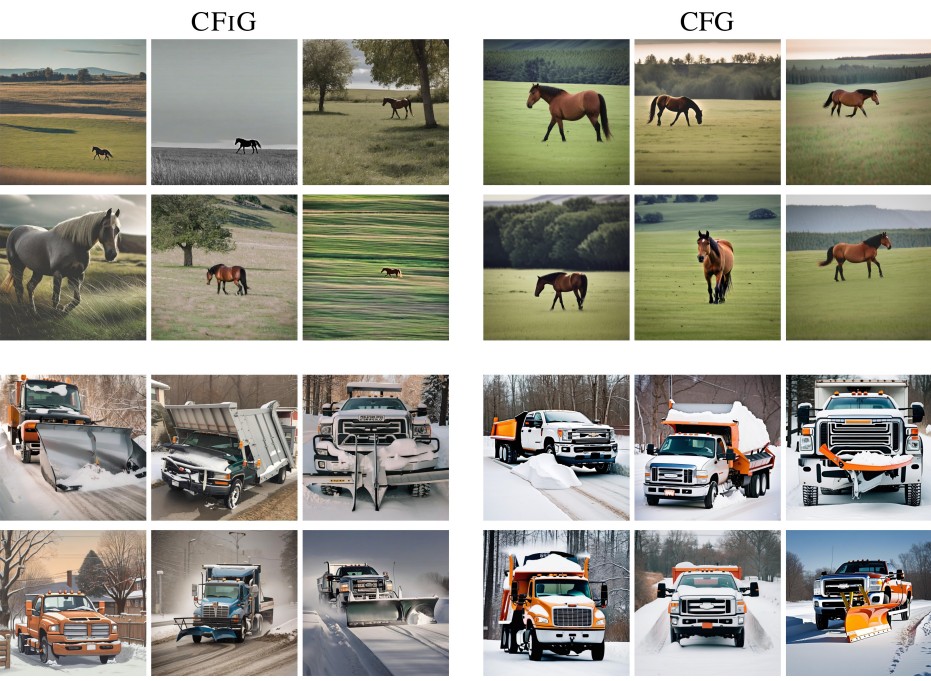

Figure 9: Comparison of sample diversity between CFIG and CFG across a batch of 6 samples. Samples are generated using SDXL model for two for two prompts: *"A lone horse walking through a grassy field."* (top) and *"A truck with a snow plow attached to the front."* (bottom). Both algorithms are run with a total of 32 Heun steps and the same seed. CFG is run with a guidance scale of $w = 16$. CFIG is run with 10 initial steps at guidance scale $w_0 = 1$, followed by two refinement stages starting at $\sigma_* = 2.5$ and using a guidance scale $w = 16$.

## A  EXPERIMENTS DETAILS

### A.1  MODELS SETUPS AND EVALUATION DATASETS

**Class Conditional Image Generation.**   We employ the class conditional latent diffusion models introduced in Karras et al. (2024b) and trained on `ImageNet`-512 dataset. These model operate in a latent space, with dimension $4 \times 64 \times 64$, defined by the pre-trained VAE (Rombach et al., 2022). The experiments are performed using two model sizes: small (EDM-S) and largest (EDM-XXL); see (Karras et al., 2024b, Table 2) for their model sizes. We use the publicly available weights[1], more specifically the models under the pseudos `edm2-img512-s-guid-fid` and `edm2-img512-xxl-guid-fid`. These weights were obtained by tuning the EMA length to optimize FID as detailed in (Karras et al., 2024b, Section 3). In both models, we use $\sigma_0 = 2 \times 10^{-3}$ and $\sigma_{\max} = 80$ for sampling.

*Dataset and evaluation.*  We assess the overall quality of the generated samples by computing the Fréchet distance between the reference and generated samples. We follow the recommendations in Stein et al. (2023) and report, in addition to the FID, the $\text{FD}_{\text{DINOv2}}$; in the former the images representations are computed using InceptionV3 (Szegedy et al., 2016), whereas in the latter they are computed with DINOv2 ViT-L/14 (Oquab et al., 2023). We reuse the precomputed statistics provided in EDM2 repository[1] for the reference distribution, and use, following common practices (Stein et al., 2023), 50k samples to compute the statistics of the generative distribution. We also evaluate individually the fidelity and diversity of the generated samples using Precision/Recall and Density/Coverage (Naeem et al., 2020). We follow the setting in Kynkäänniemi et al. (2024), where the manifold of the data is estimated using 50k samples with representations computed using DINOv2 ViT-L/14 (Oquab et al., 2023) and using 3 neighbors, except that for the reference images, where we use `ImageNet` validation set instead of training set.

---

[1]https://github.com/NVlabs/edm2

**Text-to-Image Generation.** We use Stable Diffusion XL (SDXL) (Podell et al., 2023), a text-to-image latent diffusion model with latent size $4 \times 128 \times 128$ for generating images at resolution $1024 \times 1024$. SDXL relies on pre-trained text encoders (Ilharco et al., 2021; Radford et al., 2021) to extract semantic features from text; see (Podell et al., 2023, Section 2.1) for details. In our experiments, we employ the publicly available pre-trained model from HuggingFace[2]. Since SDXL follows the variance-preserving (VP) formulation, we convert it to the variance-exploding (VE) formulation following Karras et al. (2022). For sampling, we set $\sigma_0 = 2.92 \times 10^{-2}$ and $\sigma_{\max} = 14.61$.

*Dataset and evaluation.* We evaluate on the Microsoft COCO 2017 (MS-COCO) validation dataset (Lin et al., 2014), which contains 5k image-prompt pairs. Prompt alignment is measured using CLIP-Score (Radford et al., 2019) with the clip-vit-base-patch32 model[3] for feature extraction, and image fidelity is assessed with FID.

**Text-to-Audio Generation.** We use the model AudioLDM 2-Full-Large Liu et al. (2024). It is a latent diffusion model tailored for general-purpose audio generation. It operates on compressed representations of mel-spectrograms obtained through a variational autoencoder. The model uses a self-supervised AudioMAE Xu et al. (2022) to extract a semantic embedding known as the Language of Audio (LOA), which captures both acoustic and semantic information. A GPT-2 language model Radford et al. (2019) translates text prompts into LOA features, which then condition the diffusion model for audio generation in a computationally efficient latent space. We use the publicly available pre-trained model at HuggingFace[4] under the pseudo name audioldm2-large. Similar to SDXL, AudioLDM 2-Full-Large follows the VP formulation and hence we converted it to VE formulation following Karras et al. (2022). In the experiments, we use $\sigma_0 = 3.88 \times 10^{-2}$ and $\sigma_{\max} = 83.33$ for sampling.

*Dataset and evaluation.* We sampled 1k prompts randomly from the AudioCaps Kim et al. (2019) test set. Following Liu et al. (2024), each algorithm was run with a negative prompt *"low quality"* along side the conditioning prompt and the best out of 3 samples was selected for each conditioning prompt. The evaluation was performed on 10 seconds audios at 16 kHz. The VGGish model Hershey et al. (2017) was used to extract the features to be used to compute FAD. Similarly, PANNs Kong et al. (2020a) was used to compute the features for the KL and IS metrics.

A.2 IMPLEMENTATION OF ALGORITHMS

Table 4: CFIG hyperparameters for the considered models. The symbol # stands for *"number of"*.

| | Sampler | # Total steps | # Initial steps | Initial guidance scale | # Repetitions | Noise level | Guidance scale |
|---|---|---|---|---|---|---|---|
| EDM-S | Heun with $\begin{cases}\sigma_0 & = 2\times10^{-3} \\ \sigma_{\max} & = 80\end{cases}$ | $T = 32$ | $T_0 = 12$ | $w_0 = 1$ | $R = 2$ | $\sigma_* = 2$ | $w = 2.3$ |
| EDM-XXL | | | $T_0 = 12$ | $w_0 = 1$ | $R = 2$ | $\sigma_* = 1$ | $w = 2$ |
| SDXL | Heun with $\begin{cases}\sigma_0 & = 2.92\times10^{-2} \\ \sigma_{\max} & = 14.61\end{cases}$ | $T = 32$ | $T_0 = 12$ | $w_0 = 2$ | $R = 2$ | $\sigma_* = 3.5$ | $w = 14$ |
| AudioLDM 2-Full-Large | DDIM with $\begin{cases}\sigma_0 & = 3.88\times10^{-2} \\ \sigma_{\max} & = 83.33\end{cases}$ | $T = 200$ | $T_0 = 100$ | $w_0 = 1.5$ | $R = 2$ | $\sigma_* = 5$ | $w = 5$ |

**Our proposed algorithm.** We implement CFIG as described in Algorithm 1. Notably, since at each refinement stage, we simulate the PF-ODE over a newly redefined interval $[0, \sigma_*]$, it is possible to adapt the sequence of noise levels for it. We use the noise-scheduling method proposed in (Karras et al., 2022, Eqn. (269)), *i.e.*, for some $\rho \geq 1$,

$$\sigma_t = \left( \sigma_0^{1/\rho} + \frac{t}{T-1}(\sigma_{\max}^{1/\rho} - \sigma_0^{1/\rho}) \right)^{\rho} .$$

For ImageNet, we use $\rho = 7$ and recompute the discretization steps accordingly for the refinement starting from $\sigma_{\max} = \sigma_*$. For AudioLDM and Stable Diffusion, which follow the VP formulation, we found that selecting uniformly over the diffusion steps and then mapping these diffusion steps to noise levels yields better performance.

---

[2]https://huggingface.co/stabilityai/stable-diffusion-xl-base-1.0
[3]https://huggingface.co/openai/clip-vit-base-patch32
[4]https://huggingface.co/cvssp/audioldm2

**CFG++ (Chung et al., 2025).** For CFG++, we leverage the equivalence between the proposed algorithm and standard CFG with a dynamic guidance scale, as described in Chung et al. (2025, Appendix B). We use CFG denoiser when the dymamic guidance scale is greater than 1 and the plain denoiser otherwise.

**Interval-CFG (Kynkäänniemi et al., 2024).** For INTERVAL-CFG, we implement the procedure as defined in Kynkäänniemi et al. (2024, Equations 5 and 6).

**FK (Skreta et al., 2025).** We build on the insight provided in Appendix C and use it to adapt the authors' algorithm based on the discretization of the SDE Skreta et al. (2025, 1, Proposition 3.1).

**PG (Corso et al., 2023).** We implement the Fixed Potential PG as described in Corso et al. (2023, Equation 4) based on the implementation provided in the released code[5].

### A.3 CHOICE OF THE HYPERPARAMETERS OF THE ALGORITHM

For each algorithm, we perform a grid search over its hyperparameter space and select the configuration that optimizes FID for class conditional image generation, CLIP-Score for text-to-image generation, and FAD for audio generation. The used hyperparameters are summarized in Table 4 for CFIG and in Table 5 for the other algorithms.

Table 5: Competitors' hyperparameters for the considered models. The symbol # stands for *"number of"*. For space contraint, we use *"G"* to denote *"Guidance"*.

| | Sampler | # Total steps | CFG | INTERVAL-CFG | | CFG++ | FK | | PG | |
|---|---|---|---|---|---|---|---|---|---|---|
| | | | $G$ scale | $G$ interval | $G$ scale | $G$ factor | $G$ scale | # particles | $G$ scale | # particles |
| EDM-S | Heun with $\begin{cases}\sigma_0 = 2\times 10^{-3} \\ \sigma_{\max} = 80\end{cases}$ | $T=32$ | $w=1.4$ | $[\sigma_{\text{lo}}, \sigma_{\text{hi}}] = [0.28, 2.9]$ | $w=2.1$ | $\lambda=0.35$ | $w=1.8$ | $K=4$ | $w=2.4$ | $K=4$ |
| EDM-XXL | | | $w=1.2$ | $[\sigma_{\text{lo}}, \sigma_{\text{hi}}] = [0.19, 1.61]$ | $w=2$ | $\lambda=0.35$ | $w=1.2$ | $K=4$ | $w=1.8$ | $K=4$ |
| SDXL | Heun with $\begin{cases}\sigma_0 = 2.92\times 10^{-2} \\ \sigma_{\max} = 14.61\end{cases}$ | $T=12$ | $w=12$ | $[\sigma_{\text{lo}}, \sigma_{\text{hi}}] = [0, 7.5]$ | $w=16$ | $\lambda=0.8$ | $w=10$ | $K=4$ | $w=16$ | $K=4$ |
| AudioLDM 2-Full-Large | DDIM with $\begin{cases}\sigma_0 = 3.88\times 10^{-2} \\ \sigma_{\max} = 83.33\end{cases}$ | $T=200$ | $w=4.5$ | $[\sigma_{\text{lo}}, \sigma_{\text{hi}}] = [0, 8.5]$ | $w=5$ | $\lambda=0.1$ | $w=2.5$ | $K=4$ | $w=5$ | $K=4$ |

### A.4 RUNTIME AND GPU MEMORY REQUIREMENT

Table 6: Runtime in seconds for the considered algorithms.

| | CFG | INTERVAL-CFG | CFG++ | CFIG | FK | PG |
|---|---|---|---|---|---|---|
| EDM-S (500 samples) | 907 | 638 | 811 | 760 | 1758 | 3590 |
| EDM-XXL (500 samples) | 1623 | 1334 | 1480 | 1445 | 3170 | 6410 |
| SDXL (8 prompts) | 22.3 | 21.16 | 22.33 | 21.59 | 46.1 | 614 |

All experiments were conducted on an NVIDIA H100 SXM5 80GB. The reported runtimes in Table 6 were measured as follows: for EDM2-S and EDM2-XXL, we averaged a single run that generated 50k images with a batch size of 500, while for SDXL we averaged a single run that generated 5k samples with a batch size of 8.

Particle-based algorithms complicate direct timing comparisons because each sample is obtained by evolving multiple particle trajectories—four particles in our experiments. This imposes additional memory constraints, preventing us from using the same batch size as the other methods. For a relevant comparison, we therefore report runtimes scaled linearly to the equivalent batch size of the other methods. Notably, in Table 6, particle-based methods are separated from the others with a vertical line, and we discuss their performance separately below.

---

[5] https://github.com/gcorso/particle-guidance

Table 7: Number of Function Evaluations (NFEs) split as CFG and plain denoiser evaluations. Note that one CFG denoiser NFE accounts for 2 plain denoiser NFEs.

| | | CFG | INTERVAL-CFG | CFG++ | CFIG (ours) | FK | PG |
|---|---|---|---|---|---|---|---|
| EDM-S | CFG denoiser | 63 | 14 | 45 | 38 | 31 | 63 |
| | Plain denoiser | 0 | 49 | 18 | 23 | 0 | 0 |
| | Total plain denoiser NFEs | 126 | 77 | 108 | 99 | 62 | 126 |
| EDM-XXL | CFG denoiser | 63 | 12 | 39 | 38 | 31 | 63 |
| | Plain denoiser | 0 | 51 | 24 | 23 | 0 | 0 |
| | Total plain denoiser NFEs | 126 | 75 | 102 | 99 | 62 | 126 |
| SDXL | CFG denoiser | 63 | 56 | 63 | 61 | 31 | 63 |
| | Plain denoiser | 0 | 7 | 0 | 0 | 0 | 0 |
| | Total plain denoiser NFEs | 126 | 119 | 126 | 122 | 62 | 126 |
| AudioLDM 2-Full-Large | CFG denoiser | 200 | 142 | 194 | 200 | 199 | 200 |
| | Plain denoiser | 0 | 58 | 6 | 0 | 0 | 0 |
| | Total plain denoiser NFEs | 400 | 342 | 394 | 400 | 398 | 400 |

**Comments on runtime and memory.** Overall, CFIG achieves a runtime close to INTERVAL-CFG, and is faster than CFG and CFG++ for class-conditional image generation, namely EDM-S and EDM-XXL models. In contrast, the runtime of CFIG relative to CFG and CFG++ in text-to-image and audio experiments is similar, since CFIG relies on a CFG denoiser with a moderate guidance scale to generate the initial samples.

This runtime improvement in class-conditional image generation can be attributed to the reduced number of NFEs, as detailed in Table 7: standard classifier-free guidance requires two evaluations per denoising step—one conditional and one unconditional. By contrast, CFIG first generates an initial sample using the plain denoiser—hence one single plain call—and then applies iterative refinement in few diffusion steps in which the CFG denoiser is invoked. Similarly, INTERVAL-CFG applies the CFG denoiser only within a restricted interval of noise levels. This reduction in NFE accounts for the lower runtime of CFIG and INTERVAL-CFG compared to methods—CFG and CFG++—that apply CFG throughout the full sampling trajectory.

In terms of memory consumption, CFG, CFG++, INTERVAL-CFG, and CFIG are the same. Particle-based methods, however, require additional memory that scales with the number of particles, since they must be processed concurrently.

Regarding runtime, on EDM models, PG is approximately $4\times$ slower than non-particle baselines due to its use of four particles. FK runs at roughly half the runtime of PG, as it rely on an Euler discretization of the SDE without the second-order Heun correction used in PG. On SDXL, FK exhibits a similar trend. Finally, we note that the PG implementation experienced overflow when using `float16`, which forced us to fall back to `float32` and hence further increased runtime.

Table 8: Comparison with respect to CFIG when the initial stage is performed with INTERVAL-CFG. The EDM2-S model was considered. Quality metrics are computed for 50k samples. Runtime is reported for generating a batch of 500 samples.

| | FID ↓ | $FD_{DINOv2}$ ↓ | Precision ↑ | Recall ↑ | Density ↑ | Coverage ↑ | Runtime (seconds) |
|---|---|---|---|---|---|---|---|
| INTERVAL-CFG | 1.71 | 80.75 | 0.61 | 0.61 | 0.58 | 0.56 | 638 |
| CFIG with INTERVAL-CFG | 1.82 | 68.56 | 0.65 | 0.58 | 0.65 | 0.60 | 785 |
| CFIG | 1.78 | 75.38 | 0.64 | 0.59 | 0.63 | 0.58 | 760 |

## B  Proofs

### B.1  Score expressions

In this section, we adopt measure-theoretic notations. Denote by $\mathcal{P}(\mathbb{R}^d)$ the set of probability distribution on $\mathbb{R}^d$ endowed with its Borel $\sigma$-algebra. We first restate the definition of the $w$-Rényi divergence between two arbitrary probability distributions $\mu$ and $\nu$ in $\mathcal{P}(\mathbb{R}^d)$:

$$\mathsf{R}_w(\mu\|\nu) := \begin{cases} \frac{1}{w-1} \log \int \left[\frac{\mathrm{d}\mu}{\mathrm{d}\nu}\right]^w \mathrm{d}\nu & \text{if } \mu \ll \nu, \\ \infty & \text{otherwise.} \end{cases} \tag{B.1}$$

Let $p_0 \in \mathcal{P}(\mathbb{R}^d)$ and denote, for any $\mathbf{c} \in \mathcal{C}$, $\mathscr{Z}(\mathbf{c}) := \int g_0(\mathbf{c}|\tilde{\mathbf{x}}_0) p_0(\mathrm{d}\tilde{\mathbf{x}}_0)$. We impose the following assumptions on the likelihood function $\mathbf{x}_0 \mapsto g_0(\mathbf{c}|\mathbf{x}_0)$ in order to make $p_0(\mathrm{d}\mathbf{x}_0|\mathbf{c}) = g_0(\mathbf{c}|\mathbf{x}_0) p_0(\mathrm{d}\mathbf{x}_0) / \mathscr{Z}(\mathbf{c})$ a well-defined probability distribution.

(A1) For every $\mathbf{x} \in \mathbb{R}^d$ and $\mathbf{c} \in \mathcal{C}$, $g_0(\mathbf{c}|\mathbf{x}_0) \geq 0$ and $0 < \mathscr{Z}(\mathbf{c}) < \infty$.

Moreover, we recall the definitions

$$p_{0|\sigma}(\mathrm{d}\mathbf{x}_0|\mathbf{x}_\sigma) := \frac{p_0(\mathrm{d}\mathbf{x}_0) q_{\sigma|0}(\mathbf{x}_\sigma|\mathbf{x}_0)}{p_\sigma(\mathbf{x}_\sigma)} , \tag{B.2}$$

$$p_{0|\sigma}(\mathrm{d}\mathbf{x}_0|\mathbf{x}_\sigma, \mathbf{c}) := \frac{p_0(\mathrm{d}\mathbf{x}_0|\mathbf{c}) q_{\sigma|0}(\mathbf{x}_\sigma|\mathbf{x}_0)}{p_\sigma(\mathbf{x}_\sigma|\mathbf{c})} . \tag{B.3}$$

**Proposition** (Restatement of Proposition 1). *Assume (A1). Then for every $\sigma > 0$, $\mathbf{x}_\sigma \in \mathbb{R}^d$, and $\mathbf{c} \in \mathcal{C}$,*

$$\nabla_{\mathbf{x}_\sigma} \log p_\sigma(\mathbf{x}_\sigma|\mathbf{c}; w) = (w-1)\nabla_{\mathbf{x}_\sigma} \mathsf{R}_w(p_{0|\sigma}(\cdot|\mathbf{x}_\sigma, \mathbf{c})\|p_{0|\sigma}(\cdot|\mathbf{x}_\sigma)) + \nabla_{\mathbf{x}_\sigma} \log p_\sigma^{\mathsf{cfg}}(\mathbf{x}_\sigma|\mathbf{c}; w) .$$

*Proof of Proposition 1.* Combining definitions (B.2) and (B.3) yields

$$p_{0|\sigma}(\mathrm{d}\mathbf{x}_0|\mathbf{x}_\sigma, \mathbf{c}) = \frac{1}{\mathscr{Z}(\mathbf{c})} \frac{g_0(\mathbf{c}|\mathbf{x}_0) p_\sigma(\mathbf{x}_\sigma)}{p_\sigma(\mathbf{x}_\sigma|\mathbf{c})} p_{0|\sigma}(\mathrm{d}\mathbf{x}_0|\mathbf{x}_\sigma) ,$$

from which it follows that $p_{0|\sigma}(\cdot|\mathbf{x}_\sigma, \mathbf{c}) \ll p_{0|\sigma}$ and $\frac{\mathrm{d}p_{0|\sigma}(\cdot|\mathbf{x}_\sigma, \mathbf{c})}{\mathrm{d}p_{0|\sigma}(\cdot|\mathbf{x}_\sigma)}(\mathbf{x}_0) = \frac{g_0(\mathbf{c}|\mathbf{x}_0) p_\sigma(\mathbf{x}_\sigma)}{\mathscr{Z}(\mathbf{c}) p_\sigma(\mathbf{x}_\sigma|\mathbf{c})}$. Hence,

$$\int \left[\frac{\mathrm{d}p_{0|\sigma}(\cdot|\mathbf{x}_\sigma, \mathbf{c})}{\mathrm{d}p_{0|\sigma}(\cdot|\mathbf{x}_\sigma)}\right]^w (\mathbf{x}_0) \, p_{0|\sigma}(\mathrm{d}\mathbf{x}_0|\mathbf{x}_\sigma)$$

$$= \frac{1}{\mathscr{Z}(\mathbf{c})^w} \frac{p_\sigma(\mathbf{x}_\sigma)^w}{p_\sigma(\mathbf{x}_\sigma|\mathbf{c})^w} \int g_0(\mathbf{x}_0|\mathbf{c})^w \, p_{0|\sigma}(\mathrm{d}\mathbf{x}_0|\mathbf{x}_\sigma)$$

$$= \frac{1}{\mathscr{Z}(\mathbf{c})^w} \frac{p_\sigma(\mathbf{x}_\sigma)^{w-1}}{p_\sigma(\mathbf{x}_\sigma|\mathbf{c})^w} \int g_0(\mathbf{x}_0|\mathbf{c})^w q_{\sigma|0}(\mathbf{x}_\sigma|\mathbf{x}_0) \, p_0(\mathrm{d}\mathbf{x}_0)$$

$$= \frac{\int g_0(\mathbf{c}|\tilde{\mathbf{x}}_0)^w \, p_0(\mathrm{d}\tilde{\mathbf{x}}_0)}{\mathscr{Z}(\mathbf{c})^w} \frac{p_\sigma(\mathbf{x}_\sigma)^{w-1}}{p_\sigma(\mathbf{x}_\sigma|\mathbf{c})^w} p_\sigma(\mathbf{x}_\sigma|\mathbf{c}; w) ,$$

where we used, in the last step, the definition

$$p_\sigma(\mathbf{x}_\sigma|\mathbf{c}; w) := \int q_{\sigma|0}(\mathbf{x}_\sigma|\mathbf{x}_0) \frac{g_0(\mathbf{c}|\mathbf{x}_0)^w \, p_0(\mathrm{d}\mathbf{x}_0)}{\int g_0(\mathbf{c}|\tilde{\mathbf{x}}_0)^w \, p_0(\mathrm{d}\tilde{\mathbf{x}}_0)} .$$

Taking the logarithm of both sides of the identity and then differentiating with respect to $\mathbf{x}_\sigma$ yields

$$(w-1)\nabla_{\mathbf{x}_\sigma} \mathsf{R}_w(p_{0|\sigma}(\cdot|\mathbf{x}_\sigma, \mathbf{c})\|p_{0|\sigma}(\cdot|\mathbf{x}_\sigma))$$

$$= \nabla_{\mathbf{x}_\sigma} \log p_\sigma(\mathbf{x}_\sigma|\mathbf{c}; w) - w\nabla_{\mathbf{x}_\sigma} \log p_\sigma(\mathbf{x}_\sigma|\mathbf{c}) + (w-1)\nabla_{\mathbf{x}_\sigma} \log p_\sigma(\mathbf{x}_\sigma) ,$$

which is the desired result upon rearranging. $\square$

Similarly, we establish the following generalization of the identity in Proposition 1, involving the scores at two different noise levels.

**Proposition** (Restatement of Proposition 3). *Assume (**A1**) and let $w > 1$. Then for every $\delta > 0$, $\sigma > 0$, $\mathbf{x}_\sigma \in \mathbb{R}^d$, and $\mathbf{c} \in \mathcal{C}$,*

$$\nabla_{\mathbf{x}_\sigma} \log p_\sigma(\mathbf{x}_\sigma | \mathbf{c}; w) = (w - 1) \nabla_{\mathbf{x}_\sigma} \mathsf{R}_w(p_{0|\sigma_-}(\cdot | \mathbf{x}_\sigma, \mathbf{c}) \| p_{0|\sigma_+}(\cdot | \mathbf{x}_\sigma))$$
$$+ w \nabla_{\mathbf{x}_\sigma} \log p_{\sigma_-}(\mathbf{x}_\sigma | \mathbf{c}) + (1 - w) \nabla_{\mathbf{x}_\sigma} \log p_{\sigma_+}(\mathbf{x}_\sigma) ,$$

*where $\sigma_- := \sigma\sqrt{w/(1 + \delta)}$ and $\sigma_+ := \sigma\sqrt{(w - 1)/\delta}$.*

*Proof of Proposition 3.* Since $w > 1$ and $\delta > 0$, the two noise levels $\sigma_+$ and $\sigma_-$ are well defined and by combining definitions (B.2) and (B.3) we obtain that

$$p_{0|\sigma_-}(\mathrm{d}\mathbf{x}_0 | \mathbf{x}_\sigma, \mathbf{c}) = \frac{1}{\mathscr{Z}(\mathbf{c})} \frac{g_0(\mathbf{c} | \mathbf{x}_0) p_{\sigma_+}(\mathbf{x}_\sigma) q_{\sigma_-|0}(\mathbf{x}_\sigma | \mathbf{x}_0)}{p_{\sigma_-}(\mathbf{x}_\sigma | \mathbf{c}) q_{\sigma_+|0}(\mathbf{x}_\sigma | \mathbf{x}_0)} p_{0|\sigma_+}(\mathrm{d}\mathbf{x}_0 | \mathbf{x}_\sigma) .$$

Thus,

$$\int \left[ \frac{\mathrm{d}p_{0|\sigma_-}(\cdot | \mathbf{x}_\sigma, \mathbf{c})}{\mathrm{d}p_{0|\sigma_+}(\cdot | \mathbf{x}_\sigma)} \right]^w (\mathbf{x}_0) \, p_{0|\sigma_+}(\mathrm{d}\mathbf{x}_0 | \mathbf{x}_\sigma)$$
$$= \frac{1}{\mathscr{Z}(\mathbf{c})^w} \frac{p_{\sigma_+}(\mathbf{x}_\sigma)^w}{p_{\sigma_-}(\mathbf{x}_\sigma | \mathbf{c})^w} \int \left[ \frac{q_{\sigma_-|0}(\mathbf{x}_\sigma | \mathbf{x}_0)}{q_{\sigma_+|0}(\mathbf{x}_\sigma | \mathbf{x}_0)} \right]^w g_0(\mathbf{c} | \mathbf{x}_0)^w p_{0|\sigma_+}(\mathrm{d}\mathbf{x}_0 | \mathbf{x}_\sigma)$$
$$= \frac{1}{\mathscr{Z}(\mathbf{c})^w} \frac{p_{\sigma_+}(\mathbf{x}_\sigma)^{w-1}}{p_{\sigma_-}(\mathbf{x}_\sigma | \mathbf{c})^w} \left[ \int \frac{q_{\sigma_-|0}(\mathbf{x}_\sigma | \mathbf{x}_0)^w}{q_{\sigma_+|0}(\mathbf{x}_\sigma | \mathbf{x}_0)^{w-1}} g_0(\mathbf{c} | \mathbf{x}_0)^w p_0(\mathrm{d}\mathbf{x}_0) \right] .$$

Here, by the definition of $\sigma_-$,

$$q_{\sigma_-|0}(\mathbf{x}_\sigma | \mathbf{x}_0)^w = \mathrm{N}\big(\mathbf{x}_\sigma; \mathbf{x}_0, \sigma^2 w/(\delta + 1)\mathbf{I}\big)^w$$
$$\propto \mathrm{N}\big(\mathbf{x}_\sigma; \mathbf{x}_0, \sigma^2 \mathbf{I}\big)^{(\delta+1)}$$
$$\propto q_{\sigma|0}(\mathbf{x}_\sigma | \mathbf{x}_0)^{\delta+1} ,$$

where the constant of proportionality is independent of $(\mathbf{x}_0, \mathbf{x}_\sigma)$. Similarly, $q_{\sigma_+|0}(\mathbf{x}_\sigma | \mathbf{x}_0)^{w-1} \propto q_{\sigma|0}(\mathbf{x}_\sigma | \mathbf{x}_0)^\delta$. Hence, $q_{\sigma_-|0}(\mathbf{x}_\sigma | \mathbf{x}_0)^w / q_{\sigma_+|0}(\mathbf{x}_\sigma | \mathbf{x}_0)^{w-1} \propto q_{\sigma|0}(\mathbf{x}_\sigma | \mathbf{x}_0)$, which implies that

$$\int \left[ \frac{\mathrm{d}p_{0|\sigma_-}(\cdot | \mathbf{x}_\sigma, \mathbf{c})}{\mathrm{d}p_{0|\sigma_+}(\cdot | \mathbf{x}_\sigma)} \right]^w (\mathbf{x}_0) \, p_{0|\sigma_+}(\mathrm{d}\mathbf{x}_0 | \mathbf{x}_\sigma)$$
$$\propto \frac{1}{\mathscr{Z}(\mathbf{c})^w} \frac{p_{\sigma_+}(\mathbf{x}_\sigma)^{w-1}}{p_{\sigma_-}(\mathbf{x}_\sigma | \mathbf{c})^w} \int q_{\sigma|0}(\mathbf{x}_\sigma | \mathbf{x}_0) g_0(\mathbf{c} | \mathbf{x}_0)^w p_0(\mathrm{d}\mathbf{x}_0)$$
$$\propto \frac{\int g_0(\mathbf{c} | \tilde{\mathbf{x}}_0)^w p_0(\mathrm{d}\tilde{\mathbf{x}}_0)}{\mathscr{Z}(\mathbf{c})^w} \frac{p_{\sigma_+}(\mathbf{x}_\sigma)^{w-1}}{p_{\sigma_-}(\mathbf{x}_\sigma | \mathbf{c})^w} p_\sigma(\mathbf{x}_\sigma | \mathbf{c}; w) .$$

Finally, taking the logarithm of both sides and then the gradient with respect to $\mathbf{x}_\sigma$ yields

$$(w - 1) \nabla_{\mathbf{x}_\sigma} \mathsf{R}_w(p_{0|\sigma_-}(\cdot | \mathbf{x}_\sigma, \mathbf{c}) \| p_{0|\sigma_+}(\cdot | \mathbf{x}_\sigma))$$
$$= \nabla_{\mathbf{x}_\sigma} \log p_\sigma(\mathbf{x}_\sigma | \mathbf{c}; w) - w \nabla_{\mathbf{x}_\sigma} \log p_{\sigma_-}(\mathbf{x}_\sigma | \mathbf{c}) + (w - 1) \nabla_{\mathbf{x}_\sigma} \log p_{\sigma_+}(\mathbf{x}_\sigma) ,$$

which is the desired expression upon rearranging. $\square$

A direct consequence of Proposition 3 is that the score of the unconditional distribution can be expressed in terms of the scores at two different noise levels.

**Corollary 1.** *Let $p_0 \in \mathcal{P}(\mathbb{R}^d)$ and $w > 1$. Then for every $\delta > 0$, $\sigma > 0$, and $\mathbf{x}_\sigma \in \mathbb{R}^d$,*

$$\nabla \log p_\sigma(\mathbf{x}_\sigma) = (w - 1) \nabla \mathsf{R}_w(p_{0|\sigma_-}(\cdot | \mathbf{x}_\sigma) \| p_{0|\sigma_+}(\cdot | \mathbf{x}_\sigma))$$
$$+ w \nabla \log p_{\sigma_-}(\mathbf{x}_\sigma) + (1 - w) \nabla \log p_{\sigma_+}(\mathbf{x}_\sigma) ,$$

*where $\sigma_- := \sigma\sqrt{w/(1 + \delta)}$ and $\sigma_+ := \sigma\sqrt{(w - 1)/\delta}$.*

*Proof.* The result follows by applying Proposition 3 with $g_0$ being the constant mapping $\mathbf{x} \mapsto 1$. ☐

## B.2 Score approximations

Throughout this section, we will often assume that the relevant densities belong to the following class.

**Definition 1.** *For $m \in \mathbb{N}^*$, denote by $\mathcal{Q}_3(\mathbb{R}^m)$ the set of functions $f_0 : \mathbb{R}^m \to \mathbb{R}$ satisfying*

(i) *$f_0$ is strictly positive,*

(ii) *$f_0$ is three times continuously differentiable with third-order partial derivatives of at most polynomial growth, i.e., there exist $C \geq 0$ and $m \geq 0$ such that for any $(i, j, k) \in [\![1, d]\!]^3$, $|\partial_i \partial_j \partial_k f_0(\mathbf{x})| \leq C(1 + \|\mathbf{x}\|_2)^m$.*

The proof of Proposition 1 relies on the following general result, which is proven subsequently. For any $\sigma > 0$ and $\mathbf{x} \in \mathbb{R}^d$, define

$$f_\sigma(\mathbf{x}) := \int f_0(\mathbf{x}_0) q_{\sigma|0}(\mathbf{x}|\mathbf{x}_0) \, \mathrm{d}\mathbf{x}_0 .$$

**Lemma 1.** *Let $f_0 \in \mathcal{Q}_3(\mathbb{R}^d)$. Then for every $\mathbf{x} \in \mathbb{R}^d$,*

$$\nabla \log f_\sigma(\mathbf{x}) = \nabla \log f_0(\mathbf{x}) + O(\sigma^2) \quad as \ \sigma \to 0 .$$

**Proposition** (Restatement of Proposition 2)**.** *Assume that $p_0$, $g_0(\mathbf{c}|\cdot)$ and $g_0(\mathbf{c}|\cdot)^w$ belong to $\mathcal{Q}_3(\mathbb{R}^d)$ for some $\mathbf{c} \in \mathcal{C}$ and $w > 1$. Then for any fixed $\mathbf{x} \in \mathbb{R}^d$,*

$$\nabla \mathsf{R}_w(p_{0|\sigma}(\cdot|\mathbf{x}) \| p_{0|\sigma}(\cdot|\mathbf{x})) = O(\sigma^2) \quad as \ \sigma \to 0 .$$

*Proof.* Since the third derivatives of $p_0$ and $g_0(\mathbf{c}|\cdot)$ satisfy the polynomial growth assumption (ii) in Definition 1, so does $p_0(\cdot|\mathbf{c})$. Thus, by Lemma 1 it holds that

$$w\nabla \log p_\sigma(\mathbf{x}|\mathbf{c}) + (1-w)\nabla \log p_\sigma(\mathbf{x}) = w\nabla \log p_0(\mathbf{x}|\mathbf{c}) + (1-w)\nabla \log p_0(\mathbf{x}) + O(\sigma^2)$$

as $\sigma \to 0$. Recall that we assume that $g_0(\mathbf{c}|\mathbf{x})^w$ and $p_0$ belong to $\mathcal{Q}_3(\mathbb{R}^d)$. Consequently, $p_0(\cdot|\mathbf{c}; w)$ satisfies the same condition. Thus, since $p_0(\mathbf{x}|\mathbf{c}; w) \propto p_0(\mathbf{x}|\mathbf{c})^w p_0(\mathbf{x})^{1-w}$, we have that

$$\nabla \log p_\sigma(\mathbf{x}|\mathbf{c}; w) = w\nabla \log p_0(\mathbf{x}|\mathbf{c}) + (1-w)\nabla \log p_0(\mathbf{x}) + O(\sigma^2)$$

as $\sigma \to 0$. Finally, by Proposition 1, as $\sigma \to 0$,

$$(w-1)\nabla \mathsf{R}_w(p_{0|\sigma}(\cdot|\mathbf{x}) \| p_{0|\sigma}(\cdot|\mathbf{x})) = \nabla \log p_\sigma(\mathbf{x}|\mathbf{c}; w) - \nabla \log p_\sigma^{\mathsf{cfg}}(\mathbf{x}_\sigma|\mathbf{c}; w)$$
$$= O(\sigma^2) .$$

☐

*Proof of Lemma 1.* For every $\mathbf{x} \in \mathbb{R}^d$,

$$\nabla \log f_\sigma(\mathbf{x}) = \frac{\int \nabla_\mathbf{x} q_{\sigma|0}(\mathbf{x}|\mathbf{x}_0) f_0(\mathbf{x}_0) \, \mathrm{d}\mathbf{x}_0}{\int f_0(\tilde{\mathbf{x}}_0) q_{\sigma|0}(\mathbf{x}|\tilde{\mathbf{x}}_0) \, \mathrm{d}\tilde{\mathbf{x}}_0}$$

$$= \frac{\int \nabla_\mathbf{x} \log q_{\sigma|0}(\mathbf{x}|\mathbf{x}_0) f_0(\mathbf{x}_0) q_{\sigma|0}(\mathbf{x}|\mathbf{x}_0) \, \mathrm{d}\mathbf{x}_0}{\int f_0(\tilde{\mathbf{x}}_0) q_{\sigma|0}(\mathbf{x}|\tilde{\mathbf{x}}_0) \, \mathrm{d}\tilde{\mathbf{x}}_0} . \tag{B.4}$$

Since $q_{\sigma|0}(\mathbf{x}|\mathbf{x}_0) = \mathrm{N}(\mathbf{x}; \mathbf{x}_0, \sigma^2\mathbf{I})$ and $\nabla \log q_{\sigma|0}(\mathbf{x}|\mathbf{x}_0) = (\mathbf{x}_0 - \mathbf{x})/\sigma^2$, making the change of variable $\mathbf{z} = \mathbf{x}_0 - \mathbf{x}$ yields the expression

$$\nabla \log f_\sigma(\mathbf{x}) = \frac{1}{\sigma^2} \frac{\int \mathbf{z} \, f_0(\mathbf{z} + \mathbf{x}) \varphi_d(\mathbf{z}; \sigma) \, \mathrm{d}\mathbf{z}}{\int f_0(\tilde{\mathbf{z}} + \mathbf{x}) \varphi_d(\tilde{\mathbf{z}}; \sigma) \, \mathrm{d}\tilde{\mathbf{z}}} , \tag{B.5}$$

where $\varphi_d(\cdot; \sigma)$ denotes the zero-mean $d$-dimensional Gaussian density with covariance $\sigma^2 \mathbf{I}$. Define $N(\mathbf{x}) := \int \mathbf{z} f_0(\mathbf{z} + \mathbf{x}) \varphi_d(\mathbf{z}; \sigma) d\mathbf{z}$ and $D(\mathbf{x}) := \int f_0(\mathbf{z} + \mathbf{x}) \varphi_d(\mathbf{z}; \sigma) d\mathbf{z}$, so that $\nabla \log f_\sigma(\mathbf{x}) = \frac{1}{\sigma^2} N(\mathbf{x})/D(\mathbf{x})$.

We start with the numerator $N(\mathbf{x})$ and expand $f_0$ around $\mathbf{x}$ using the Taylor's inequality and using the assumption that $f_0 \in \mathcal{Q}_3(\mathbb{R}^d)$, we get that there exist $C \geq 0$ and $m \in \mathbb{N}$ such that for any $\mathbf{z} \in \mathbb{R}^d$,

$$\left| f_\sigma(\mathbf{z} + \mathbf{x}) - \sigma^2 f_0(\mathbf{x}) - \mathbf{z}^\mathsf{T} \nabla f_0(\mathbf{x}) - \frac{1}{2} \mathbf{z}^\mathsf{T} \nabla^2 f_0(\mathbf{x}) \mathbf{z} \right| \leq C(1 + \|\mathbf{x}\|^m + \|\mathbf{z}\|^m) \|z\|^3 .$$

Hence,

$$\left| N(\mathbf{x}) - \int \left[ f_0(\mathbf{x})\mathbf{z} + (\mathbf{z}^\mathsf{T} \nabla f_0(\mathbf{x}))\mathbf{z} + \left( \frac{1}{2} \mathbf{z}^\mathsf{T} \nabla^2 f_0(\mathbf{x}) \mathbf{z} \right) \mathbf{z} \right] \varphi(\mathbf{z}; \sigma) \, d\mathbf{z} \right|$$

$$\leq C \int (1 + \|\mathbf{x}\|^m + \|\mathbf{z}\|^m) \|\mathbf{z}\|^4 \, \varphi(\mathbf{z}; \sigma) \, d\mathbf{z} . \tag{B.6}$$

Since $\int \mathbf{z} \varphi_d(\mathbf{z}; \sigma) \, d\mathbf{z} = 0$ and $\int \mathbf{z}_i \mathbf{z}_j \varphi_d(\mathbf{z}; \sigma) \, d\mathbf{z} = \sigma^2 \delta_{ij}$, $\int \|z\|^4 \mathbf{z} \varphi(\mathbf{z}; \sigma) \, d\mathbf{z} = O(\sigma^4)$, we get $N(\mathbf{x}) = \sigma^2 \nabla f(\mathbf{x}) + O(\sigma^4)$.

Similarly, we obtain $D(\mathbf{x}) = f_0(\mathbf{x}) + O(\sigma^2)$.

Combining the expansions for the numerator and denominator, we obtain, as $f_0$ is strictly positive (Definition 1(i)),

$$\nabla \log f_\sigma(\mathbf{x}) = \frac{1}{\sigma^2} \frac{N(\mathbf{x})}{D(\mathbf{x})}$$

$$= \frac{\nabla f_0(\mathbf{x}) + O(\sigma^2)}{f_0(\mathbf{x}) + O(\sigma^2)}$$

$$= \frac{1}{f_0(\mathbf{x})} (\nabla f_0(\mathbf{x}) + O(\sigma^2))(1 + O(\sigma^2))$$

$$= \nabla \log f_0(\mathbf{x}) + O(\sigma^2) .$$

$\square$

### B.3 CLASSIFIER-FREE ITERATIVE GUIDANCE

**Uniqueness of the stationary distribution in the ideal case.** Recall that the Markov chain under consideration is given by $X_0^{(r+1)} = F_{0|\sigma_*}\big( X_0^{(r)} + \sigma_* Z^{(r+1)} | \mathbf{c}; w \big)$, where $(Z^{(r)})_{r \in \mathbb{N}}$ is a sequence of i.i.d. standard Gaussian random variables and $F_{0|\sigma_*}(\mathbf{x}_{\sigma_*} | \mathbf{c}; w)$ is the solution to the PF-ODE (2.1) with denoiser $D_\sigma(\cdot | \mathbf{c}; w)$ and initial condition $\mathbf{x}_{\sigma_*}$. By definition of this denoiser, it is associated with $p_0(\cdot | \mathbf{c}; w)$, *i.e.*, if $X_{\sigma_*} \sim p_{\sigma_*}(\cdot | \mathbf{c}; w)$, then $F_{0|\sigma_*}(X_{\sigma_*} | \mathbf{c}, w) \sim p_0(\cdot | \mathbf{c}; w)$; see Section 3.

Moreover, by Step 1), if $X_0^{(r)} \sim p_0(\cdot | \mathbf{c}; w)$, then $X_{\sigma_*}^{(r)} \sim p_{\sigma_*}(\cdot | \mathbf{c}; w)$. Hence $X_0^{(r+1)} = F_{0|\sigma_*}(X_{\sigma_*}^{(r)} | \mathbf{c}; w) \sim p_0(\cdot | \mathbf{c}; w)$, showing that $p_0(\cdot | \mathbf{c}; w)$ is indeed a stationary distribution. In order to show that this invariant distribution is unique, we now establish that the chain $(X_0^{(r)})_{r \in \mathbb{N}}$ is Leb-irreducible, where Leb is the Lebesgue measure. To facilitate the analysis, we make the following simplifying assumption.

(A2) For some fixed $\sigma_* > 0$, $w > 1$, and $\mathbf{c} \in \mathcal{C}$, the flow map $\mathbf{z} \mapsto F_{0|\sigma_*}(\mathbf{z} | \mathbf{c}; w)$ is a continuously differentiable diffeomorphism on $\mathbb{R}^d$.

**Proposition 5.** *Assume (A2). Then the Markov chain $(X_0^{(r)})_{r \in \mathbb{N}}$ is Leb-irreducible and has $p_0(\cdot | \mathbf{c}; w)$ as its unique invariant distribution.*

*Proof.* Let A be a Borel set of $\mathbb{R}^d$ such that Leb(A) $> 0$. It is enough to show that for any $\mathbf{x} \in \mathbb{R}^d$,

$$\int \mathbb{1}_A(F_{0|\sigma_*}(\mathbf{z} | \mathbf{c}; w)) \mathrm{N}(\mathbf{z}; \mathbf{x}, \mathbf{I}) \, d\mathbf{z} > 0 . \tag{B.7}$$

Denote by $\mathbf{z} \mapsto F_{0|\sigma_*}^{\leftarrow}(\mathbf{z}|\mathbf{c}; w)$ the inverse of $\mathbf{z} \mapsto F_{0|\sigma_*}(\mathbf{z}|\mathbf{c}; w)$. In addition, denote by $\det \mathrm{Jac}\, \Phi$, the Jacobian determinant of any function $\Phi : \mathbb{R}^d \to \mathbb{R}^d$ and note that since $\mathbf{z} \mapsto F_{0|\sigma_*}(\mathbf{z}|\mathbf{c}; w)$ is a diffeomorphism, it holds that $\mathbf{z} \mapsto \det \mathrm{Jac} F_{0|\sigma_*}(\mathbf{z}|\mathbf{c}; w)$ and $\mathbf{z} \mapsto \det \mathrm{Jac} F_{0|\sigma_*}^{\leftarrow}(\mathbf{z}|\mathbf{c}; w)$ are both non-zero. Then a change a variables yields

$$\int \mathbb{1}_{\mathsf{A}}(F_{0|\sigma_*}(\mathbf{z}|\mathbf{c}; w)) \mathrm{N}(\mathbf{z}; \mathbf{x}, \mathbf{I})\, \mathrm{d}\mathbf{z} = \int \mathbb{1}_{\mathsf{A}}(\mathbf{y}) \mathrm{N}(F_{0|\sigma_*}^{\leftarrow}(\mathbf{y}|\mathbf{c}; w); \mathbf{x}, \mathbf{I})\, |\det \mathrm{Jac} F_{0|\sigma_*}^{\leftarrow}(\mathbf{y}|\mathbf{c}; w)|\, \mathrm{d}\mathbf{y}\ ,$$

which implies (B.7). Finally, the fact that the Markov chain $(X_0^{(r)})_{r \in \mathbb{N}}$ has $p_0(\cdot|\mathbf{c}; w)$ as its unique invariant distribution follows from Meyn & Tweedie (2012, Proposition 10.1.1. and Theorem 10.4.4). □

**Example 1 details.** Recall that $p_\sigma^{\mathsf{cfg}}(\mathbf{x}_\sigma|\mathbf{c}; w) \propto g_\sigma(\mathbf{c}|\mathbf{x}_\sigma)^w p_\sigma(\mathbf{x}_\sigma)$, where $g_\sigma(\mathbf{c}|\mathbf{x}_\sigma) := \int g_0(\mathbf{c}|\mathbf{x}_0)\, p_{0|\sigma}(\mathbf{x}_0|\mathbf{x}_\sigma)\, \mathrm{d}\mathbf{x}_0$. In addition, note that since $p_0 = \mathcal{N}(0, 1)$, it holds that $p_{0|\sigma}(\mathbf{x}_0|\mathbf{x}_\sigma) = \mathrm{N}(0; \mathbf{x}_\sigma/(1 + \sigma^2), \sigma^2/(1 + \sigma^2))$. Since $g_0(\mathbf{c}|\mathbf{x}_0) = \mathrm{N}(\mathbf{c}; \mathbf{x}_0, \gamma^2)$, this implies that $g_\sigma(\mathbf{c}|\mathbf{x}_\sigma) = \mathrm{N}(\mathbf{c}; \mathbf{x}_\sigma/(1 + \sigma^2), \gamma^2 + \sigma^2/(1 + \sigma^2))$. Therefore, we get

$$p_\sigma^{\mathsf{cfg}}(\mathbf{x}_\sigma|\mathbf{c}; w) = \mathrm{N}(\mathbf{x}_\sigma; \mathrm{v}(w, \sigma^2) w \mathbf{c}/((1 + \sigma^2)\gamma^2 + \sigma^2), \mathrm{v}(w, \sigma^2))\ , \tag{B.8}$$

where

$$\mathrm{v}(w, \sigma^2) := (1 + \sigma^2)\frac{(1 + \sigma^2)\gamma^2 + \sigma^2}{w + \gamma^2(1 + \sigma^2) + \sigma^2}\ .$$

The limit part follows by a simple characteristic-function argument, since $p_\sigma^{\mathsf{cfg}}(\cdot|\mathbf{c}; w)$ is a Gaussian distribution and we make the assumption that it is the result of a Gaussian convolution. It remains to show that $\mathrm{v}(w, \sigma^2) < \mathbb{V}(X_\sigma)$, where we recall that $\mathbb{V}(X_\sigma) = \sigma^2 + \gamma^2/(\gamma^2 + w)$. However, since $w > 1$, we have that

$$\begin{aligned}
(1 + \sigma^2)^{-1}\mathbb{V}(X_\sigma) &= \frac{\gamma^2(1 + \sigma^2) + \sigma^2 w}{w + \gamma^2(1 + \sigma^2) + w\sigma^2} \\
&= 1 - \frac{w}{w + \gamma^2(1 + \sigma^2) + w\sigma^2} \\
&> 1 - \frac{w}{w + \gamma^2(1 + \sigma^2) + \sigma^2} \\
&= (1 + \sigma^2)^{-1}\mathrm{v}(w, \sigma^2)\ .
\end{aligned}$$

**Proofs for Proposition 4.** We assume that $p_0 = \mathcal{N}(0, 1)$ and $g_0(\mathbf{c}|\mathbf{x}_0) = \mathrm{N}(\mathbf{c}; \mathbf{x}_0, \gamma^2)$. Then it holds using that $p_0(\mathbf{x}_0|\mathbf{c}) = p_0^{\mathsf{cfg}}(\mathbf{x}_0|\mathbf{c}; 1)$ and (B.8) that

$$D_\sigma(\mathbf{x}_\sigma) = \frac{1}{1 + \sigma^2}\mathbf{x}_\sigma\ , \quad D_\sigma(\mathbf{x}_\sigma|\mathbf{c}) = \frac{\gamma^2}{\gamma^2(1 + \sigma^2) + \sigma^2}\mathbf{x}_\sigma + \frac{\sigma^2}{\gamma^2(1 + \sigma^2) + \sigma^2}\mathbf{c}\ .$$

The next lemma provides the expression of the distribution obtained by integrating exactly the PF-ODE (2.1) from $\sigma$ to zero using the CFG denoiser (2.3). We assume $\mathbf{c} = 0$, as the solution to the PF-ODE cannot, in general, be expressed using elementary functions when $\mathbf{c} \neq 0$. In that setting, we have the following standard result.

**Lemma 2.** *Let $w > 1$ and consider the first-order linear homogeneous ODE*

$$\frac{\mathrm{d}\mathbf{x}(\sigma)}{\mathrm{d}\sigma} = \left[\frac{w(1 + \gamma^2)\sigma}{\gamma^2 + (1 + \gamma^2)\sigma^2} + \frac{(1 - w)\sigma}{1 + \sigma^2}\right]\mathbf{x}(\sigma)\ , \quad \sigma \geq 0. \tag{B.9}$$

*Let $\mathbf{x}$ be the unique solution to (B.9) on the interval $[0, \sigma_*]$, for some $\sigma_* > 0$, satisfying the condition $\mathbf{x}(\sigma_*) = X_{\sigma_*}$. Then the value of the solution at $\sigma = 0$ is given by*

$$X_0 = \frac{\gamma^w(1 + \sigma_*^2)^{(w-1)/2}}{(\gamma^2 + (1 + \gamma^2)\sigma_*^2)^{w/2}}X_{\sigma_*}\ . \tag{B.10}$$

*Proof.* We define

$$P(\sigma) = \frac{w(1+\gamma^2)\sigma}{\gamma^2 + (1+\gamma^2)\sigma^2} + \frac{(1-w)\sigma}{1+\sigma^2} \ . \tag{B.11}$$

The function $P$ is continuous on $[0, \sigma_*]$ and thus, by the Cauchy–Lipschitz theorem, it admits a unique solution satisfying $\mathbf{x}(\sigma_*) = X_{\sigma_*}$. If $\mathbf{x}(\sigma_*) = 0$, then the unique solution to (B.9) with $\mathbf{x}(\sigma_*) = 0$ satisfies $\mathbf{x}(\sigma) = 0$ for any $\sigma \in [0, \sigma_*]$.

Now, assume $\mathbf{x}(\sigma_*) \neq 0$. Then $\mathbf{x}(\sigma) \neq 0$ for any $\sigma \in [0, \sigma_*]$, and integrating from $\sigma_*$ to zero yields

$$\int_{X_{\sigma_*}}^{X_0} \frac{\mathrm{d}\mathbf{x}(\sigma)}{\mathbf{x}(\sigma)} = \int_{\sigma_*}^0 P(s)\,\mathrm{d}s = F(0) - F(\sigma_*),$$

where $F(s) = \frac{w}{2}\log(\gamma^2 + (1+\gamma^2)s^2) + \frac{1-w}{2}\log(1+s^2)$ for any $s \in [0, \sigma_*]$. Note that $F(0) = \log\gamma^w$ and

$$F(\sigma_*) = \frac{w}{2}\log(\gamma^2 + (1+\gamma^2)\sigma_*^2) + \frac{1-w}{2}\log(1+\sigma_*^2) \ ;$$

thus,

$$\log\left|\frac{X_0}{X_{\sigma_*}}\right| = \log\frac{\gamma^w}{(\gamma^2 + (1+\gamma^2)\sigma_*^2)^{w/2}(1+\sigma_*^2)^{(1-w)/2}} \ .$$

Applying the exponential function to both sides and rearranging terms leads to

$$\left|\frac{X_0}{X_{\sigma_*}}\right| = \frac{\gamma^w(1+\sigma_*^2)^{(w-1)/2}}{(\gamma^2 + (1+\gamma^2)\sigma_*^2)^{w/2}} \ .$$

By continuity of the solution and the fact that $\mathbf{x}(\sigma) \neq 0$ for any $\sigma$, $\mathbf{x}$ has a fixed sign on $[0, \sigma_*]$, which yields the result. $\qquad\square$

As a sanity check, note that when $w = 0$, it holds that $X_0 = X_{\sigma_*}/\sqrt{1+\sigma_*^2}$. Thus, if $X_{\sigma_*} \sim p_{\sigma_*} = \mathcal{N}(0, 1+\sigma_*^2)$, then $X_0 \sim p_0$. If $w = 1$, we get $X_0 = \gamma X_{\sigma_*}/\sqrt{\gamma^2 + (1+\gamma^2)\sigma_*^2}$, and hence, if $X_\sigma \sim p_{\sigma_*}(\cdot|\mathbf{c}) = \mathcal{N}(0, \sigma_*^2 + \gamma^2/(1+\gamma^2))$, then $X_0 \sim p_0(\cdot|\mathbf{c})$.

**Analysis as $\sigma_* \to 0$ and $R \to \infty$.** We now consider the iterative procedure defined by, for any $r \geq 1$,

$$X_0^{(r)} = c(\sigma_*)\big(X_0^{(r-1)} + \sigma_* Z^{(r)}\big) \ , \tag{B.12}$$

where $c(\sigma_*) \coloneqq \gamma^w(1+\sigma_*^2)^{(w-1)/2}/(\gamma^2 + (1+\gamma^2)\sigma_*^2)^{w/2}$ and $(Z^{(r)})_{r\in\mathbb{N}}$ is a sequence of i.i.d. standard Gaussian random variables. Define $V_\infty(w) \coloneqq \sigma_*^2 c(\sigma_*)^2/(1 - c(\sigma_*)^2)$.

We denote by $\mathrm{W}_2(\mu, \nu)$ the 2-Wasserstein distance between distributions $\mu$ and $\nu$ in $\mathcal{P}(\mathbb{R}^d)$.

**Proposition** (Restatement of Proposition 4)**.** *Assume that $\mathbb{V}(X_0^{(0)}) < \infty$. For any $w > 1$, $(X_0^{(r)})_{r\in\mathbb{N}}$ converges to $\mathcal{N}(0, V_\infty(w))$ exponentially fast in the 2-Wasserstein distance, with rate proportional to $\sigma_*^2$. Furthermore,*

$$V_\infty(w) = V(w) + O(\sigma_*^2) \quad as \ \ \sigma_*^2 \to 0 \ .$$

*Proof.* We let $\mu_r \coloneqq \mathsf{Law}(X_0^{(r)})$ and $\mu_\infty \coloneqq \mathcal{N}(0, V_\infty(w))$. Defining the transition kernel $k(\mathbf{x}|\tilde{\mathbf{x}}) = \mathrm{N}(c(\sigma_*)\tilde{\mathbf{x}}, c(\sigma_*)^2\sigma_*^2)$, it holds that $\mu_r(\mathbf{x}) \coloneqq \int k(\mathbf{x}|\tilde{\mathbf{x}})\,\mu_{r-1}(\mathrm{d}\tilde{\mathbf{x}})$. Furthermore, it is easily checked that $\mu_\infty$ is the stationary distribution of the chain (B.12). Hence, $\mu_\infty(\mathbf{x}) = \int k(\mathbf{x}|\tilde{\mathbf{x}})\,\mu_\infty(\mathrm{d}\tilde{\mathbf{x}})$. Then,

$$\begin{aligned}
\mathrm{W}_2(\mu_r, \mu_\infty) &= \mathrm{W}_2\left(\int k(\cdot|\tilde{\mathbf{x}})\,\mu_{r-1}(\mathrm{d}\tilde{\mathbf{x}}), \int k(\cdot|\tilde{\mathbf{x}})\,\mu_\infty(\mathrm{d}\tilde{\mathbf{x}})\right)\\
&\leq c(\sigma_*)\mathrm{W}_2\left(\int q_{\sigma_*|0}(\cdot|\tilde{\mathbf{x}})\,\mu_{r-1}(\mathrm{d}\tilde{\mathbf{x}}), \int q_{\sigma_*|0}(\cdot|\tilde{\mathbf{x}})\,\mu_\infty(\mathrm{d}\tilde{\mathbf{x}})\right)\\
&\leq c(\sigma_*)\mathrm{W}_2(\mu_{r-1}, \mu_\infty) \ ,
\end{aligned}$$

where we used, in the second step, the fact that for any $c > 0$, if $X \sim \mu$ and $Y \sim \nu$, then $\mathrm{W}_2(\mathsf{Law}(cX), \mathsf{Law}(cY)) \leq c\mathrm{W}_2(\mu, \nu)$. Moreover, in the third step we used the fact that convolution

does not increase the Wasserstein distance (Santambrogio, 2015, Lemma 5.2). Iterating the bound we obtain that $W_2(\mu_r, \mu_\infty) \le c(\sigma_*)^r W_2(\mu_0, \mu_\infty)$, and as $\mu_0$ is square-integrable by assumption, $W_2(\mu_0, \mu_\infty) < \infty$. Thus, since

$$c(\sigma_*) = \frac{(\gamma^2 + \gamma^2 \sigma_*^2)^{w/2}}{\sqrt{1 + \sigma_*^2}(\gamma^2 + \gamma^2 \sigma_*^2 + \sigma_*^2)^{w/2}} < 1 \,,$$

we establish the exponential convergence of $(\mu_r)_{r \in \mathbb{N}}$ to $\mu_\infty$. Next, making a Taylor expansion of $V_\infty(w)$ around $\sigma_*^2 = 0$ yields that

$$\log c(\sigma_*) = w \log \gamma + \frac{w-1}{2} \log(1 + \sigma_*^2) - \frac{w}{2} \log(\gamma^2 + (1 + \gamma^2)\sigma_*^2)$$

$$= \frac{w-1}{2} \log(1 + \sigma_*^2) - \frac{w}{2} \log\left(1 + \frac{1+\gamma^2}{\gamma^2}\sigma_*^2\right)$$

$$= \left(\frac{w-1}{2} - w\frac{1+\gamma^2}{2\gamma^2}\right)\sigma_*^2 + o(\sigma_*^2)$$

$$= -\frac{\gamma^2 + w}{2\gamma^2}\sigma_*^2 + o(\sigma_*^2) \,,$$

which in turn yields the expansion $1 - c(\sigma_*)^2 = \frac{\gamma^2 + w}{\gamma^2}\sigma_*^2 + o(\sigma_*^2)$. Substituting the same into $V_\infty(w)$ gives

$$V_\infty(w) = \frac{\sigma_*^2\left(1 - \frac{\gamma^2 + w}{\gamma^2}\sigma_*^2 + o(\sigma_*^2)\right)}{\frac{\gamma^2 + w}{\gamma^2}\sigma_*^2 + o(\sigma_*^2)} = \frac{1 - \frac{\gamma^2 + w}{\gamma^2}\sigma_*^2 + o(\sigma_*^2)}{\frac{\gamma^2 + w}{\gamma^2} + o(1)} \,.$$

Developing the right-hand side further, we arrive at

$$V_\infty(w) = \left(1 - \frac{\gamma^2 + w}{\gamma^2}\sigma_*^2 + o(\sigma_*^2)\right)\left(\frac{\gamma^2 + w}{\gamma^2}\right)^{-1}\left(1 + \frac{o(\sigma_*^2)}{\frac{\gamma^2 + w}{\gamma^2}}\right)^{-1}$$

$$= \frac{\gamma^2}{\gamma^2 + w}\left(1 - \frac{\gamma^2 + w}{\gamma^2}\sigma_*^2 + o(\sigma_*^2)\right)(1 + o(\sigma_*^2))$$

$$= \frac{\gamma^2}{\gamma^2 + w}\left(1 - \frac{\gamma^2 + w}{\gamma^2}\sigma_*^2 + o(\sigma_*^2)\right)$$

$$= \frac{\gamma^2}{\gamma^2 + w} - \sigma_*^2 + o(\sigma_*^2) \,,$$

from which the claim follows. $\qquad\square$

**Analysis as $\sigma_* \to 0$ and $R < \infty$.** To investigate the behavior at finite $R$, we initialize the recursion (B.12) with $X_0^{(0)} \sim p_0(\cdot|\mathbf{c}) = \mathcal{N}(0, V_0)$, where $V_0 := \gamma^2/(\gamma^2 + 1)$. For any $R > 0$, the final iterate satisfies $X_0^{(R)} \sim \mathcal{N}(0, V_R(w))$, where

$$V_R(w) = c(\sigma_*)^{2R}V_0 + \frac{c(\sigma_*)^2\,\sigma_*^2\,(1 - c(\sigma_*)^{2R})}{1 - c(\sigma_*)^2}.$$

We plot the absolute error $|V_R(w) - V(w)|$ as a function of $\sigma_*$ for various values of $R$ in Figure 10. We observe that for small $R$, the error is minimized at higher values of $\sigma_*$, and that the minimal error achieved for small $\sigma_*$ is lower than that observed at larger $\sigma_*$.

## C  INSIGHTS ON THE CONSIDERED METHODS.

**CFG++ with Heun sampler.** Let $\lambda \in [0, 1]$. Following Chung et al. (2025, Appendix A), CFG++ with the Euler discretization of the PF-ODE (2.2) corresponds to the update

$$X_t = D_{t+1}^{\mathsf{cfg}}(X_{t+1}|\mathbf{c}; \lambda) + \frac{\sigma_t}{\sigma_{t+1}}\big(X_{t+1} - D_{t+1}(X_{t+1})\big),$$

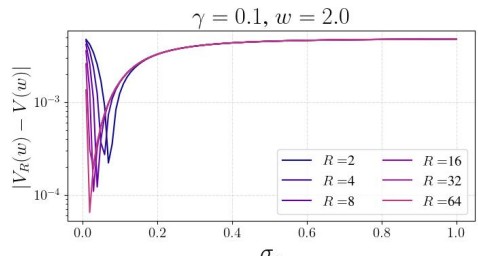 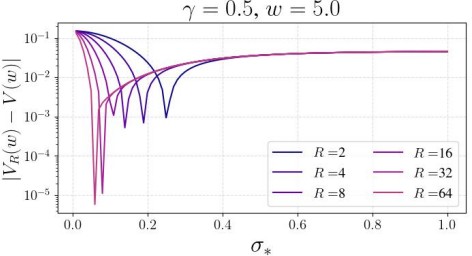

Figure 10: The difference $|V_R(w) - V(w)|$ as a function of $\sigma_*$ for various values of $R$.

which, upon replacing with the definition of the CFG denoiser (2.3), writes as

$$
X_t = \left[ (1 - \frac{\sigma_t}{\sigma_{t+1}}) D_{t+1}(X_{t+1}) + \lambda \big( D_{t+1}(X_{t+1}|\mathbf{c}) - D_{t+1}(X_{t+1}) \big) \right] + \frac{\sigma_t}{\sigma_{t+1}} D_{t+1}(X_{t+1})
$$

$$
= \frac{\sigma_t}{\sigma_{t+1}} X_{t+1} + \big( 1 - \frac{\sigma_t}{\sigma_{t+1}} \big) D_{t+1}^{\mathsf{cfg}}(X_{t+1}|\mathbf{c}; w_{t+1}^{++}),
$$

with $w_{t+1}^{++} = \lambda \sigma_{t+1} / (\sigma_{t+1} - \sigma_t)$. In short, CFG++ can be viewed as a DDIM update based on the CFG denoiser with a $\sigma$-dependent guidance scale. Hence, in our experiment, we implement CFG++ with Heun sampler by using Algorithm 2 with the CFG denoiser (2.3) and time-dependent guidance factors $(w_t^{++})_{t=1}^T$.

**Feynman–Kac correctors.** In Skreta et al. (2025), a sequential Monte Carlo (SMC) algorithm is proposed to target the distribution sequence $(p_\sigma^{\mathsf{cfg}}(\cdot|\mathbf{c}; w))_\sigma$. In what follows, we provide a simple derivation of the (approximate) recursion relating two consecutive distributions $p_{\sigma_t}^{\mathsf{cfg}}(\cdot|\mathbf{c}; w)$ and $p_{\sigma_{t+1}}^{\mathsf{cfg}}(\cdot|\mathbf{c}; w)$, for $\sigma_t < \sigma_{t+1}$. This serves as the basis for the SMC algorithm resulting from Skreta et al. (2025, Proposition 3.1). Write

$$
p_{\sigma_t}^{\mathsf{cfg}}(\mathbf{x}_t|\mathbf{c}; w) \propto \int p_{\sigma_t}^{\mathsf{cfg}}(\mathbf{x}_t|\mathbf{c}; w) q_{\sigma_{t+1}|\sigma_t}(\mathbf{x}_{t+1}|\mathbf{x}_t) \, \mathrm{d}\mathbf{x}_{t+1}
$$

$$
\propto \int p_{\sigma_t}(\mathbf{x}_t|\mathbf{c})^w p_{\sigma_t}(\mathbf{x}_t)^{1-w} q_{\sigma_{t+1}|\sigma_t}(\mathbf{x}_{t+1}|\mathbf{x}_t) \, \mathrm{d}\mathbf{x}_{t+1}
$$

$$
\propto \int \left[ p_{\sigma_t}(\mathbf{x}_t|\mathbf{c}) q_{\sigma_{t+1}|\sigma_t}(\mathbf{x}_{t+1}|\mathbf{x}_t) \right]^w \left[ p_{\sigma_t}(\mathbf{x}_t) q_{\sigma_{t+1}|\sigma_t}(\mathbf{x}_{t+1}|\mathbf{x}_t) \right]^{1-w} \, \mathrm{d}\mathbf{x}_{t+1}
$$

$$
\propto \int p_{\sigma_t|\sigma_{t+1}}(\mathbf{x}_t|\mathbf{x}_{t+1}, \mathbf{c})^w p_{\sigma_t|\sigma_{t+1}}(\mathbf{x}_t|\mathbf{x}_{t+1})^{1-w} p_{\sigma_{t+1}}^{\mathsf{cfg}}(\mathbf{x}_{t+1}|\mathbf{c}; w) \, \mathrm{d}\mathbf{x}_{t+1} \quad \text{(C.1)}
$$

$$
\propto \int \mathscr{Z}_{t+1}^{\mathsf{cfg}}(\mathbf{x}_{t+1}; w) p_{\sigma_t|\sigma_{t+1}}^{\mathsf{cfg}}(\mathbf{x}_t|\mathbf{x}_{t+1}, \mathbf{c}; w) p_{\sigma_{t+1}}^{\mathsf{cfg}}(\mathbf{x}_{t+1}|\mathbf{c}; w) \, \mathrm{d}\mathbf{x}_{t+1}, \quad \text{(C.2)}
$$

where we have defined

$$
p_{\sigma_t|\sigma_{t+1}}(\mathbf{x}_t|\mathbf{x}_{t+1}, \mathbf{c}) := \frac{p_{\sigma_t}(\mathbf{x}_t|\mathbf{c}) q_{\sigma_{t+1}|\sigma_t}(\mathbf{x}_{t+1}|\mathbf{x}_t)}{p_{\sigma_{t+1}}(\mathbf{x}_{t+1}|\mathbf{c})}, \quad \text{(C.3)}
$$

$$
p_{\sigma_t|\sigma_{t+1}}(\mathbf{x}_t|\mathbf{x}_{t+1}) := \frac{p_{\sigma_t}(\mathbf{x}_t) q_{\sigma_{t+1}|\sigma_t}(\mathbf{x}_{t+1}|\mathbf{x}_t)}{p_{\sigma_{t+1}}(\mathbf{x}_{t+1})}, \quad \text{(C.4)}
$$

$$
p_{\sigma_t|\sigma_{t+1}}^{\mathsf{cfg}}(\mathbf{x}_t|\mathbf{x}_{t+1}, \mathbf{c}) \propto p_{\sigma_t|\sigma_{t+1}}(\mathbf{x}_t|\mathbf{x}_{t+1}, \mathbf{c})^w p_{\sigma_t|\sigma_{t+1}}(\mathbf{x}_t|\mathbf{x}_{t+1})^{1-w}, \quad \text{(C.5)}
$$

and $\mathscr{Z}_{t+1}^{\mathsf{cfg}}(\mathbf{x}_{t+1}; w) := \int p_{\sigma_t|\sigma_{t+1}}(\mathbf{x}_t|\mathbf{x}_{t+1}, \mathbf{c})^w p_{\sigma_t|\sigma_{t+1}}(\mathbf{x}_t|\mathbf{x}_{t+1})^{1-w} \, \mathrm{d}\mathbf{x}_{t+1}$.

The recursion in Skreta et al. (2025) approximates (C.1) by replacing (C.3) and (C.4) with their Gaussian approximations. More precisely, consider, *e.g.*, the Gaussian approximations

$$
p_{\sigma_t|\sigma_{t+1}}(\mathbf{x}_t|\mathbf{x}_{t+1}, \mathbf{c}) \approx \mathrm{N}(\mathbf{x}_t; \mu_{t|t+1}(\mathbf{x}_{t+1}|\mathbf{c}), \sigma_{t|t+1}^2 \mathbf{I}),
$$

$$
p_{\sigma_t|\sigma_{t+1}}(\mathbf{x}_t|\mathbf{x}_{t+1}) \approx \mathrm{N}(\mathbf{x}_t; \mu_{t|t+1}(\mathbf{x}_{t+1}), \sigma_{t|t+1}^2 \mathbf{I})
$$

stemming from DDPM, where

$$\mu_{t|t+1}(\mathbf{x}_{t+1}|\mathbf{c}) \coloneqq \frac{\sigma_t^2}{\sigma_{t+1}^2}X_{t+1} + \left(1 - \frac{\sigma_t^2}{\sigma_{t+1}^2}\right)D_{\sigma_{t+1}}(X_{t+1}|\mathbf{c}) ,$$

$$\mu_{t|t+1}(\mathbf{x}_{t+1}) \coloneqq \frac{\sigma_t^2}{\sigma_{t+1}^2}X_{t+1} + \left(1 - \frac{\sigma_t^2}{\sigma_{t+1}^2}\right)D_{\sigma_{t+1}}(X_{t+1}) ,$$

and $\sigma_{t|t+1}^2 \coloneqq \sigma_t^2(\sigma_{t+1}^2 - \sigma_t^2)/\sigma_{t+1}^2$. The Gaussian approximation of $p_{\sigma_t|\sigma_{t+1}}^{\mathsf{cfg}}(\cdot|\mathbf{x}_{t+1}, \mathbf{c})$ is formed by plugging the two previous Gaussian approximations into (C.5). When two Gaussian densities $\mathcal{N}(\mathbf{x}_t; m_1, \Sigma)$ and $\mathcal{N}(\mathbf{x}_t; m_2, \Sigma)$ are raised to powers $w$ and $1-w$, respectively, their product is proportional to another Gaussian density with the same covariance matrix $\Sigma$ and a new mean $wm_1 + (1-w)m_2$; hence,

$$p_{\sigma_t|\sigma_{t+1}}^{\mathsf{cfg}}(\mathbf{x}_t|\mathbf{x}_{t+1}, \mathbf{c}) \approx \mathrm{N}\left(\mathbf{x}_t; \frac{\sigma_t^2}{\sigma_{t+1}^2}\mathbf{x}_{t+1} + \left(1 - \frac{\sigma_t^2}{\sigma_{t+1}^2}\right)D_{\sigma_{t+1}}^{\mathsf{cfg}}(\mathbf{x}_{t+1}|\mathbf{c}), \sigma_{t|t+1}^2\mathbf{I}\right) .$$

It remains to derive an approximation of the weighting term $\mathcal{Z}_{t+1}^{\mathsf{cfg}}(\mathbf{x}_{t+1}; w)$. Plugging again the two previous Gaussian approximations into the definition of the latter yields

$$\mathcal{Z}_{t+1}^{\mathsf{cfg}}(\mathbf{x}_{t+1}; w) \approx \int \mathrm{N}(\mathbf{x}_t; \mu_{t|t+1}(\mathbf{x}_{t+1}|\mathbf{c}), \sigma_{t|t+1}^2\mathbf{I})^w \mathrm{N}(\mathbf{x}_t; \mu_{t|t+1}(\mathbf{x}_{t+1}), \sigma_{t|t+1}^2\mathbf{I})^{1-w}\, \mathrm{d}\mathbf{x}_t$$

$$\propto \exp\left(\frac{w(w-1)(\sigma_{t+1}^2 - \sigma_t^2)}{2\sigma_{t+1}^2\sigma_t^2}\big\|D_{\sigma_{t+1}}(\mathbf{x}_{t+1}|\mathbf{c}) - D_{\sigma_{t+1}}(\mathbf{x}_{t+1})\big\|_2^2\right) .$$

---

**Algorithm 2** Heun sampler

**Require:** Initial Sample $X_{\sigma_{\mathrm{init}}}$
**Require:** Denoiser $(X, \sigma) \mapsto D_\sigma(X)$
**Require:** Total number of steps $T$
**Create:** $\{\sigma_t\}_{t=0}^T$ such that $\sigma_T = \sigma_{\mathrm{init}}$ and $\sigma_0 \approx 0$
**for** $t = T - 1$ **to** $0$ **do**
$\quad X_t' \leftarrow (\sigma_t/\sigma_{t+1})X_{t+1} + (1 - \sigma_t/\sigma_{t+1})D_{\sigma_{t+1}}(X_{t+1})$
$\quad X_t \leftarrow X_{t+1} + \frac{\sigma_t - \sigma_{t+1}}{2}\left[\frac{X_t' - D_{\sigma_t}(X_t')}{\sigma_t} + \frac{X_{t+1} - D_{\sigma_{t+1}}(X_{t+1})}{\sigma_{t+1}}\right]$
**end for**
**Output:** $D_{\sigma_0}(X_0)$

---

# D  DELAYED GUIDANCE

In Proposition 3 and Corollary 1 we have proposed two novels expression for the score of the tilted distribution $p_0(\cdot|\mathbf{c}; w)$. Similarly to the CFG score, it can be shown that the Rényi divergence term involved in Proposition 3 is negligible. Thus, we get the following new denoiser approximation:

$$D_\sigma^{\mathsf{del}}(\mathbf{x}_\sigma|\mathbf{c}; w) \coloneqq wD_{\sigma_-}(\mathbf{x}_\sigma|\mathbf{c}) + (1-w)D_{\sigma_+}(\mathbf{x}_\sigma) \tag{D.1}$$

where we recall that $\sigma_+ = \sigma\sqrt{(w-1)/\delta}$ and $\sigma_- = \sigma\sqrt{w/(1+\delta)}$. Note that for the practical implementation, $\delta$ cannot be too small since in this case we would have $\sigma_+ \gg \sigma$ and the inputs to the denoiser at noise level $\sigma_+$ would be too of this distribution.

In Table 9, we report FID and FD$_{\mathrm{DINOv2}}$ scores for various combinations of $(w, \delta)$. We observe that using a slightly reduced guidance scale in conjunction with the approximation (D.1) substantially improves FD$_{\mathrm{DINOv2}}$, albeit at the cost of a higher FID. Conversely, increasing the guidance scale to 2.3—which achieves the best FID in Table 1—leads to a notable degradation in both FID and FD$_{\mathrm{DINOv2}}$ performance. We provide a qualitative assessment of D-CFɪG in Appendix E.5 using the EDM2-XXL model. Overall, it tends to produce sharp and diverse images, though often with less detailed or rich textures compared to CFIG.

Table 9: Comparison of average FID, FD$_{\text{DINOv2}}$ on `ImageNet`-512 for EDM2-S with 50k samples. Besides $w$, we use the default hyperparameters given in Table 4.

| | Quality metrics | |
|---|---|---|
| Algorithm | FID ↓ | FD$_{\text{DINOv2}}$ ↓ |
| **EDM2-S** | | |
| D-CFIG ($w = 2, \delta = 0.85$) | 3.63 | 56.53 |
| D-CFIG ($w = 2, \delta = 0.90$) | 3.57 | 54.81 |
| D-CFIG ($w = 2, \delta = 0.95$) | 3.53 | 54.13 |
| D-CFIG ($w = 2.3, \delta = 0.85$) | 6.35 | 86.79 |
| D-CFIG ($w = 2.3, \delta = 0.90$) | 5.51 | 74.31 |
| D-CFIG ($w = 2.3, \delta = 0.95$) | 5.02 | 65.63 |

# E    QUALITATIVE ASSESSMENT

In this section we qualitatively assess CFIG. We begin by juxtaposing its outputs with those of competing methods, then vary individual parameters to observe their effect on the generated images. All figures share the default hyperparameters from Tables 4 and 5; only the parameter indicated above each figure is altered.

## E.1    COMPARISON OF CFIG, INTERVAL-CFG, CFG AND CFG++ WITH A HIGH GUIDANCE SCALE

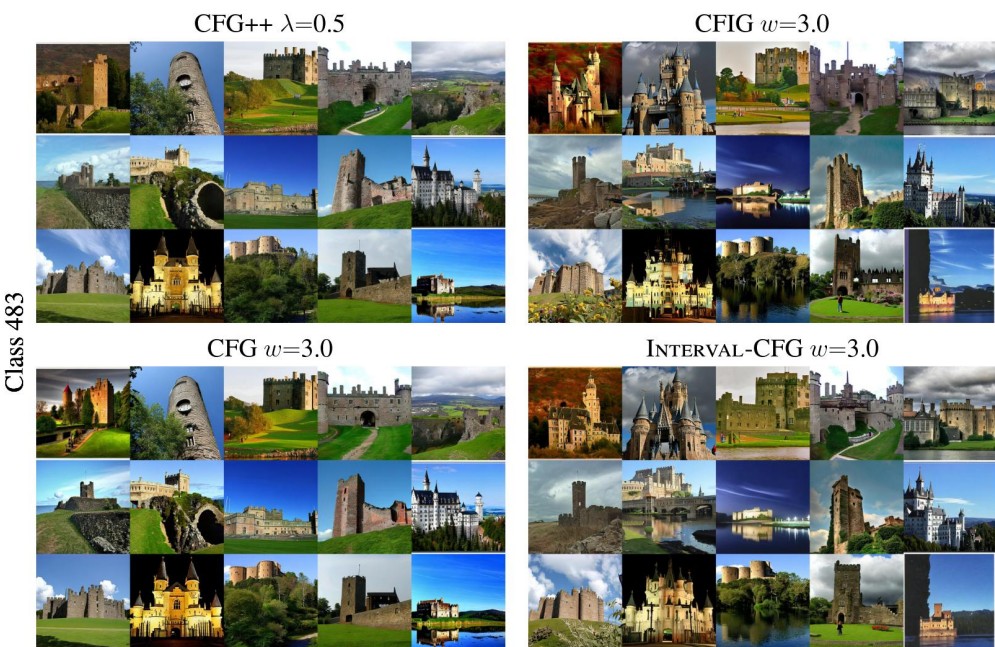

Class 555

CFG++ $\lambda$=0.5    CFIG $w$=3.0

CFG $w$=3.0    INTERVAL-CFG $w$=3.0

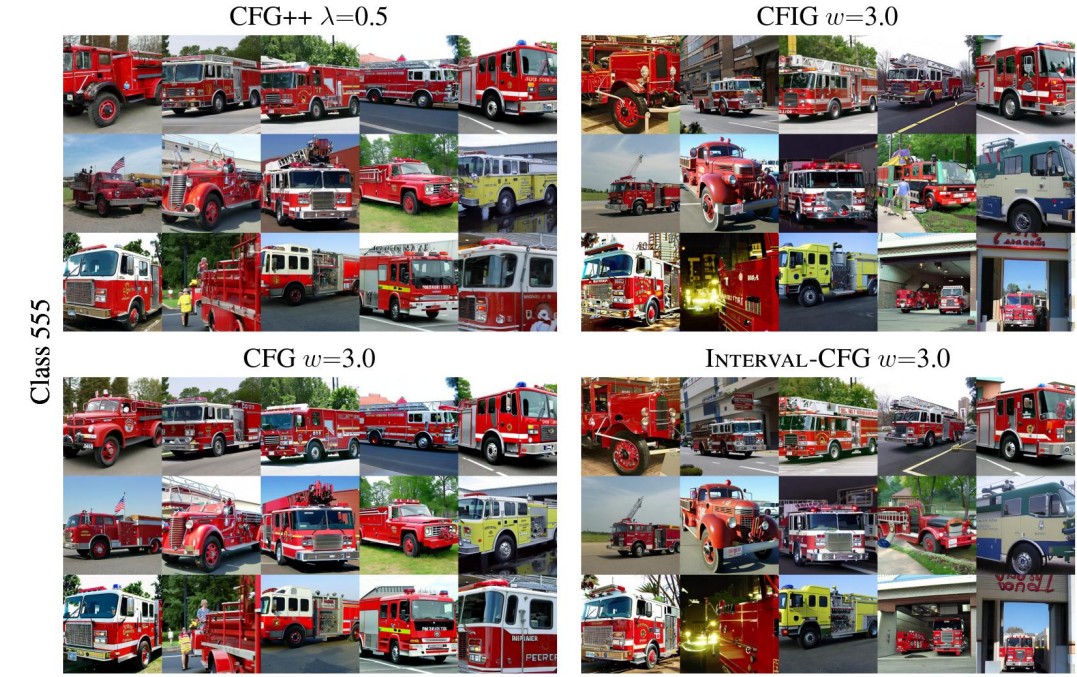

Class 962

CFG++ $\lambda$=0.5    CFIG $w$=3.0

CFG $w$=3.0    INTERVAL-CFG $w$=3.0

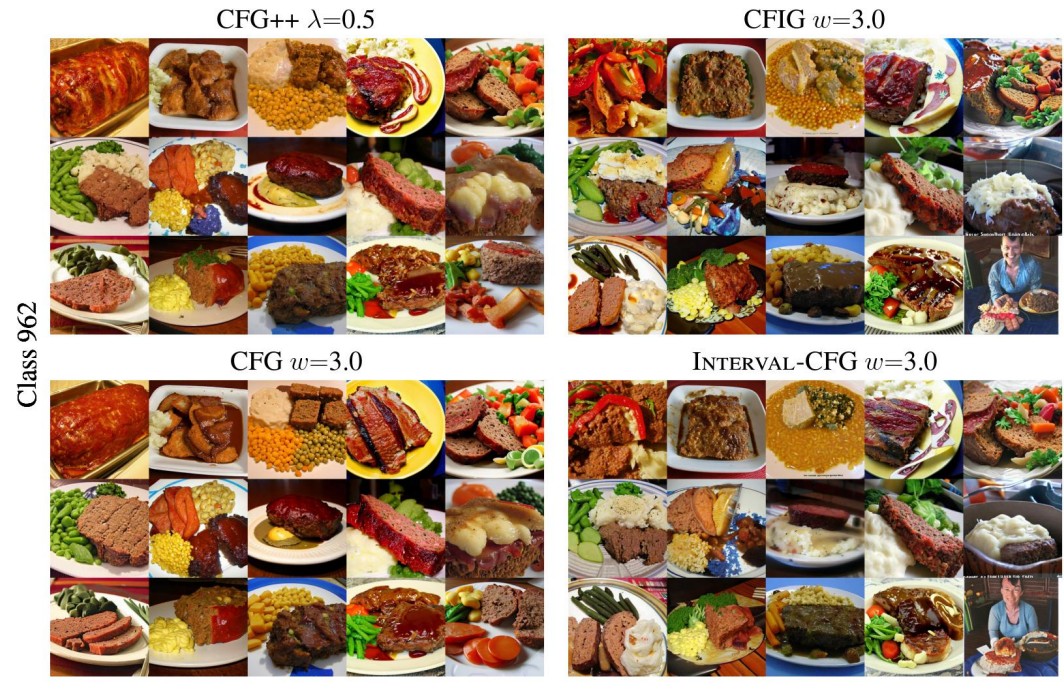

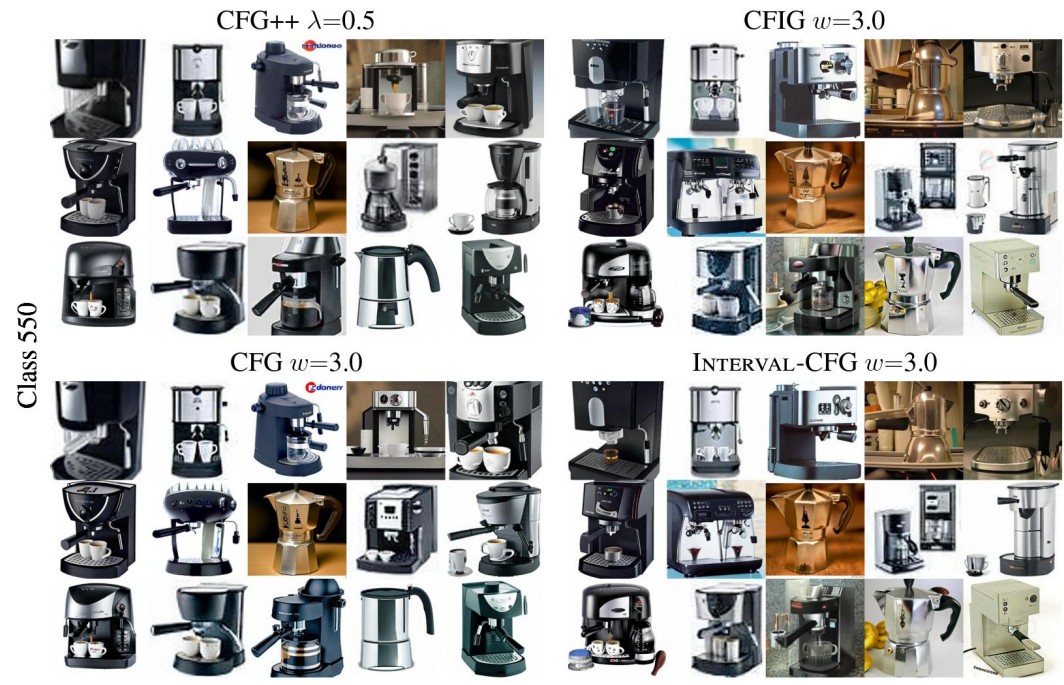

## E.2 VARYING $w$

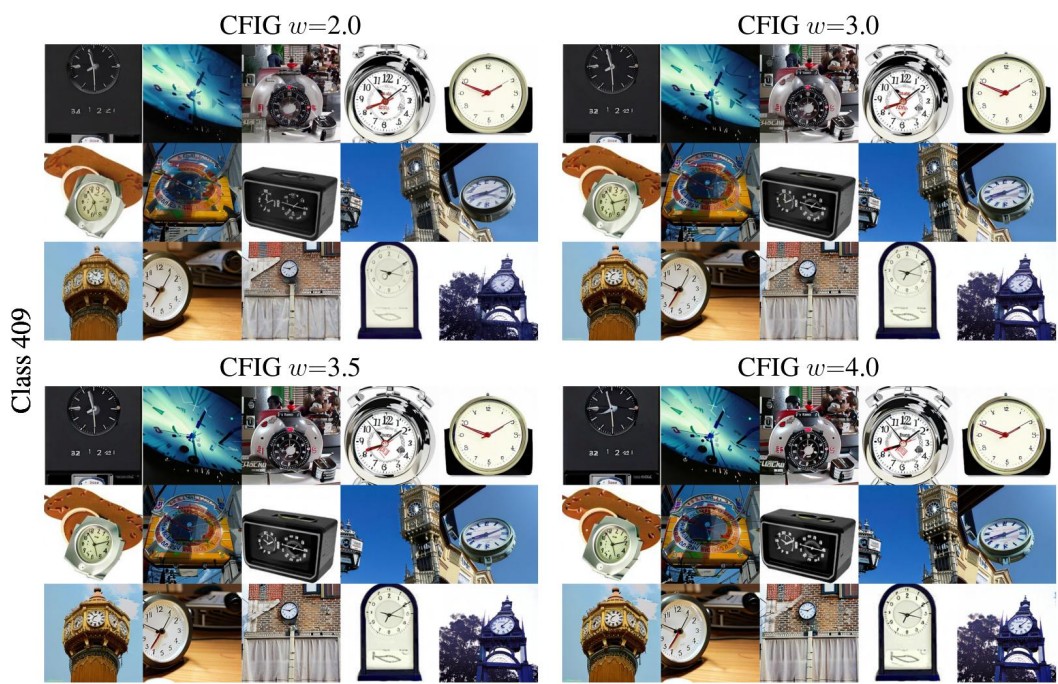

CFIG *w*=2.0          CFIG *w*=3.0

CFIG *w*=3.5          CFIG *w*=4.0

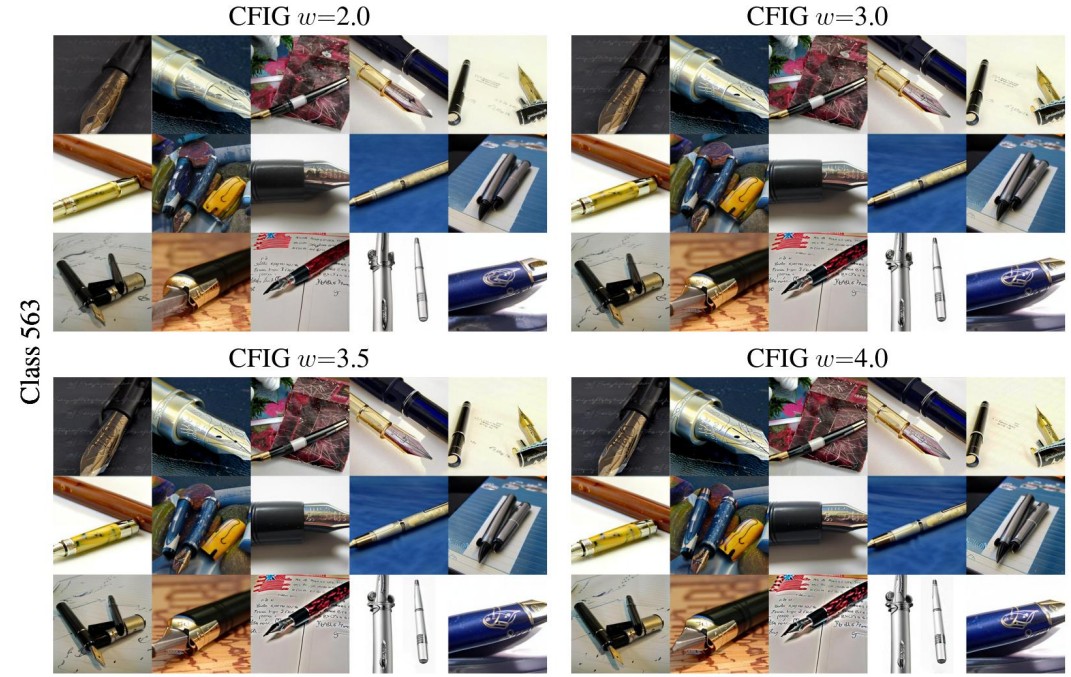

Class 563

CFIG *w*=2.0          CFIG *w*=3.0

CFIG *w*=3.5          CFIG *w*=4.0

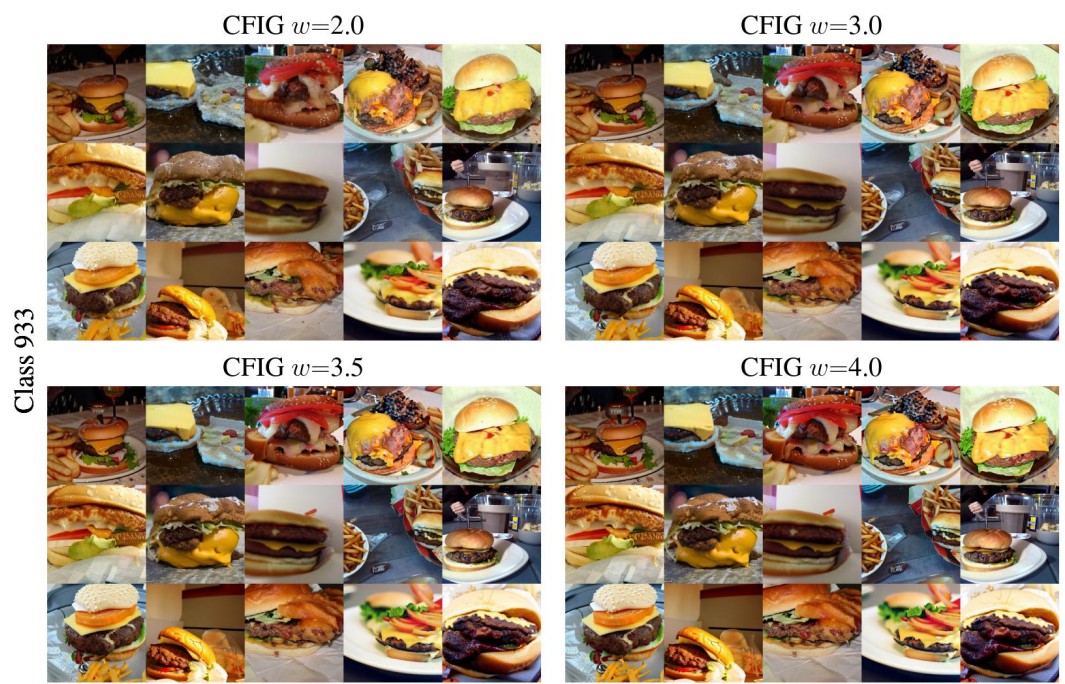

Class 933

## E.3 VARYING $\sigma_*$

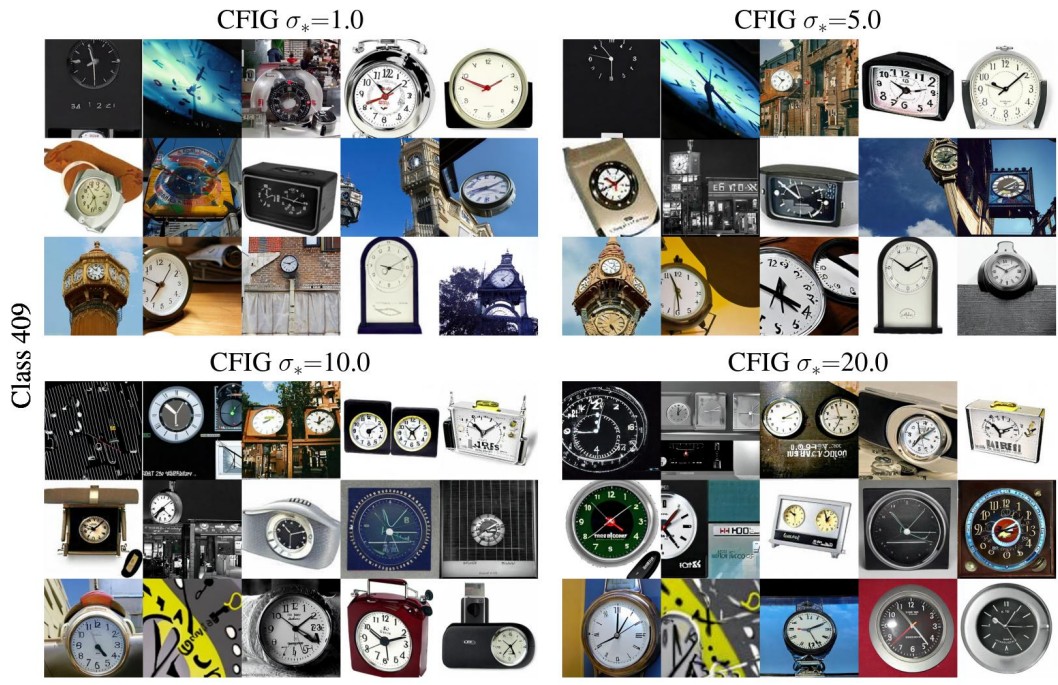

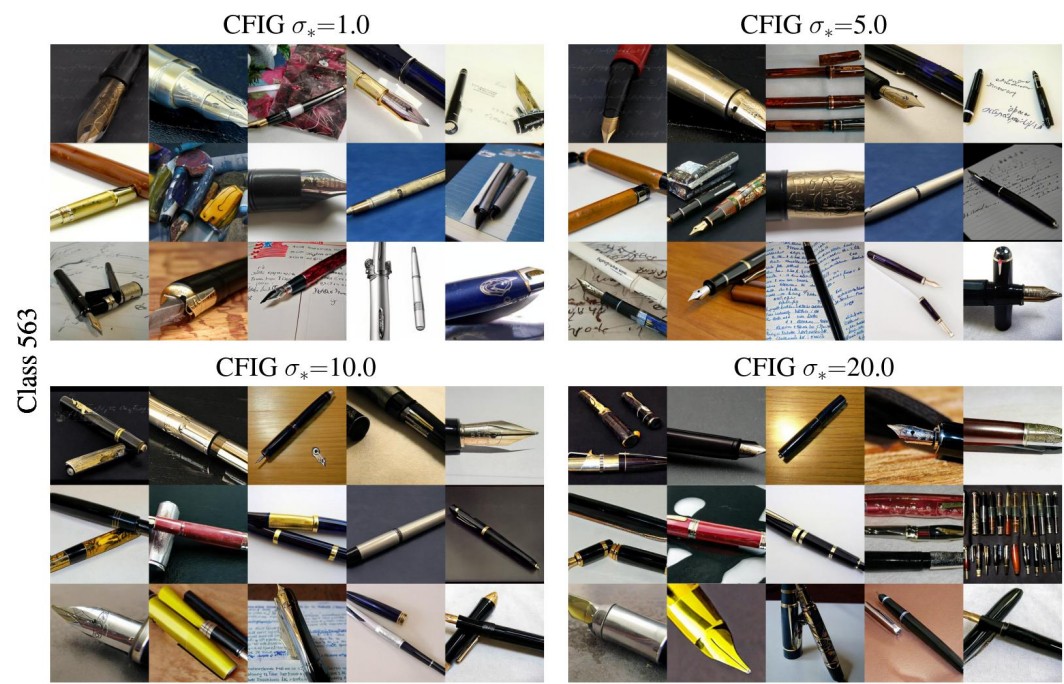

CFIG $\sigma_*$=1.0        CFIG $\sigma_*$=5.0

CFIG $\sigma_*$=10.0        CFIG $\sigma_*$=20.0

Class 933

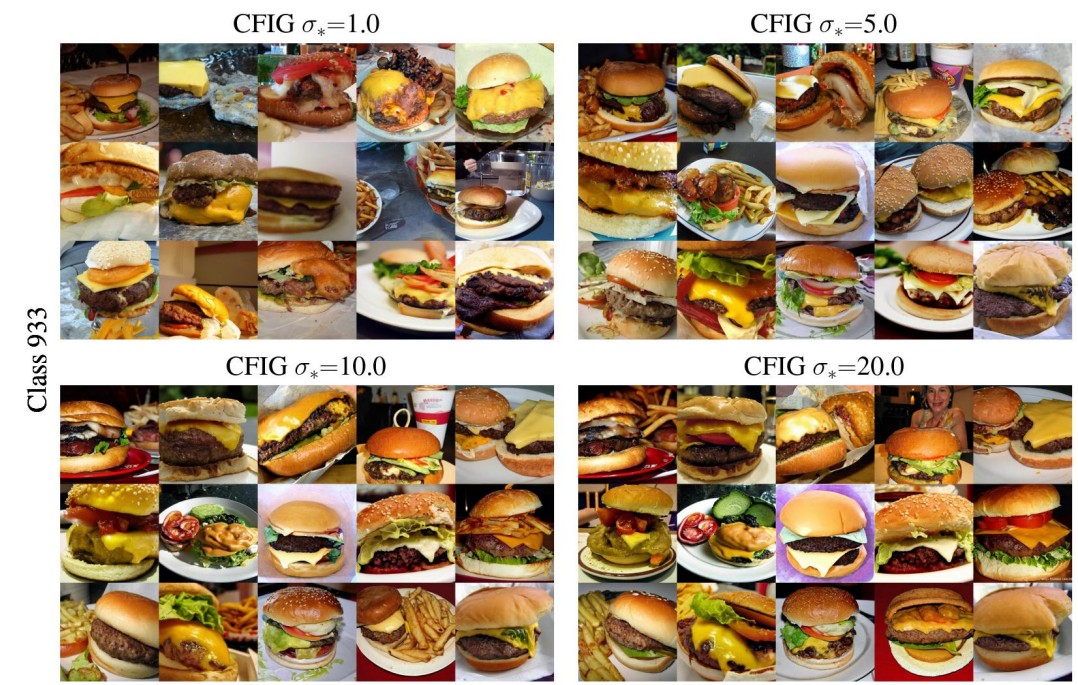

### E.4 VARYING $R$

CFIG $R$=1        CFIG $R$=2

CFIG $R$=3        CFIG $R$=4

Class 409

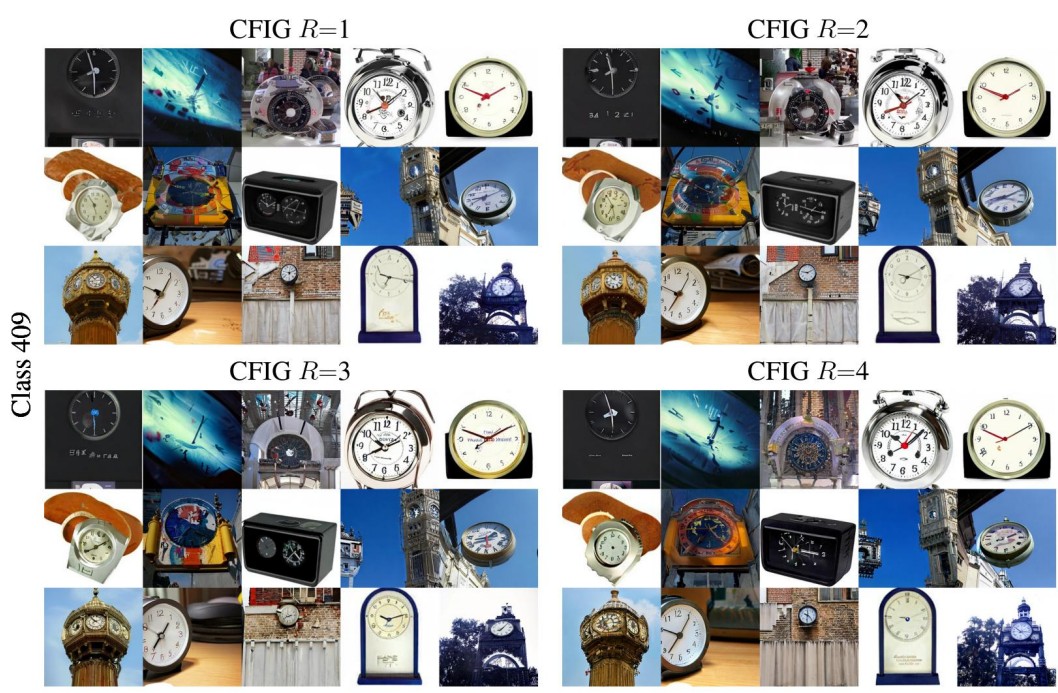

CFIG $R=1$ CFIG $R=2$

Class 563

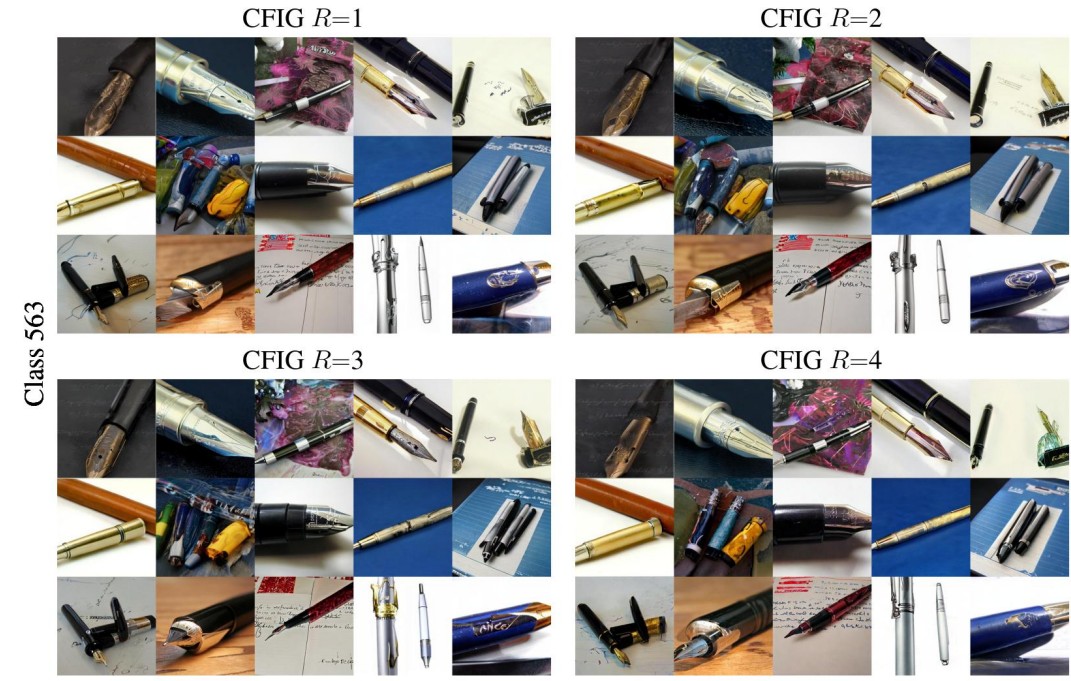

CFIG $R=3$ CFIG $R=4$

CFIG $R=1$ CFIG $R=2$

Class 933

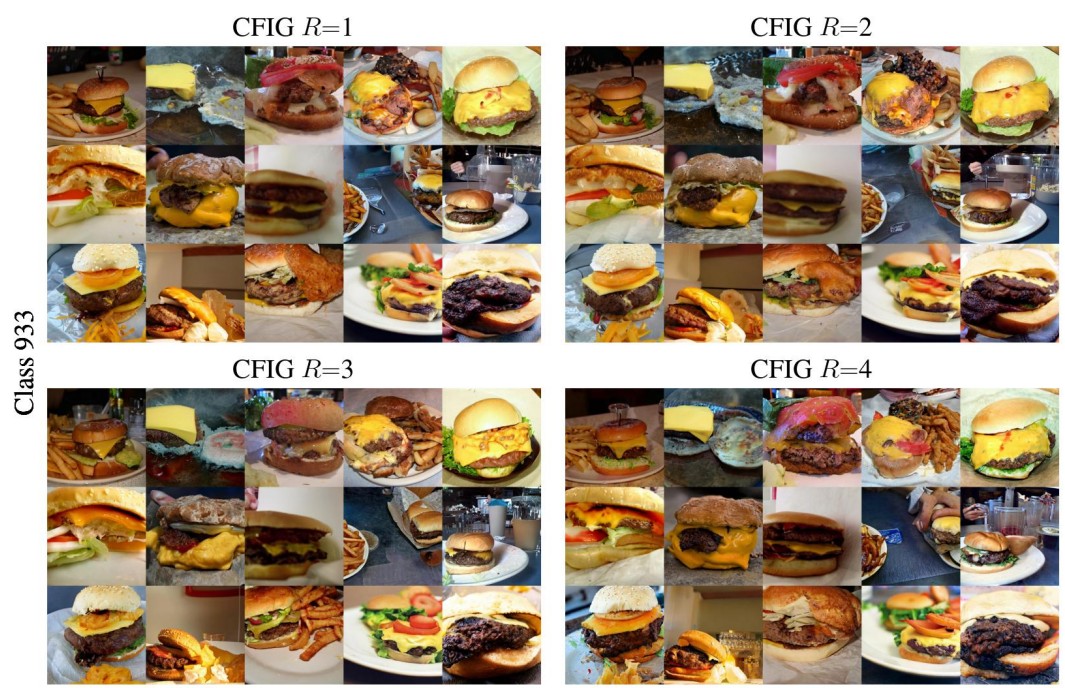

CFIG $R=3$ CFIG $R=4$

## E.5 Varying $\delta$ in D-CFiG

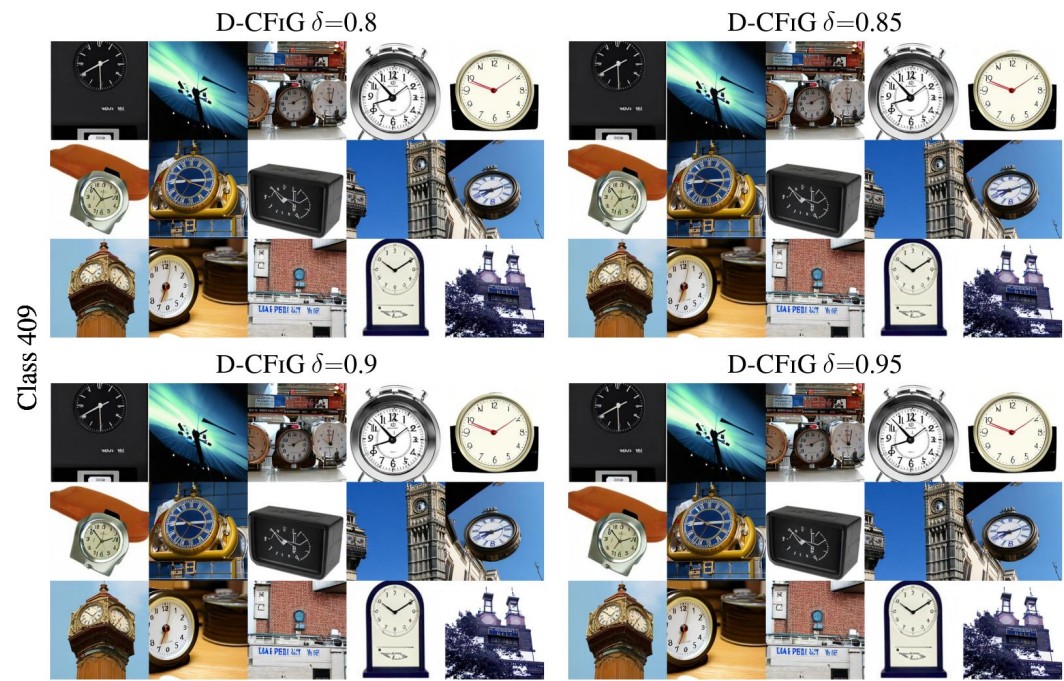

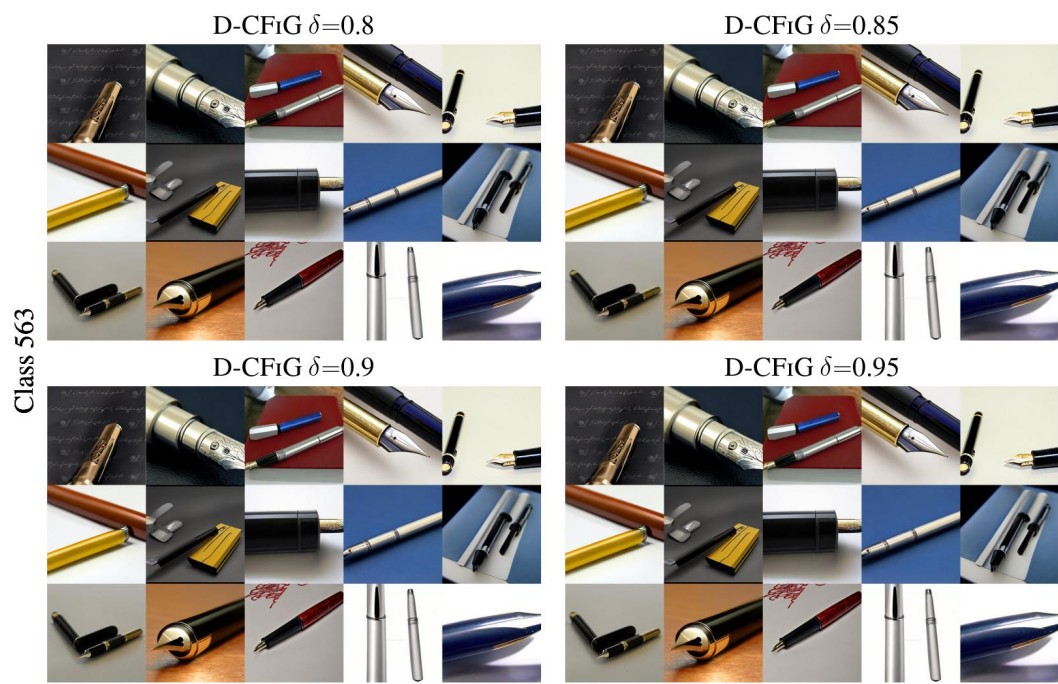

D-CFɪG $\delta$=0.8         D-CFɪG $\delta$=0.85

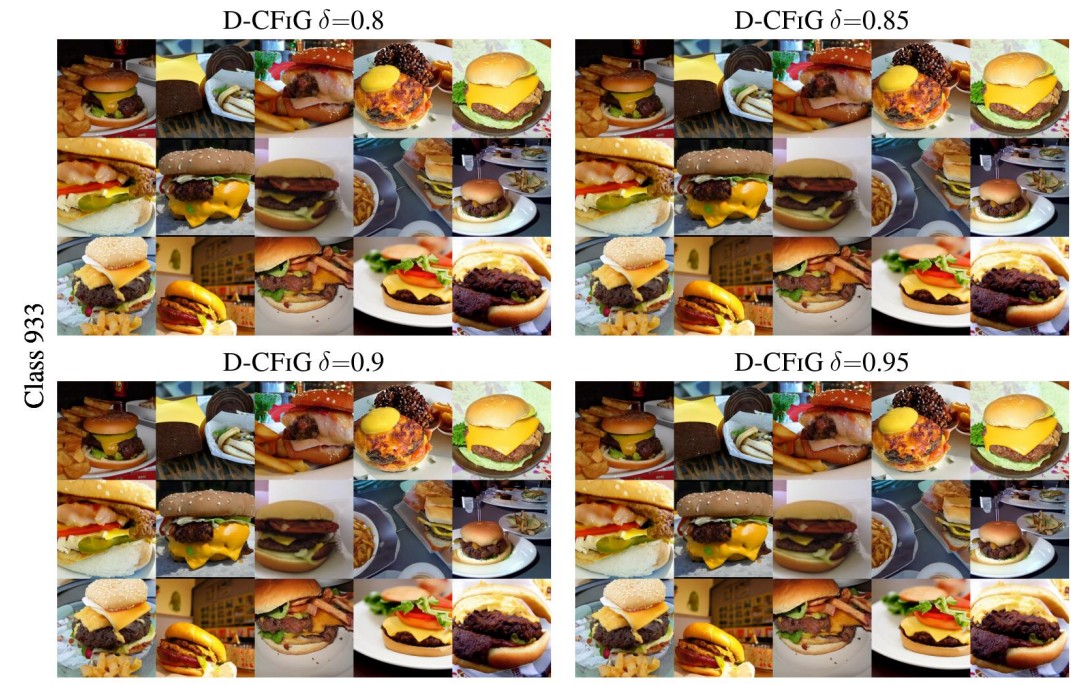

D-CFɪG $\delta$=0.9         D-CFɪG $\delta$=0.95

Class 933

