# OpenReview forum: "Conditional Diffusion Models with Classifier-Free Iterative Guidance"
_ICLR.cc/2026/Conference — Submitted to ICLR 2026_

### Official Review · Reviewer_M9WA · 2025-10-22

**Soundness:** 4
**Presentation:** 2
**Contribution:** 4
**Rating:** 6
**Confidence:** 5

**Summary:**

The paper theoretically analyzes the bias of CFG with the aid of the Renyi divergence. To be more detailed, the authors describe quantitatively the discrepancy between scores of conditional distribution and CFG-guided one. Benefiting from this finding, the bias in CFG could be expressed as some function with respect to the noise schedule term $\sigma$, enabling more precise control of guidance weight and trajectories in CFG-guided synthesis. Both qualitative and quantitative experiments confirm the efficacy of the proposed method.

**Strengths:**

- The quantitative description of the bias in CFG is solid and insightful. The error analysis is also precise and of great soundness.
- The motivation of the proposed method is clear and easy to follow. Reducing the guidance weight at the stage which may lead to huge bias indeed improves the synthesis performance.
- The experimental results are comprehensive. Results under different settings consistently confirm the efficacy, which is greatly convincing.

**Weaknesses:**

- Despite the solid theoretical part, the proposed pipeline is relatively not well-explored. First, as is reported in Fig. 3, increasing Repetitions fails to further improve the performance. Involving only one or two iterations seems a local correction or refinement and inconsistent with "iterative guidance", since most denoising steps before $T_0$ fall back to the native CFG with a smaller weight. Besides, as is claimed in Prop. 2, the discrepancy is an $O(\sigma^2)$ term when $\sigma$ is small. This seems to conflict with the results in Fig. 3, in which extremely small $\sigma_*$ almost corrupts both FID and FD_DINOv2. Note that EDM2 does not need large CFG weight, why is performance so sensitive to $\sigma_*$? Finally, to more properly use the current name "iterative guidance", could the authors come up with some pipelines to shorten the native CFG period and use more iterations and some dynamic $\sigma_*$ and guidance weights to better follow the theory?
- The current notation system is kind of hard to follow. Four notations $p$, $q$, $\pi$, and $g$ all represent probabilities, making the understanding a bit difficult. Besides, what does the last term in ODE_Solver in Alg. 1 mean? From my understanding in the iterative stage the denoising process should be the same, then why use something like $(T-T_0)/R$ which seems like the average length of the denoising step? If this means the step length, then what is $k$ and $T_0+k$? When $R=1$, $T_0$ mod $R = 0$ and $k=T$. I try my best but cannot understand this part.

**Questions:**

Could the authors provide more detailed ablation on the functionalities under both guidance scale and noise level? For example, effects of guidance scale under different noise level, or what if guidance scale increases but noise level decreases? The ablation of multi-factor could offer deeper insight of the pipeline and make it easy to use.

---

> ### Author Response · Authors · 2025-11-21
>
> Dear Reviewer, we thank you for your comments and feedback on our paper.
>
> >  Despite the solid theoretical part, the proposed pipeline is relatively not well-explored. First, as is reported in Fig. 3, increasing Repetitions fails to further improve the performance. Involving only one or two iterations seems a local correction or refinement and inconsistent with "iterative guidance", since most denoising steps before  fall back to the native CFG with a smaller weight. Besides, as is claimed in Prop. 2, the discrepancy is an  term when  is small. This seems to conflict with the results in Fig. 3, in which extremely small  almost corrupts both FID and FD_DINOv2. Note that EDM2 does not need large CFG weight, why is performance so sensitive to ?
>
> The reported behavior in Figure 3 is primarily explained by the fixed compute budget of diffusion steps. As noted in the original submission, the number of diffusion steps is held constant. Under this setting, increasing the number of refinement stages $R$ reduces the number of denoising steps allocated per refinement stage and thus leads to suboptimal sample quality. We clarified this trade-off discussion in the revised manuscript and referenced it more explicitly alongside Appendix Figure 10 where we analyse this trade-off in the Gaussian case.
>
> Following your suggestion (and as also reflected in our response to your question), we added an ablation in which the compute budget scales with the number of refinement stages $R$; the number of diffusion steps per refinement stage is fixed, however. We then plot the performance as a function of $\sigma_*$. The results are presented in Figure 7 of the updated paper. We see that the performance improves with increased $R$ within a range of $\sigma_*$. Past this range the performance degrades for the FID only. Actually we have observed a similar trend for our Gaussian analysis in Figure 10, which is backed by clear theoretical formulas. When $\sigma_*$ is relatively large, the bias increases and using more refinement steps does not necessarily improve the performance.
>
> > This seems to conflict with the results in Fig. 3, in which extremely small $\sigma_*$ almost corrupts both FID and FD_DINOv2. Note that EDM2 does not need large CFG weight, why is performance so sensitive to $\sigma_*?
>
> For the Gaussian example we also observe that if $\sigma_*$ is too small then the mixing is too slow and $R$ has to be very large; see Figure 3. We believe this behavior is intuitive: after the initial unguided stage, the generated samples are still of relatively low quality, and adding only a very small amount of noise limits how much they can evolve. As a result, many more refinement steps are needed before noticeable improvement occurs.
>
> > Finally, to more properly use the current name "iterative guidance", could the authors come up with some pipelines to shorten the native CFG period and use more iterations and some dynamic $\sigma_*$ and guidance weights to better follow the theory?
>
> Thank you for this interesting suggestion. Unfortunately, given the time constraints, we are not able to provide proper adaptive methods and we leave this for future work.
>
> > The current notation system is kind of hard to follow. Four notations p, q, g, … all represent probabilities, making the understanding a bit difficult.
>
> We apologize if aspects of the notation hindered the review process and appreciate your feedback. We have revised the notation accordingly in the updated manuscript. In the revised version, we consolidate notation by using a single symbol, $p$, for both conditional and unconditional distributions to improve readability.
>
> We hope this addresses your concern, and we remain open to further suggestions.
>
> >  Besides, what does the last term in ODE_Solver in Alg. 1 mean? From my understanding in the iterative stage the denoising process should be the same, then why use something like $(T - T_0) / R$ which seems like the average length of the denoising step? If this means the step length, then what is $k$ and $T_0 + k$? When $R=1$, $T_0  mod $R$ and $k = T$. I try my best but cannot understand this part.
>
> Thank you for pointing this out. Regarding the final term in the call to ``ODE_Solver`` in Algorithm 1, the last argument specifies the number of denoising steps used. Concretely, $T$ denotes the total number of diffusion steps, $T_0$ the number of steps used during initialization, and $R$ the number of refinement stages. Under equal allocation, each refinement stage therefore uses $(T - T_0)/R$ denoising steps. When $(T - T_0)$ is not divisible by $R$, the remaining steps $k$ are added to the initialization stage.
>
> To avoid confusion, we revised Algorithm 1 to omit these implementation details and instead describe them in the main text. We also provide an explicit implementation example in Appendix Algorithm 2 to illustrate the argument parameters of ``ODE_Solver``.

---

> > ### Author Response · Authors · 2025-11-21
> >
> > > Could the authors provide more detailed ablation on the functionalities under both guidance scale and noise level? For example, effects of guidance scale under different noise level, or what if guidance scale increases but noise level decreases? The ablation of multi-factor could offer deeper insight of the pipeline and make it easy to use.
> >
> > We thank you for the suggestion. We have added an ablation study examining how different guidance scales interact with the small-noise level $\sigma_*$ under a fixed compute budget of diffusion steps; see Figure 5. The key observation is that stronger guidance shifts the optimal value of $\sigma_*$ toward smaller values. This effect arises because larger guidance increases the contribution of the Rényi divergence term, thereby amplifying the bias introduced when it is omitted. As a result, the range in which neglecting this term can be acceptable becomes narrower.
> >
> > **Further changes in the paper**: we have improved the presentation of the paper by providing the main steps of the proofs.
> >
> > We again thank you for your constructive feedback and suggestions. We remain open to further questions/suggestions you may have.

---

> > > ### Comment · Reviewer_M9WA · 2025-11-23
> > >
> > > Thanks for the clear clarifications and the additional experiments with great efforts. To summarize the rebuttal part, The native CFG process and the proposed iterative refinement process somewhat **face with trade-offs**, *i.e.*, for a fixed total NFEs, too many iterative refinements will harm the synthesis of native CFG. Consequently, I have two more questions:
> > >
> > >   1. Despite the extensive ablations, the whole pipeline seems sensitive to hyper-parameters, and the choice of hyper-parameters is somewhat ad-hoc and empirical. Could the authors provide more experimental examples on other models or CFG weights, *e.g.*, DiT or EDM2 with unofficially recommended CFG weights?
> > >
> > >   2. I am curious about the functionality of **only increasing number of iterations while fixing native CFG process**. For example, under EDM2 settings, 30 native CFG steps + 2 iterative steps v.s. 30 native CFG steps + 3/4/5 iterative steps. Given that EDM2 involves 32 denoising steps (NFE = 63), I fail to understand why 30 native CFG steps + 2 iterative steps is better than 29 native CFG steps + 3 iterative steps. The difference or accuracy of 30 or 29 denoising steps should be marginal from my perspective.

---

> ### Author Response · Authors · 2025-11-23
>
> We thank you for the prompt reply.
>
> > I am curious about the functionality of only increasing number of iterations while fixing native CFG process. For example, under EDM2 settings, 30 native CFG steps + 2 iterative steps v.s. 30 native CFG steps + 3/4/5 iterative steps. Given that EDM2 involves 32 denoising steps (NFE = 63), I fail to understand why 30 native CFG steps + 2 iterative steps is better than 29 native CFG steps + 3 iterative steps. The difference or accuracy of 30 or 29 denoising steps should be marginal from my perspective.
>
> To avoid potential misconceptions, we would like to clarify the structure of our generation pipeline which consists of two subcomponents:
>
> - **Initialization**: A total of $T_0$ diffusion steps are performed using a denoiser with moderate or no guidance scale (as for the EDM experiments). This stage produces diverse but coarse samples.
> - **Refinement**: we perform $R$ refinement stages, each consisting of $T_R$ diffusion steps, to progressively improve the initial samples.
>
> Please see Lines 301–309 of our paper for more on this description. Under this setup, the total compute budget is given by $T = T_0 + R \times T_R$
>
> In the fixed compute budget scenario (only $T$ denoising steps are performed in total), we set $T_R = (T - T_0) / R$.  For EDM models specifically, we use $T_0 = 12$ (not 29 or 30 steps). With this choice, each refinement stage uses $T_R = (32 - 12) / 2 = 10$ diffusion steps. Please refer to Table 4 in the appendix where we provide the used hyperparameters for each model.
>
> Among others, the conducted ablations evaluate different values of $R$ under a fixed compute budget (here, increasing $R$ decreases $T_R$). These results are shown in Figure 3. Following your suggestion in the initial review, we additionally include a setting where the compute budget is dynamic (here $T_R$ is fixed hence increasing $R$ increases the total compute budget $T$). The results for this variant are presented in Figure 7.
>
> We kindly point you towards the ablation section of our paper where we discuss the behavior of the method on these different settings. We remain open for further inquiries/suggestions.
>
> > Despite the extensive ablations, the whole pipeline seems sensitive to hyper-parameters, and the choice of hyper-parameters is somewhat ad-hoc and empirical. Could the authors provide more experimental examples on other models or CFG weights, e.g., DiT or EDM2 with unofficially recommended CFG weights?
>
> As with any method, our proposed method has also hyperparameters, which arise from the theoretical framework and the adopted MCMC approach. We provide intuition for each hyperparameter and discuss how it governs the relevant trade-offs in the framework. We further assess the sensitivity of the method to these choices and observe trends that are consistent with our theoretical analysis.
>
> Unless we’ve misunderstood your question, we believe that Figures 3, 5 and 6 already show how the algorithm behaves under unofficially recommended CFG weights (for instance AudioLDM 2 model). Figure 3 shows how both FID and DINOV2 behave as a function of the CFG weight while all the other parameters are fixed to the ones in Table 4. We do the same in Figure 6 for the audio model and then in Figure 5 we show how the FID and DINOV2 behave when we vary CFG weight and $\sigma_*$ at the same time.
>
> We again thank you for the time and effort spent reviewing our work. We remain open to further suggestions and inquiries.

---

> > ### Comment · Reviewer_M9WA · 2025-11-24
> >
> > Sorry for the misunderstanding about the first part.
> >
> > As for the second part, I hope the authors could summarize some general rules about the hyper-parameters both empirically and theoretically (*e.g.*, the scale or trend of the hyper-parameters), so that when facing a new task one can easily set a proper group of hyper-parameters.

---

> > > ### Author Response · Authors · 2025-11-24
> > >
> > > Regarding the hyperparameters, the various ablations in the paper suggest that the most critical parameter (besides the guidance scale) is the noise level $\sigma_*$. Figure 5 shows for example that the FID can drop rapidly past a certain $\sigma_*$ threshold, but before this threshold the performance improves predictably as a function of $R$. Thus, we recommend performing a grid search on $\sigma_*$ first. Once the optimal parameter is determined, one can try a couple of values of $R$, i.e. 1, 2, and 3. The rule of thumb is to ensure that each refinement stage has sufficient diffusion steps, i.e at least 10 diffusion steps
> > > For the initialization stage, the guidance scale $w_0$ should be kept small close to $1$ to preserve the diversity of the target distribution. As for the number initialization steps $T_0$, we recommend allocating one-third to one-half of the total compute budget.
> > >
> > >
> > > We would like to once again thank you for engaging with us in this discussion and we highly value your feedback on our work. In light of our exchange, we would be grateful if you could share any other suggestions on how we might further improve the clarity of the overall presentation of the paper as you have assessed it as “fair”.

---

> > > > ### Comment · Reviewer_M9WA · 2025-11-25
> > > >
> > > > Thanks for further clarifications. It would be a pity there is some potential grid search on hyper-parameters. I highly recommend to leave as some future works.
> > > >
> > > > As for the presentation rank, it is mainly because the unclear algorithm description and notations. I would like to change my rank and the score as well.

---

> > > > > ### Author Response · Authors · 2025-11-25
> > > > >
> > > > > We appreciate your response.
> > > > >
> > > > > > Thanks for further clarifications. It would be a pity there is some potential grid search on hyper-parameters. I highly recommend to leave as some future works.
> > > > >
> > > > > In fact, mitigating the hyperparameter search is part of our on-going work. In the revisited version, we will include the hyperparameters selection guidelines outlined in our previous comment.
> > > > >
> > > > > We sincerely thank you for your input throughout the discussion which helped improve the clarity and overall presentation of our work.

---

### Official Review · Reviewer_4nXS · 2025-11-01

**Soundness:** 2
**Presentation:** 2
**Contribution:** 1
**Rating:** 2
**Confidence:** 4

**Summary:**

This paper investigates improving the widely used classifier-free guidance (CFG) to simultaneously enhance sample quality and diversity. The authors first provide a theoretical analysis, formulating the gap between the ideal conditional score and the CFG score (with guidance scale w) using the Rényi divergence of order w. Separately, they introduce a two-stage sampling scheme called Classifier-Free Iterative Guidance (CFIG). In the first stage (a traditional CFG procedure), a probability flow ODE is integrated from a high noise level (sigma_max) using a moderate guidance scale w_0 to obtain diverse initial samples. In the second stage, these samples are repeatedly refined by being perturbed to a low noise level (sigma_*) and then denoised by solving a reverse SDE guided by a larger CFG scale w. The method is evaluated on class-conditional image generation and text-to-audio synthesis tasks, with ablation studies conducted on key hyperparameters.

**Strengths:**

1. Code is provided in the supplementary materials, which facilitates reproducibility.
2. The paper studies classifier-free guidance (CFG), an important component in diffusion-based generative models that is widely applicable across different diffusion frameworks.
3. A theoretical analysis is presented to characterize the gap between the ideal conditional score and the CFG score.
4. A new sampling procedure is proposed to potentially improve generation quality.

**Weaknesses:**

1. The connection between the proposed CFIG method (Section 4) and the theoretical analysis (Section 3) appears weak. None of the key parameters -- such as sigma_*, T-T_0, or R -- is quantitatively derived from the theoretical results. Moreover, the central idea of CFIG, namely iterative refinement on a weakly guided CFG results, seems only tenuously related to the theoretical observations in Section 3. The current presentation overcomplicates a rather straightforward method, which may obscure the core insights and hinder the reader’s understanding.
2. The idea of *iteratively perturbing samples to higher noise levels and then re-denoising them* has been widely explored in prior diffusion sampling methods [1][2][3] for result refinement. The paper lacks a clear discussion of how the proposed CFIG method relates to, or differs from, these existing approaches.
[1] Yu, Jiwen, et al. "Freedom: Training-free energy-guided conditional diffusion model." Proceedings of the IEEE/CVF International Conference on Computer Vision. 2023.
[2] Bansal, Arpit, et al. "Universal guidance for diffusion models." Proceedings of the IEEE/CVF conference on computer vision and pattern recognition. 2023.
[3] Ma, Nanye, et al. "Inference-time scaling for diffusion models beyond scaling denoising steps." arXiv preprint arXiv:2501.09732 (2025).
3. The experimental results are not convincing, and several important details remain unclear.
a. The performance improvement over INTERVAL-CFG in the image generation experiments is marginal and could easily fall within the range of randomness.
b. According to Algorithm 1, CFIG appears to use a CFG denoiser to obtain the initial samples, implying a runtime at least comparable to that of CFG. However, the appendix (“Comments on runtime and memory”) claims that CFIG first generates an initial sample using the plain denoiser, resulting in a faster process. This inconsistency makes it unclear how CFG is actually applied in the first stage.
c. The paper reports only the number of sampling steps when comparing with prior works. It would be more appropriate to report the number of function evaluations (NFE), as different sampling methods entail varying numbers of model forward passes during guided generation.

**Questions:**

1.The paper presents many conclusions and propositions but lacks intuitive explanations to help readers grasp the core ideas. For instance, what specifically causes the gap (the gradient of the Rényi divergence) between the ideal score and the CFG score? Why is this score gap bounded by \sigma^2? I suggest that the authors include the key steps or intuitions behind the proofs in the main text, rather than placing all derivations in the appendix.
2.The text-to-audio generation experiment is not necessary for evaluating the proposed methods, though CFIG shows relatively better performance than in the image experiments.
3.A more convincing ablation would be directly use the generated result of CFG/INTERVAL-CFG as the initial sample, and then apply the iterative resampling process to it to see to what extent can this process refine the initial sample.
4.The iterative resampling procedure is, in principle, not tightly coupled with CFG. It would be interesting to investigate whether this iterative refinement is also effective for non-guided sampling processes.

---

> ### Author Response · Authors · 2025-11-21
>
> Dear Reviewer, we thank you for your comments and feedback on our paper.
>
> > The connection between the proposed CFIG method (Section 4) and the theoretical analysis (Section 3) appears weak. None of the key parameters -- such as sigma_*, T-T_0, or R -- is quantitatively derived from the theoretical results. Moreover, the central idea of CFIG, namely iterative refinement on a weakly guided CFG results, seems only tenuously related to the theoretical observations in Section 3. The current presentation overcomplicates a rather straightforward method, which may obscure the core insights and hinder the reader’s understanding.
>
> We thank you for pointing out that the connection between our theoretical analysis and its practical implications may not be sufficiently clear. We have revised the manuscript to clarify this crucial link, which we detail below as well.
>
> First, we would like to emphasize that one core contribution of our paper is the theoretical insights on CFG: the characterization of the missing term, how it behaves as $\sigma$ goes to $0$ and then the novel expression of the score of the tilted distribution. These findings stand on their own as novel theoretical insights on CFG. The proposed algorithm is presented as a practical byproduct of this analysis, an illustration of how these theoretical results can be translated into a refinement procedure.
>
> Next, our result in Proposition 2 directly informs our central design choice: the noise level $\sigma_*$ in Algorithm 1 must be kept sufficiently small to ensure valid approximations. We observe this empirically for a toy example in Appendix B.3 Figure 10 but also on images and audio; see the evolution of the metrics as a function of $\sigma_*$ in Figure 3 and 6. In particular, see trade-off in FID and FD-DINOV2 for images. We observe the same core insights; if the noise level is too small then the MCMC chain mixes too slowly, and if it is too large the bias increases too much. Hence finding a good balance for $\sigma_*$ is crucial. The other hyperparameters, such as $T-T_0$ and $R$, are related to the MCMC procedure itself (governing initialization and iterations) and hence are not derived from the analysis in Section 3.
>
> > The idea of iteratively perturbing samples to higher noise levels and then re-denoising them has been widely explored in prior diffusion sampling methods [1][2][3] for result refinement. The paper lacks a clear discussion of how the proposed CFIG method relates to, or differs from, these existing approaches. [1] Yu, Jiwen, et al. "Freedom: Training-free energy-guided conditional diffusion model." Proceedings of the IEEE/CVF International Conference on Computer Vision. 2023. [2] Bansal, Arpit, et al. "Universal guidance for diffusion models." Proceedings of the IEEE/CVF conference on computer vision and pattern recognition. 2023. [3] Ma, Nanye, et al. "Inference-time scaling for diffusion models beyond scaling denoising steps." arXiv preprint arXiv:2501.09732 (2025).
>
> We already noted  in our related work section that iterative refinement is not a new idea, in particular, it was initially introduced for image editing in SDEdit [Meng et al. 2021]. We thank you for the additional reference. We have added them to the revised manuscript with a discussion of their relation to our work. We wish to highlight, however, that these works, apply iterative refinement in other settings, namely [Yu, Jiwen, et al. 2023] and [Bansal, Arpit, et al. 2023] for guiding a (potentially unconditional) DDMs with an auxiliary guidance function, and [Ma, Nanye, et al. 2025] for improving the inference time scaling for DDMs. To the best of our knowledge, the application of this technique in the context of CFG for conditional generation has not been explored before.
>
> > The experimental results are not convincing, and several important details remain unclear. a. The performance improvement over INTERVAL-CFG in the image generation experiments is marginal and could easily fall within the range of randomness.
>
> We acknowledge that the results for class conditional image generation on ImageNet between CFIG and INTERVAL-CFG are close. As our proposed method is derived from the presented theoretical framework that analyzes the bias of CFG, we view these tied results as evidence for the correctness of our theoretical framework. Moreover, CFIG achieves an improvement in the text-to-image and text-to-audio generation task over interval-CFG, which is a good indicator of its applicability across modalities. Overall, we believe that our experiments show that CFIG offers a good balance across all the tasks compared among the evaluated methods, making it a relevant contribution.

---

> > ### Author Response · Authors · 2025-11-21
> >
> > > b. According to Algorithm 1, CFIG appears to use a CFG denoiser to obtain the initial samples, implying a runtime at least comparable to that of CFG. However, the appendix (“Comments on runtime and memory”) claims that CFIG first generates an initial sample using the plain denoiser, resulting in a faster process. This inconsistency makes it unclear how CFG is actually applied in the first stage.
> >
> > We thank you for identifying this discrepancy in our description. Our method uses a moderate guidance scale $w_0$  for initialization. While we used $w_0=1$ (the plain denoiser) for conditional image generation tasks, we indeed used $w_0=1.5$ for audio experiments and $w_0=2$ for SDXL, contrary to our initial appendix note which was meant for the conditional image generation tasks. We have rectified this statement and updated the corresponding discussion in the revised version. We also provided a clearer explanation on our choice of initial sample in Section 4.
> >
> > > c. The paper reports only the number of sampling steps when comparing with prior works. It would be more appropriate to report the number of function evaluations (NFEs), as different sampling methods entail varying numbers of model forward passes during guided generation.
> >
> > We apologize if our formulation regarding the computation budget was unclear and welcome the suggestion of adding NFEs. We now detail the NFEs, namely the number of CFG denoiser calls, number of plain denoiser call, and the total in terms of plain denoiser calls in Table 7 of the appendix for a more comprehensive comparison between the methods across the considered experiments.
> >
> > > The paper presents many conclusions and propositions but lacks intuitive explanations to help readers grasp the core ideas. For instance, what specifically causes the gap (the gradient of the Rényi divergence) between the ideal score and the CFG score? Why is this score gap bounded by \sigma^2? I suggest that the authors include the key steps or intuitions behind the proofs in the main text, rather than placing all derivations in the appendix.
> >
> > We thank you for this suggestion. We now include proof sketches for all the Propositions to provide the reader with more intuition. We hope that it addresses your concerns and are open to further discussions if needed.
> >
> > > The text-to-audio generation experiment is not necessary for evaluating the proposed methods, though CFIG shows relatively better performance than in the image experiments.
> >
> > We believe the audio experiments are valuable for two reasons. First, they provide additional quantitative results on the competitiveness of our method. Second, they demonstrate applicability of our method across different data modalities. In fact, Reviewer (UCMX) has specifically requested experiments on an additional third modality (video) to further enrich the experimental validation.
> >
> > > A more convincing ablation would be directly use the generated result of CFG/INTERVAL-CFG as the initial sample, and then apply the iterative resampling process to it to see to what extent can this process refine the initial sample
> >
> > CFIG comprises two components: (1) generating initial samples, (2) refining samples. As you noticed, the initial sample phase can be performed with virtually any sampler. The critical point is that the sampler used should maintain the diversity of the base model as much as possible. To achieve this, we use in class conditional image generation the plain denoiser ($w_0=1$, hence no CFG) and then the CFG denoiser with weak guidance scale (moderate value of $w_0$) for text-to-image and text-to-audio generation tasks.
> >
> > We thank you for suggesting the use of INTERVAL-CFG. We have conducted a new ablation, taking the same setup as the one associated with Table 2, but where we run CFIG starting from a sample from INTERVAL-CFG. We basically ran it for $T_0$ steps with the same hyperparameters as Table 5. We report the results in Table 8 of the appendix side-by-side with CFIG and INTERVAL-CFG. Among others, CFIG with INTERVAL-CFG as the first stage shows close results in FID and improved performance in FD_DINOv2. Similar results are observed also for Precision and Density/Coverage. These findings suggest that using a more informative initialization than a plain denoiser (or a weakly guided CFG sample) can be beneficial. We note that this is a preliminary ablation without hyperparameter tuning, and additional improvements may be possible with further optimization.
> >
> > > The iterative resampling procedure is, in principle, not tightly coupled with CFG. It would be interesting to investigate whether this iterative refinement is also effective for non-guided sampling processes.
> >
> > We thank you for this observation. However, considering refinements similar to the one we propose in our paper to other settings than CFG appear to us out of the scope of the current submission and leave it for future work.

---

> > > ### Author Response · Authors · 2025-11-21
> > >
> > > **Further changes in the paper**: Additionally to providing the main steps for the proofs, we have overall improved the presentation of the paper following a reviewer’s suggestions by simplifying our notation. We also included
> > > - Figure 5: that analyzes the behavior with different guidance scales as a function of $\sigma_*$
> > > - Figure 7: That analysis the behavior for dynamical computation budget that scales with $R$, here again as a function of $\sigma_*$.
> > > - Table 7: details analysis of Number of Function Evaluation of each method across experiments
> > >
> > > We again thank you for the detailed feedback. We are genuinely surprised by the low scores for contribution and soundness. We would greatly appreciate it if you could reconsider your evaluation of our work in light of the rebuttal. We remain open to further questions/suggestions you may have.

---

### Official Review · Reviewer_UCMX · 2025-11-01

**Soundness:** 3
**Presentation:** 3
**Contribution:** 2
**Rating:** 6
**Confidence:** 3

**Summary:**

This paper theoretically analyzes the inconsistency problem in classifier-free guidance for conditional diffusion models and traces its root cause. The authors rigorously prove that the widely used linear combination of conditional and unconditional denoisers in CFG does not correspond to any valid denoising diffusion process. Through detailed mathematical derivation, they identify a missing gradient term of the Rényi divergence. Based on this analysis, the authors propose a new sampling procedure named Classifier-Free Iterative Guidance. The method first generates an initial sample using a weak or zero guidance scale, and then performs multiple rounds of re-noising and denoising under a normal guidance scale. This iterative process gradually refines sample quality while preserving diversity. Extensive experiments on several tasks demonstrate the effectiveness of the proposed method.

**Strengths:**

1. The paper provides a clear and rigorous derivation showing CFG’s theoretical inconsistency and identifies the missing Rényi divergence term that explains its mode-collapse behavior.
2. The proposed CFIG algorithm is easy to implement since it only modifies the sampling phase, requiring no model retraining. It integrates seamlessly with existing diffusion backbones, making it practically useful for real-world applications.

**Weaknesses:**

1. Although the paper theoretically derives the Rényi divergence correction, it is not actually incorporated in the proposed algorithm. It's a bit of a shame that CFIG circumvents this problem by using a low-guided approximation instead of explicitly using the Rényi correction.
2. CFG exhibits different problems in different modal generation tasks. The paper validates its performance on image and audio generation. I'd like to see its performance on more modalities, such as video generation tasks.
3. What do “CFGIG” stands for in Figure 6 and Figure 7?
4. In the image experiment, what is the R value of CFIG? Given that the quantitative metrics are comparable to INTERVAL-CFG without significant improvement, if R is greater than 1, does this imply that more denoising steps were used only to achieve performance similar to that of INTERVAL-CFG? If so, it would indicate a lack of cost-effectiveness in comparison.
5. Following up on the previous question, there is a lack of experimental analysis regarding efficiency.
6. Considering that DDMs are more commonly used for generation tasks, the absence of qualitative comparisons makes it difficult to evaluate their practical application value.

**Questions:**

Please refer to the questions and suggestions in the “Weaknesses” part.

---

> ### Author Response · Authors · 2025-11-21
>
> Dear Reviewer, we thank you for your comments and feedback on our work.
>
> > Although the paper theoretically derives the Rényi divergence correction, it is not actually incorporated in the proposed algorithm. It's a bit of a shame that CFIG circumvents this problem by using a low-guided approximation instead of explicitly using the Rényi correction.
>
> Thank you for raising this point. We initially sought to incorporate the Rényi divergence correction directly. However, this quantity is computationally intractable, as its calculation involves the distributions $p( . | x_\sigma, c)$ and $p( . | x_\sigma)$, which are not available in closed form in general. We experimented with several approximations, but the results were inconclusive. If you believe it would be helpful, we would be glad to include a summary of these attempts in the Appendix.
> This challenge motivated the development of the alternative workaround we propose, which remains firmly based on our theoretical analysis since it directly leverages a key conclusion from Section 3, namely that the Rényi divergence term is negligible when $\sigma_*$ is small.
>
> > CFG exhibits different problems in different modal generation tasks. The paper validates its performance on image and audio generation. I'd like to see its performance on more modalities, such as video generation tasks.
>
> We appreciate your suggestion. We agree that validation on video generation is an important direction, and we have striven to demonstrate the versatility of our method through its application across several distinct modalities: class-conditional image generation, text-to-image, and text-to-audio.
>
> Unfortunately, due to the current time constraints and our limited computational resources, we are unable to include a full experiment on video generation with proper hyperparameter tuning for our method and baselines. We will prioritize adding this modality to our set of experiments for the final version or future work.
>
> > In the image experiment, what is the R value of CFIG? Given that the quantitative metrics are comparable to INTERVAL-CFG without significant improvement, if R is greater than 1, does this imply that more denoising steps were used only to achieve performance similar to that of INTERVAL-CFG? If so, it would indicate a lack of cost-effectiveness in comparison.
>
> In the image experiments, we used $R=2$ for CFIG. Please also note that all the hyperparameters for each experiment are specified in Table 4 in the appendix. We would like to clarify that all methods were compared under a fixed compute budget of 32 denoising steps for image modalities and 200 for audio, as stated in our original manuscript (lines 376-377).  To be more specific, for CFIG, the total number of denoising steps is $R \times$ (steps per refinement) plus the number of initial steps $T_0$. Therefore, increasing $R$ would necessarily reduce the number of steps available for each refinement under this fixed budget. To prevent any future confusion, we have now emphasized this explanation regarding the constant computational budget in the revised manuscript.
>
> > Following up on the previous question, there is a lack of experimental analysis regarding efficiency.
>
> We would be very grateful for any specific suggestions to further strengthen the efficiency analysis. We believe that our paper already includes such an experimental analysis; please see Figure 10 and the accompanying discussion in Appendix B.3. We have also added a new figure in the main paper (Figure 7) where we show how our method behaves under increased computational budget; we increase $R$ while maintaining the number of steps per refinement stage fixed, while varying $\sigma_*$. The trade-off we observe is coherent with the theoretical finding in Figure 10. We acknowledge that this analysis was not mentioned in the main paper; this is now the case in the revised version.
>
>
> > Considering that DDMs are more commonly used for generation tasks, the absence of qualitative comparisons makes it difficult to evaluate their practical application value.
>
> Thank you for this feedback. To ensure we address your concern accurately, could you please clarify what you mean by "the absence of qualitative comparisons"? Our current understanding is that the qualitative results in Appendix E, where we display sample images of CFIG alongside the considered baselines, serve this purpose. So we want to be certain we are aligned. We are happy to provide further analysis based on your guidance.

---

> ### Author Response · Authors · 2025-11-21
>
> **Further changes in the paper**: we have improved the presentation of the paper following other reviewers’ suggestions by simplifying our notation, providing the main steps of the proofs, We also included
> - Figure 5: that analyzes the behavior with different guidance scales
> - Figure 7: That analysis the behavior for dynamical computation budget that scales with $R$
> - Table 7: details analysis of Number of Function Evaluations (NFEs) of each method across experiments
>
> We again thank you for your constructive feedback and suggestions and for the typo (in Weakness 3) you pointed out which has now been corrected. We remain open to further questions/suggestions you may have.

---

### Meta-Review · Area_Chair_qxEW · 2026-01-10

**Summary:**

The paper presents a theoretical analysis of the inconsistency in classifier-free guidance (CFG) for diffusion models, identifying a missing Rényi divergence term. It then proposes a new sampling algorithm, Classifier-Free Iterative Guidance (CFIG), which uses iterative refinement to improve sample quality and diversity. While reviewers appreciated the theoretical rigor and clarity of the derivation, significant concerns were raised regarding the connection between theory and method, novelty, experimental strength, and presentation.

The proposed algorithm does not directly incorporate the derived Rényi divergence correction, instead using a low-guided approximation.

The iterative refinement idea is not novel and has been explored in prior work (e.g., SDEdit, Freedom, Universal Guidance).

Experimental results are marginal (e.g., small improvements over INTERVAL-CFG) and not convincingly significant; inconsistencies in algorithm description; lack of proper.

Request for more ablation studies on guidance scale and noise level interaction.

**Reviewer Concerns:**

Addressed Concerns:

Provided qualitative comparisons in Appendix E and requested clarification on additional needs.

Added ablation on the guidance scale and noise level.

Outstanding Concerns:

The connection between theory and method remains tenuous; parameters are not theoretically derived, and the method is presented as a practical byproduct.

Experimental improvements are marginal and may not be significant.

**Reviewer Scores:**

Reviewer UCMX (initial 6): Likely score unchanged at 6.

Reviewer 4nXS (initial 2): Likely score unchanged at 2.

Reviewer M9WA (initial 6): Likely score unchanged at 6.

---

### Decision · Program_Chairs · 2026-01-26

Reject